**Technical Report**

# Visualizing suborganellar lipid distribution using correlative light and electron microscopy

H. Mathilda Lennartz ®[1]✉, Suman Khan ®[2,4], Weihua Leng[1,4], Kristin Böhlig[1], Gunar Fabig ®[3], Yannick Kieswald ®[1], Falk Elsner[1], Nadav Scher[2], Michaela Wilsch-Bräuninger[1], Ori Avinoam ®[2] & André Nadler ®[1]✉

Lipids and proteins compartmentalize biological membranes into nanoscale domains, which are crucial for signalling, intracellular trafficking and many other cellular processes. Studying nanodomain function requires the ability to measure protein and lipid localization at the nanoscale. Current methods for visualizing lipid localization do not meet this requirement. Here we introduce a correlative light and electron microscopy workflow to image lipids (Lipid-CLEM), combining near-native lipid probes and on-section labelling by click chemistry. This approach enables the quantification of relative lipid densities in membrane nanodomains. We find differential partitioning of sphingomyelin into intraluminal vesicles, recycling tubules and the boundary membrane of the early endosome, representing a degree of nanoscale organization previously observed only for proteins. We anticipate that our Lipid-CLEM workflow will greatly facilitate the mechanistic analysis of lipid functions in cell biology, allowing for the simultaneous investigation of proteins and lipids during membrane nanodomain assembly and function.

The eukaryotic cell is compartmentalized by membranes, which are organized into functional nano- and mesoscale domains that feature distinct geometries, lifetimes and protein compositions. Membrane domains have been implicated in cell signalling[1–3], cell adhesion[3,4], membrane remodelling[1,2,5] and trafficking[5,6]. In contrast to the well-studied protein composition of membrane domains, much less is known about their lipid content as visualizing individual lipid species at the ultrastructural level is largely an unsolved methodological problem[7,8]. Key challenges include rapid lipid dynamics[9] and localization artefacts introduced by fluorescently labelled lipid probes[10–12]. A suitable lipid visualization technique should report on molecularly distinct lipid species rather than entire lipid classes and employ minimally modified lipid probes to avoid artefacts. Furthermore, membrane ultrastructure must be preserved and spatially resolved. Finally, high temporal

resolution during sample preparation must be achieved to capture dynamic membrane remodelling events.

Bifunctional diazirine-alkyne lipid probes ('bifunctional lipids' in the following) are a powerful tool to visualize lipid localization by fluorescence microscopy (FM) in situ[13–18]. Bifunctional lipids closely mimic native lipid species as their small chemical modifications limit localization artefacts to a minimum[19–21]. We have previously shown that they are metabolized similarly to native lipids, and that the combined effect of the diazirine and alkyne modifications is roughly equivalent to that of one additional double bond, both with regard to biophysical properties and lipid metabolism[21]. Bifunctional lipids can be introduced into membranes of living cells and rapidly photo-crosslinked to neighbouring proteins. The resulting lipid–protein conjugates can, in turn, be fluorescently labelled post fixation by click chemistry,

[1]Max Planck Institute of Molecular Cell Biology and Genetics, Dresden, Germany. [2]Department of Biomolecular Sciences, Weizmann Institute of Science, Rehovot, Israel. [3]Experimental Center, Faculty of Medicine Carl Gustav Carus, Technische Universität Dresden, Dresden, Germany. [4]These authors contributed equally: Suman Khan, Weihua Leng. ✉e-mail: hlennart@mpi-cbg.de; nadler@mpi-cbg.de

circumventing the use of heavily modified lipid probes in living cells while capturing lipids that cannot be chemically fixed otherwise[19,21]. Fluorescence lipid imaging based on bifunctional probes has been instrumental for mapping interorganelle lipid distribution and lipid transport pathways[21,22].

Measuring lipid localization at the nanoscale, however, requires a resolution that light microscopy alone cannot provide[23]. A correlative light and electron microscopy (CLEM) approach[24] to measure lipid densities within nanodomains would allow us to investigate lipid localization–function relationships[25], for example, regarding lipid sorting mechanisms during membrane trafficking[26]. CLEM approaches to image lipids have been reported previously by us[14] and others[27,28]. The required sample processing procedures of these approaches, however, are typically not ideal for the preservation of membrane ultrastructure, are limited to studying the outer leaflet of the plasma membrane or target entire lipid classes.

Here, we report a Lipid-CLEM workflow based on bifunctional lipid probes for faithfully visualizing lipids on the ultrastructural level. Our CLEM workflow is optimized for membrane ultrastructure preservation, sample preparation speed and correlation precision. We used a sequence of rapid lipid photo-crosslinking[20] and high-pressure freezing[29,30] to arrest lipid localization within the sample, followed by freeze substitution[29], sectioning and on-section fluorescence labelling of lipid probes by click chemistry. The combined time required for crosslinking and high-pressure freezing (approximately 10 s) allows for capturing membrane domains with lifetimes in the second to minute range. We developed Lipid-CLEM variants that either feature labelled bifunctional lipids solely on the surface of the section (optimized for membrane ultrastructure preservation) or throughout the section (optimized for lipid density measurements).

Using whole-section labelling, we investigated the capacity of the early endosome to sort lipids. It is well established that proteins within the early endosome segregate into distinct membrane domains as a prerequisite for sorting into different trafficking pathways[31-34]. For instance, transferrin (Tf) segregates into recycling tubules (RTs) to be trafficked back to the plasma membrane[32,34] and low-density lipoprotein (LDL) particles accumulate in the endosomal lumen for transport towards lysosomes[33,34]. Whether lipids are sorted in a similar fashion during retrograde membrane trafficking is not known[24]. Several studies have addressed this question indirectly by quantifying uptake of lipid–fluorophore conjugates into endosomes[35,36], but owing to the experimental set up, direct partitioning into endosomal subcompartments could not be assessed. Using Lipid-CLEM we measured the density of a single sphingomyelin (SM) species in the subcompartments of individual early endosomes. We found that SM is relatively enriched in intraluminal vesicles (ILVs) and depleted from RTs, suggesting active lipid sorting mechanisms within the early endosome. Furthermore, protein and lipid cargoes exhibit distinctly different localization patterns within the endosomal membrane system, implying that lipid and protein cargo trafficking routes diverge in the early endosome.

## Results

### Bifunctional lipid probes enable on-section Lipid-CLEM

In principle, Lipid-CLEM can be performed using cryo- or room temperature workflows. Cryo-CLEM facilitates the highest membrane preservation but necessitates the use of fluorescently labelled lipids in the living cell. On the other hand, room temperature CLEM allows for the use of near-native bifunctional lipid probes while compromising on ultrastructural preservation (Extended Data Fig. 1a). To assess the viability of both strategies, we first compared two phosphatidylcholine (PC) derivatives, a fluorescent nitrobenzoxadiazole (NBD)-PC and its near-native bifunctional PC derivative bearing a palmitate at the *sn*-1 and the bifunctional fatty acid (Y) at the *sn*-2 position (PC(16:0|Y)) (Extended Data Fig. 1b). We found that the transport behaviour of the fluorescent NBD PC deviated substantially from the bifunctional lipid

probe (Extended Data Fig. 1c–e). This implies a trade off between the improved ultrastructure of cryo-preserved samples using fluorescent lipid analogues and much more faithful recapitulation of native lipid behaviour by minimally modified lipid probes. We thus opted for a room temperature Lipid-CLEM approach (Fig. 1a,b) using bifunctional lipids based on our recently reported workflow for fluorescence imaging of lipids[21].

We aimed to maximize temporal resolution, preserve the membrane ultrastructure and improve lipid signal correlation precision. Thus, we decided to introduce the fluorescent label directly on the section as a final step of the sample processing routine and omitted the initial chemical permeabilization steps. Bifunctional lipid probes were loaded into the outer leaflet of the plasma membrane of living cells using an α-methyl-cyclodextrin-mediated lipid exchange reaction between donor liposomes and the plasma membrane[21,37] (Supplementary Fig. 1). After a 4 min chase time, lipid probes were photo-crosslinked to generate covalent lipid–protein conjugates, followed by immediate cryo-fixation using high-pressure freezing (Fig. 1b). For CLEM experiments, cells were cultured on carbon-coated sapphire disks. We found that the carbon coat reduced photoactivation efficiency by absorption of UV light (Supplementary Fig. 2). To improve lipid crosslinking, we developed a 365 nm (±10 nm) UV light-emitting diode (LED) device, which was placed directly above the sample in the aqueous medium. A spacer ring between the sample and the LED ensured homogeneous illumination. With this set up, a 3 s irradiation pulse was sufficient to achieve signal intensities comparable to irradiation through uncoated coverslips (Supplementary Fig. 2), in line with the results of an optimization of crosslinking conditions we previously performed[38]. Crosslinked and frozen samples were subjected to freeze substitution to dehydrate and embed samples into resin. This procedure also removed residual bifunctional lipid probes not crosslinked to proteins (Extended Data Fig. 2), which greatly reduced unspecific fluorescence lipid signal. Resin-embedded samples were sectioned, and individual sections were stained by on-section copper-catalysed click chemistry ('click chemistry' in the following)[39,40]. Sections were stained twice since we observed an increase in signal-to-noise ratio after repeated click labelling (Supplementary Fig. 3). Sections were then labelled with multicolour fiducials (TetraSpecs) to facilitate high-precision correlation[29,41]. Sections were imaged by four-colour widefield imaging and transmission electron microscopy (TEM) or tomography.

To ensure that our approach faithfully captures nanoscale lipid localization, we conducted a series of control experiments. We previously showed that (1) almost no probe metabolism occurs during the 4-min loading pulse and (2) that the native lipidome is not remodelled, indicating that the fluorescence signal is specific for the initially loaded lipid species and that no membrane stress responses are triggered[21]. Here we also monitored membrane remodelling and cellular signalling. We measured the number of clathrin-coated pits and quantified Tf uptake as a proxy for bulk endocytosis using a panel of lipids (bifunctional sphingomyelin (SM(Y)), PC(16:0/Y), PC(Y/20:4), PC(18:1/Y), PC(20:0/Y) and PC(18:0/Y)). Tf uptake and the number of clathrin-coated pits were unchanged, whereas a small reduction in the number of caveolae (8 ± 2%) was observed in SM(Y)-loaded cells (Extended Data Fig. 3). We furthermore monitored three signalling readouts: diacylglycerol (DAG) signalling (via a DAG biosensor)[9], phosphatidic acid (PA) signalling (via a PA biosensor)[37] and calcium signalling (using Fluo4AM) after lipid loading with a phosphatidylethanolamine PE(18:1/Y), PC(18:1/Y) and SM(Y). The responses were compared with two positive controls: Ionomycin, which raises intracellular calcium levels, and phosphatidic acid (PA(18:1/Y)), which recruits PA and DAG (after PA dephosphorylation) biosensors to the plasma membrane. We found that PE(18:1/Y), PC(18:1/Y) and SM(Y) did not trigger any responses, whereas PA(18:1/Y) and ionomycin induced robust transients (Extended Data Fig. 4). Taken together, bifunctional lipids that mimic structural membrane lipids are 'innocent' tracer molecules

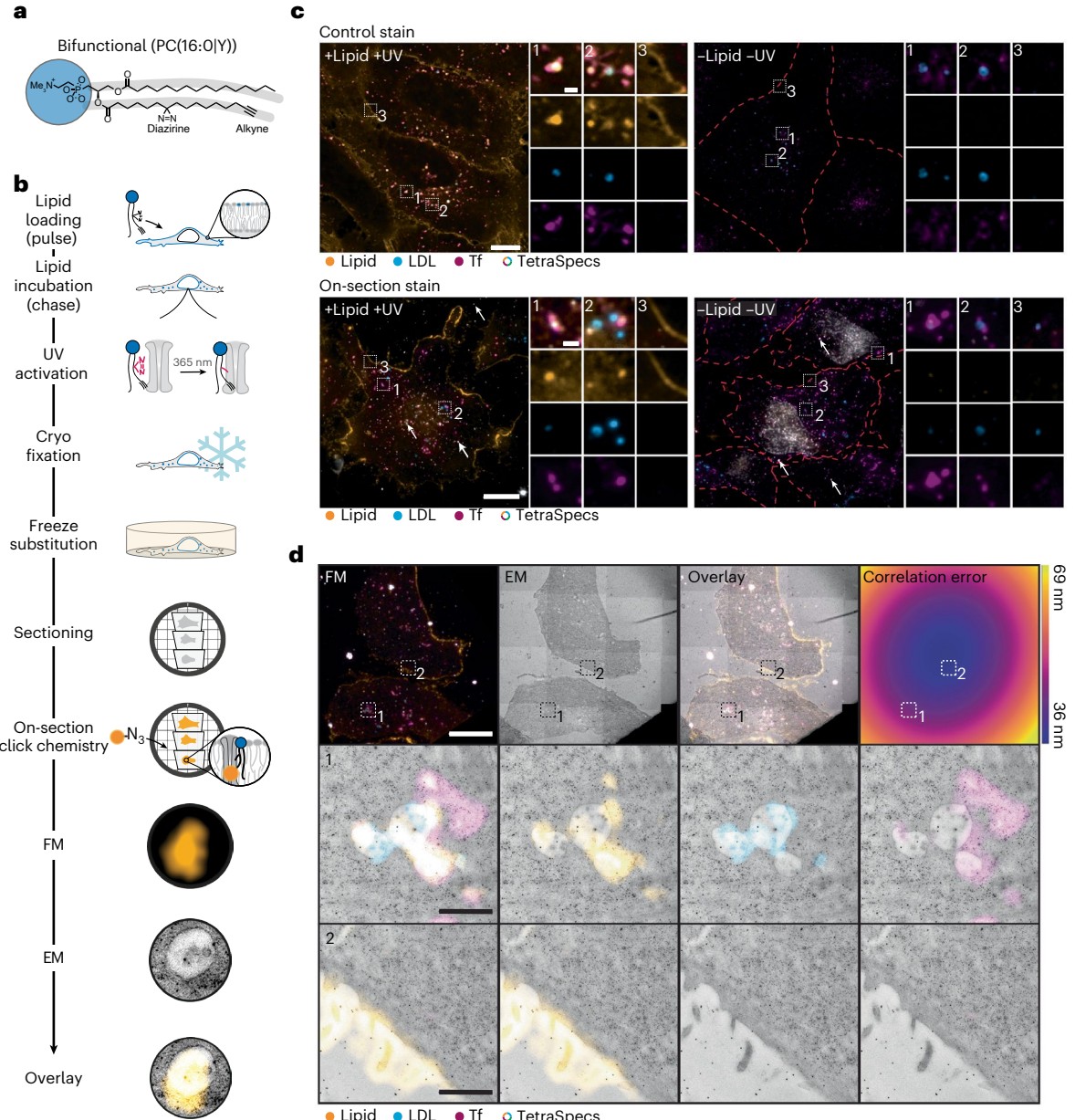

**Fig. 1 | On-section copper-catalysed click reaction for visualizing lipids using CLEM. a**, The chemical structure of the bifunctional lipid probe PC(16:0|Y). **b**, The Lipid-CLEM workflow: cells are loaded with bifunctional lipid probes, probes are UV crosslinked and samples are high-pressure frozen. Lipids are fluorescently labelled directly on sections after resin embedding via freeze substitution. Samples are imaged by FM and EM. The images are overlaid by correlation. **c**, Cells were co-labelled with fluorescent LDL (blue) and Tf (magenta). The lipid signal is shown in orange. Control stain: cells with (+lipid +UV) and without (−lipid −UV) bifunctional PC(16:0|Y) were chemically fixed, permeabilized, labelled by click chemistry and imaged using confocal FM. On-section stain: thin sections (100 nm) of HM20 samples treated with (+lipid +UV) and without

(−lipid −UV) bifunctional PC(16:0|Y) were labelled according to the workflow. The dataset includes three independent biological replicates that yielded similar results. Multicoloured TetraSpecs are indicated by arrows. The dotted red lines represent cell borders if not visible otherwise. Scale bars, 10 μm (overview images) and 1 μm (ROIs). **d**, CLEM of thin HM20 sections (100 nm) shows localization of the lipid at the plasma membrane and endosomes. The fluorescent overview image (FM), medium magnified TEM image (EM), overlay and correlation error map of 100-nm-thick sections are shown left to right. Representative ROIs for an endosome (1) and the plasma membrane (2) are shown, acquired at higher magnification by TEM. Scale bar, 10 μm (overview images) and 1 μm (ROIs). The HM20 dataset includes one replicate.

for the investigation of intracellular lipid localization, transport and metabolism when used at the concentration reported here.

To establish the specificity of the click chemistry-derived fluorescent signal, we compared signal localization between samples of fixed cells grown on glass-bottom 96-well plates with sectioned resin-embedded samples. In both cases, U2OS cells were loaded with PC(16:0/Y) (Fig. 1a). After a 4-min lipid loading pulse, the probe was primarily localized at the plasma membrane and in endosomes

for both sample types, confirming the specificity of the on-section click labelling (Fig. 1c,d). We validated the endosomal localization by co-staining with fluorescently labelled Tf and LDL. The use of Tf and LDL as suitable endosomal markers was demonstrated by co-staining with Rab5 (early endosomes), Rab7 (late endosomes) and Rab11 (recycling endosomes, antibody validated by the co-expression of Rab11–GFP) (Supplementary Fig. 4 and Extended Data Fig. 5). The lipid localization was further validated by assessing the organelle ultrastructure using

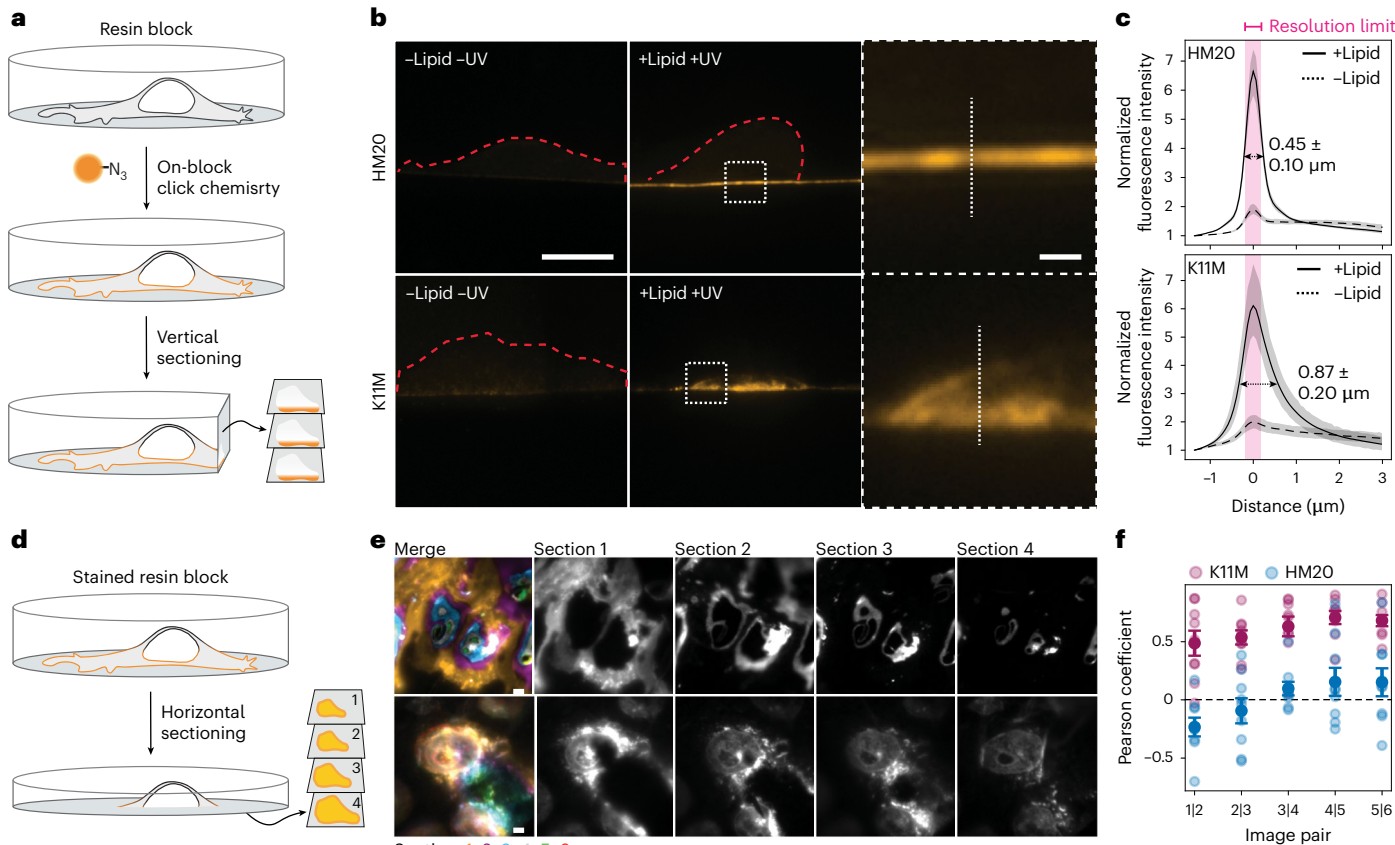

**Fig. 2 | Comparison of dye diffusibility into HM20 and K11M resins. a**, The workflow for testing the penetration depth of the click reaction mixture into resins. Whole resin blocks were stained and subsequently sectioned vertically to assess penetration depth. **b**, Representative images of stained and vertical sectioned blocks of HM20 (upper) and K11M (lower), of samples loaded with (+lipid +UV) and without (−lipid −UV) PC(16:0|Y). Scale bars, 10 μm (overview images) and 1 μm (ROIs). The dotted red lines represent cell borders if not visible otherwise. **c**, The average line profile (indicated in **b**) of the fluorescence signal of vertical sections of HM20 and K11M with (black line) and without (dotted line) PC(16:0|Y). The error of the average line profile is shaded in grey and given as the confidence interval of 95%. The full width at half maximum (± s.d.) for the lipid-loaded samples is indicated. The purple bar indicates the mean resolution limit of the optical system as determined with 100-nm beads. Two independent samples were analysed for each resin, $n$ (K11M sections) = 29 and $n$ (HM20 sections) = 29.

**d**, The workflow for testing the penetration depth of the click reaction mixture into resins. Whole resin blocks were stained and subsequently sectioned horizontally to assess the penetration depth. **e**, Representative images are shown of the first six horizontally cut sections merged and the first four sections separately of stained blocks for HM20 and K11M. Scale bar, 10 μm. The colours indicate the number of the section, with section 1 being the first section collected from the block. Section thickness, 100 nm. **f**, The Pearson coefficients between horizontal serial sections were determined. Coefficients were calculated pairwise for subsequent sections. Two independent samples were analysed for each resin and each analysis ($n$) comprises a ROI that contains at least one whole cell, $n$ (K11M ROI) = 10 and $n$ (HM20 ROI) = 9. The mean and s.e.m. are indicated as solid circles and error bars, respectively. Single data points are shown as transparent circles. High Pearson coefficients indicate an overlap of signal between sections, whereas low coefficients indicate mutually exclusive signals.

CLEM. The plasma membrane and endosomes were identified based on the fluorescent signal and imaged at high magnification by TEM (Fig. 1d). Taken together, we conclude that the Lipid-CLEM workflow (the detailed workflow has been previously published at protocols.io (ref. 42)) allows us to assess the localization of near-native lipid probes within organelle membranes at the ultrastructural level.

**Polar resins enable homogeneous labelling across lipid classes**
Measuring lipid densities at membrane nanodomains requires measuring domain surface area as well as the lipid amount within the domain. Three-dimensional (3D) surface areas are obtained from electron tomography data and relative lipid amounts from fluorescent signal intensities. Thus, to acquire accurate densities, lipid probes need to be evenly stained throughout the sections by on-section click chemistry. To determine the homogeneity of the fluorescent signal throughout the resin, we labelled whole HM20 resin blocks derived from U2OS cells loaded with PC(16:0|Y). Resin blocks were then sectioned vertically and the sections imaged by FM (Fig. 2a). HM20 samples showed signal confined to the edge of sections, with a full width at the half maximum

of 0.45 ± 0.10 μm (Fig. 2b,c). The resolution limit of the imaging set up was measured to 0.35 ± 0.03 μm using 100-nm fluorescent beads. The signal width of HM20 exceeds the resolution limit of the imaging set up. However, the section edge has a width of unknown dimension, which must be taken into consideration. Therefore, we additionally sectioned fully stained HM20 blocks horizontally into thin sections of 100 nm (Fig. 2d–f). The signal of the consecutive sections was found to be mutually exclusive, suggesting that the click labelling is limited to the section surface in HM20 samples.

We hypothesized that the polar reactants of the click chemistry reaction mixture might not penetrate the hydrophobic HM20 matrix and therefore tested the polar K11M resin for fluorescent labelling. In K11M, the signal derived from vertically cut sections showed a significantly higher (Student's $t$-test $P$ value of $1.82 \times 10^{-13}$) average full width at half maximum of 0.87 ± 0.20 μm with visible internal membranes (Fig. 2b,c). Next, fully stained K11M blocks were cut into horizontal sections and the fluorescent signal was shown to be overlapping in subsequent sections. Overall, this indicates that the click reagents penetrate into the K11M resin (Fig. 2d–f). The high penetration depth

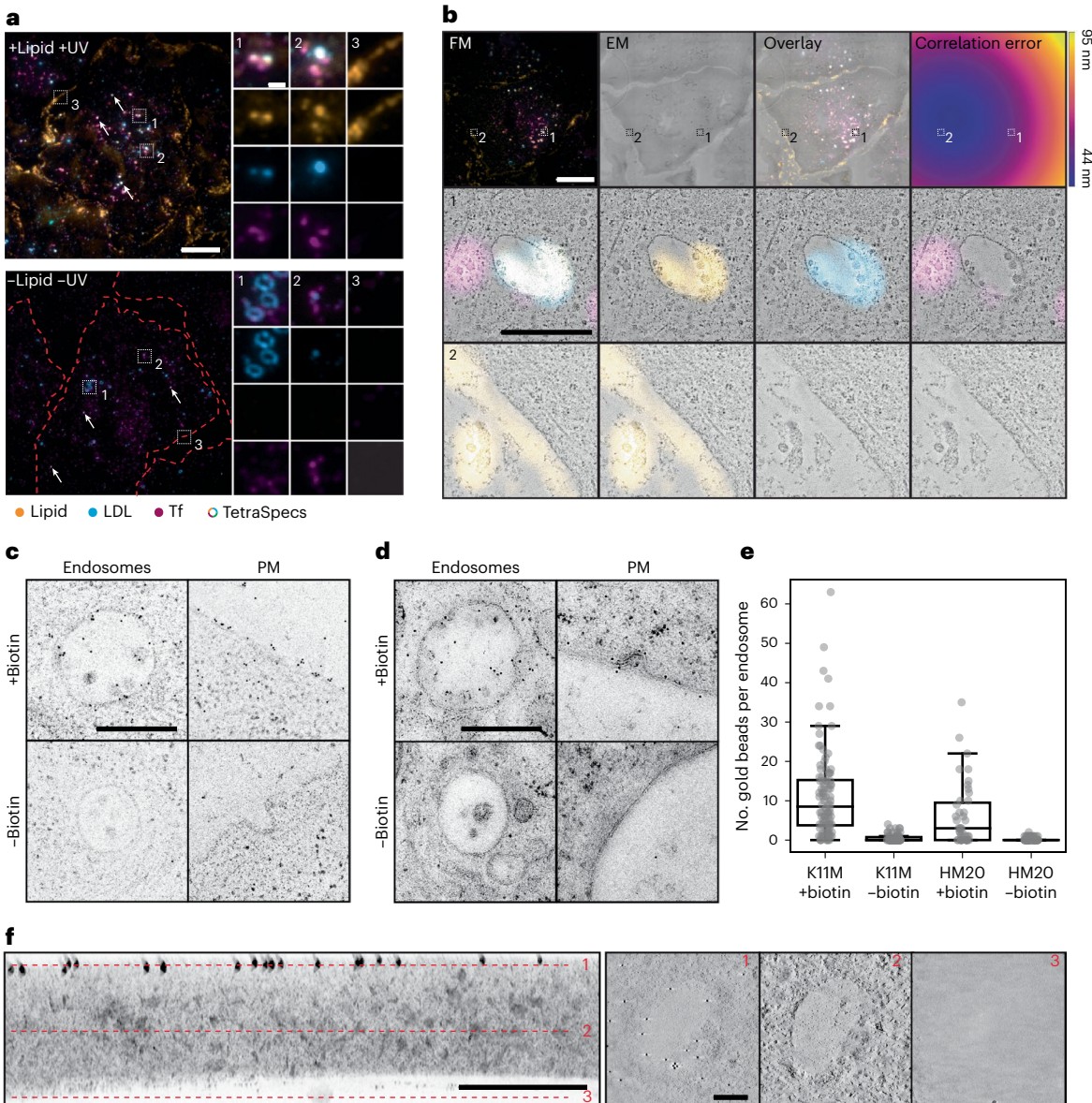

**Fig. 3 | Lipid signal is well preserved and specific in K11M resin. a**, U2OS wild-type cells were loaded for 4 min with PC(16:0|Y), UV crosslinked, high-pressure frozen, embedded in K11M by automated freeze substitution and sectioned into 100-nm-thick sections (+lipid +UV). Negative controls were not UV activated and not loaded with PC(16:0|Y) (−lipid −UV). All cells were co-labelled with fluorescent LDL (blue) and Tf (magenta). The lipid signal is shown in orange. White dots correspond to multicoloured TetraSpecs indicated by arrows. The dotted red lines represent cell borders if not visible otherwise. Scale bars, 10 μm (overview images) and 1 μm (ROIs). The dataset includes three independent biological replicates that yielded similar results. **b**, The fluorescent overview image (FM), medium magnified TEM image (EM), overlay and correlation error map of 500-nm-thick K11M sections are shown left to right in that order. A representative ROI for an endosome (1) and the plasma membrane (2) are shown, acquired at higher magnification by tomography. One image plane of the tomogram is shown. Scale bars, 10 μm (overview images) and 1 μm (ROIs).

Representative images of two biological replicates are shown that yielded similar results. **c,d**, SM(Y)-loaded samples were embedded in K11M (**c**) or HM20 (**d**) and labelled with gold beads. Exemplary ROIs of endosomes and the plasma membrane (PM) are shown for samples labelled with and without biotin, followed by immunolabelling against biotin and subsequent immunolabeling with 10-nm gold beads. Scale bar, 500 nm. **e**, The quantification of gold beads per endosome (*n* (endosomes) = 96, 82, 42 and 68 in the box plots from left to right) for two independent samples per condition. The box spans from the first quartile (Q1, 25th percentile) to the third quartile (Q3, 75th percentile), with the median (50th percentile) shown as the central line. Tukey-style whiskers extend to a maximum of 1.5× the interquartile range beyond the box, indicating the range of non-outlier data. Single data points are shown as grey dots. **f**, A sideview of a tomogram of a 500-nm K11M section labelled with gold beads (10 nm) against SM(Y) is shown. Three exemplary views in plane (*xy*) are shown, as indicated by the dotted red lines. Scale bars, 200 nm.

observed for K11M ensures the faithful assignment of lipid enrichment to membranes even in thicker sections used for tomography.

To ensure the specificity of the lipid signal in the K11M resin we next assessed the PC(16:0|Y) signal using CLEM. We observed the characteristic plasma membrane/endosome localization of the lipid signal, indicating a specific staining (Figs. 1c and 3a,b). However, K11M samples exhibited some non-specific signal in nucleoli, a known click chemistry

artefact[43], which has to be taken into account. One additional consideration to keep in mind is that membrane preservation in K11M is not as strong as in HM20 (refs. 44,45) (Extended Data Fig. 6).

To assess whether Lipid-CLEM offers unique advantages over more established labelling techniques, we adapted our workflow for imaging single lipid species at high resolution to allow for nanogold labelling, an approach that has previously been used to visualize

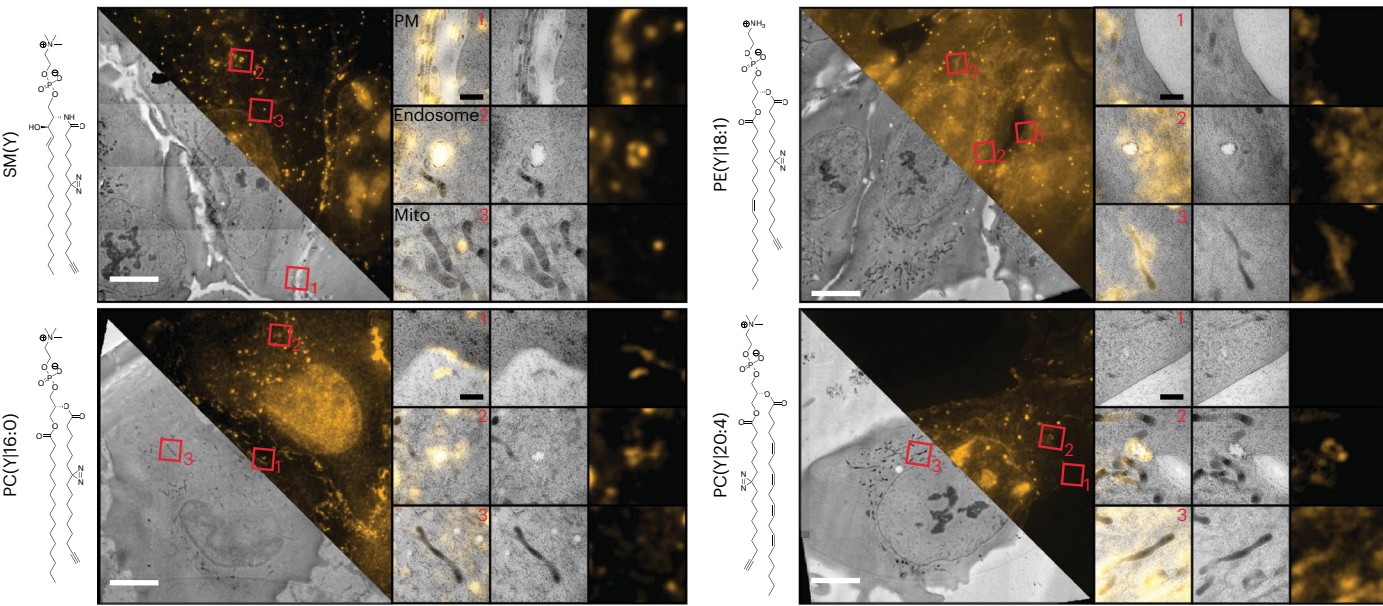

EM FM lipid

**Fig. 4 | Lipid-CLEM of various lipid classes and species.** Cells were treated with various lipid classes and species, including SM(Y), bifunctional phosphatidylcholine of different fatty acid composition (PC(Y|16:0) and PC(Y|20:4)) and phosphatidylethanolamine (PE(18:1|Y)). All chemical structures are indicated on the left. Representative fluorescent images of the lipid signal are shown for all lipids in orange, highlighting the overall differential lipid localization across whole cells for the different lipids. The respective correlated EM image is shown to the right. Correlated ROIs are shown for the plasma membrane (PM), mitochondria (mito) and endosomes. Fluorescence intensities for images of the same lipid are scaled equally. Scale bar, 10 μm or 1 μm. Correlation error maps of the images shown here can be found in Extended Data Fig. 7. All datasets include two independent biological replicates per lipid species that yielded similar results.

entire lipid classes by metabolic labelling[46]. For immunogold labelling, individual sections containing crosslinked lipids were labelled with biotin-picolyl-azide, followed by immunolabelling against biotin and subsequent labelling with 10-nm Protein A gold beads. We found that immunolabelling was highly specific in both K11M and HM20, with barely any background detected (Fig. 3c–e). However, labelling was relatively sparse, in line with previous results[46]. On average, we found $11 \pm 1$ (mean ± s.e.m.) gold beads per endosome in K11M samples and $6 \pm 1$ (mean ± s.e.m.) gold beads per endosome in HM20 samples. Tomograms of K11M samples confirmed that gold beads were exclusively localized at the section surface, in line with the expectation for nanogold labelling[47,48] (Fig. 3f).

We therefore propose two possible applications for on-section Lipid-CLEM depending on the choice of resin used. Apolar resins such as HM20 are ideally suited to assign the localization of individual lipid species to well-preserved membrane ultrastructures at the surface of the resin. Similarly, nanogold labelling can be used as an alternative technique if there is a requirement to detect a molecular label directly in the electron microscopy (EM) image, but this will probably come at a lower labelling density owing to the multistep labelling procedure and steric bulk of the reagents introducing additional experimental constraints. Direct measurements of lipid densities on 3D membrane structures, on the other hand, are feasible by combining the Lipid-CLEM approach with polar resins such as K11M. To demonstrate that the Lipid-CLEM workflow is compatible with various lipid classes and enables analyses of their organelle distribution, we acquired Lipid-CLEM datasets for different lipids: PC(Y/16:0), SM(Y), PC(Y/20:4) and PE(18:1/Y). We compared lipid localization in the plasma membrane, in endosomes and in mitochondria. PC(Y/16:0) and SM(Y) showed a pronounced plasma membrane and endosomal localization with no noticeable localization in mitochondria. PC(Y/20:4) plasma membrane localization was found to be lower and some signal was present in mitochondria, while we found a pronounced localization of PE(18:1/Y) in mitochondria, observations that are in line with our previous data[21] (Fig. 4 and Extended Data Fig. 7).

## SM partitions into endosomal subcompartments

We next analysed lipid densities in the membrane compartments of the early endosome. Its complex membrane architecture features well-characterized protein domains, which are critical for protein sorting during membrane trafficking[49,50] and may also play a role in lipid sorting[35,36,51]. The partitioning of endosomal proteins into these long-lived compartments has been extensively studied and allows for straightforward classification. Here, we considered three endosomal compartments: the globular boundary membrane (BM), ILVs and RTs. We chose to analyse the distribution of SM in these endosomal compartments as an example of a typical plasma membrane and late secretory pathway lipid. To compare with bulk lipid density, we performed metabolic labelling with bifunctional palmitic acid, which is widely incorporated into the cellular lipidome. All quantifications of lipid densities were based on high-resolution tomograms of the entire section depth to ensure that there would be no difference in dimensionality ('sample thickness') between FM and EM data.

Early endosomes were identified by the presence of both LDL and Tf fluorescence signals and high-resolution tomograms were acquired for the corresponding regions. All tomograms were reconstructed and the set of endosomes that featured both ILVs and RTs was used for further analysis. The 3D membrane models were generated manually to determine the surface area of each compartment (Supplementary Fig. 5 and Supplementary Viedos 1 and 2). The outlines of the 3D domains were $z$ projected into two-dimensional (2D) images and blurred to generate a mask (Fig. 5a) to assign the fluorescent signals of SM, LDL and Tf to the compartments.

Fluorescent intensities were determined for each mask and normalized to the respective membrane domain surface area, resulting in fluorescence intensity densities. In regions where masks overlapped, partial pixel values were assigned to the respective masks based on probability density functions generated from the non-overlapping parts of the masks when possible (see the Methods for details). Finally, the fluorescent density of each membrane domain was divided by

the fluorescent density of the whole endosome to determine relative enrichments (Fig. 5b and Supplementary Figs. 6–9). LDL and Tf were used as internal standards for the analysis since they are enriched in ILVs and RTs, respectively[32–34]. We observed the expected pattern of strong LDL enrichment in ILVs with a mean of 5.73 ± 0.99 relative density and strong Tf enrichment in RTs with a mean of 2.07 ± 0.23 fold density. These values can be considered as the practical upper boundaries of enrichment.

For SM, we observed a slight depletion over the whole endosomal average in the BM with a mean relative density of 0.88 ± 0.08, enrichment in ILVs with a 3.64 ± 0.50 relative density and depletion in RTs with a 0.49 ± 0.10 relative density. These results are indicative of differential localization and thus sorting of SM into different membrane domains in the early endosome (Fig. 5b,d and Supplementary Figs. 6 and 7). The respective values obtained for the metabolic labelling experiment with bifunctional palmitic acid were 0.73 ± 0.03 for the BM, 2.52 ± 0.48 for ILVs and 1.23 ± 0.19 for RTs (Fig. 5c,d and Supplementary Figs. 8 and 9). Notably, the palmitic acid signal in endosomes was generally lower compared with other organelles, such as mitochondria and the endoplasmic reticulum, suggesting preferential incorporation into other organelle membranes. Taken together, lipid partitioning in endosomes was significantly different between a lipid population derived from metabolic labelling in comparison with a single SM species and thus indicative of lipid sorting.

To exclude resin-dependent artefacts, we performed a similar experiment using HM20 resin. Here, we acquired 2D EM images of thin (100 nm) sections to maximize the number of surface-exposed lipid residues available for staining and calculated 1D densities from 2D masks (Extended Data Fig. 8). These values are not directly comparable to the 2D densities derived from K11M tomograms and effect sizes are generally smaller in thin HM20 sections. Despite these limitations, the general trend of SM(Y) depletion in RTs and enrichment in ILVs was maintained.

To confirm whether the differential localization of SM(Y) and Tf in early endosomes is due to sorting within the endosomal compartment rather than sorting into different populations of endocytic pits, we simultaneously loaded SM(Y) and Tf and quantified incorporation into clathrin-coated pits. We find that SM(Y) is present in all Tf-positive clathrin-coated pits (Extended Data Fig. 9), in line with our recent analysis of lipid partitioning into clathrin-coated pits[38]. This indicates that SM and Tf were taken up together by endocytosis but separated in early endosomal compartments, with Tf being enriched in RT and SM in ILVs. Taken together, our data imply that SM is sorted into distinct membrane subcompartments within the early endosome.

## Discussion

Here, we report a workflow for CLEM to image lipids in cells (Lipid-CLEM) based on minimally modified lipid probes and on-section click chemistry labelling. Two distinct variants were developed for fluorescence labelling of (1) section surfaces and (2) entire section volumes. Surface labelling in an apolar resin (HM20) is optimized for the preservation of ultrastructure and to precisely assign fluorescent signals in the z dimension. Full-volume labelling in a polar resin (K11M) allows for determining the densities of individual lipid species in complex membrane architectures in three dimensions.

Lipid-CLEM is best suited for studying lipid localization at ultrastructural resolution, which differs from our previously published workflow that utilized bifunctional lipids for quantifying interorganelle lipid transport by FM[21]. Both approaches have well-defined use cases. Lipid imaging based on regular confocal FM offers high throughput and allows to address questions regarding the intracellular distribution and transport of lipids between organelles. Lipid-CLEM is much lower throughput, but offers much higher resolution—in fact, this technique constitutes the first generalizable workflow to study lipid localization within the 3D ultrastructure of biological membranes. Thus, lipid imaging by confocal FM is suitable, for example, for studying lipid transport between organelles on a micrometre scale, whereas Lipid-CLEM is enables analyses of suborganellar lipid partitioning and sorting at the nanoscale.

Our Lipid-CLEM approach for imaging individual lipid species in cellular membrane nanodomains complements existing methods for studying lipids at high resolution: halogenated lipids have been visualized in vitro by single-particle EM[52]. Immunogold labelling on sections of cells[53,54] allows for visualization of lipid class distributions by EM if suitable antibodies are available. Previous CLEM approaches based on minimally modified lipid probes have localized lipids or whole lipid classes qualitatively; however, they were typically not ideal for the preservation of membrane ultrastructure or are limited to studying the outer leaflet of the plasma membrane[14,27,28].

We used Lipid-CLEM to address the question of lipid sorting in the early endosome and found that SM is enriched in ILVs and significantly depleted in RTs. The observed depletion in RTs cannot be explained by partitioning modes based on lipid asymmetry or sorting by curvature due to lipid shape[55,56], as these mechanisms can only account for much more moderate changes in partitioning. Since RTs and the endosomal BM form one continuous membrane structure, the significantly lower levels of SM in one of the compartments (RTs) suggests the existence of a diffusion barrier. The approximately threefold enrichment of SM in ILVs is most plausibly explained by either non-vesicular lipid transport in the lumen of the endosome or segregated sites of coordinated vesicle fusion and fission, which directly channel incoming lipid material to ILVs. As a similar effect is observed during broad lipidome labelling, the latter hypothesis appears more likely. These findings allow us to draw the following conclusion: protein and lipid transport routes diverge at the early endosome as SM and Tf arrive simultaneously in clathrin-coated vesicles[57,58], but partition differentially into endosomal compartments.

**Fig. 5 | SM partitions differently into early endosome subcompartments.**
**a**, CLEM data analysis: images were acquired by FM and EM at medium magnification. After correlation, the ROI was chosen based on the fluorescent signal of LDL (blue) and Tf (magenta). The lipid signal is shown in orange. At the ROI, a tomogram was acquired at high magnification. Tomograms were further analysed if the endosomes contained BM (orange arrow), ILVs (blue arrow) and a RT (magenta arrow). Based on the tomogram, 3D models were derived. The z projection of the 3D model outlines is shown. Membrane domains in the model are classified into BM (orange), ILVs (blue) and RTs (magenta) and the respective classes were blurred to create class-specific masks. Scale bars, 10 μm (overview images) and 500 nm (ROIs). **b**, Three representative endosomes are shown with the fluorescent overlay, classified 3D model and model overlays with the fluorescent signal (grey) of SM, Tf or LDL from left to right. Scale bars, 500 nm. Right: the relative densities of all fluorescent signals (SM, Tf and LDL) over all membrane classes: RT (magenta), ILVs (blue) and the BM (orange).

**c**, A representative endosome of a cell treated with the bifunctional fatty acid and the corresponding quantification of the fluorescent signals. Scale bar, 500 nm. **d**, The relative fluorescent densities are plotted for all analysed endosomes for SM and the metabolically labelled samples (FA). The mean densities are shown for the different domains: RT (magenta), BM (orange) and ILV (blue) and for the respective fluorescent signals of lipid, Tf and LDL. Maximum possible fold enrichments are indicated by the dotted horizontal lines for the BM (orange), RT (magenta) and ILVs (blue). No enrichment is indicated by the grey line at 1.0. Error bars are shown as the 95% confidence interval of the mean values. Single data points are shown in grey. P values are indicated and were determined using a two-sided Student's t-test. The exact P values are 0.00192, 0.82109, 0.14332, 0.12113, 0.53060, 0.48663, 0.10007, 0.64835 and 0.08744 from left to right in **d**. Two independent biological replicates were performed both for SM (n (endosomes) = 13) and the fatty acid (n (endosomes) = 12). **P < 0.01, *P < 0.05, non-significant (n.s.) P > 0.05.

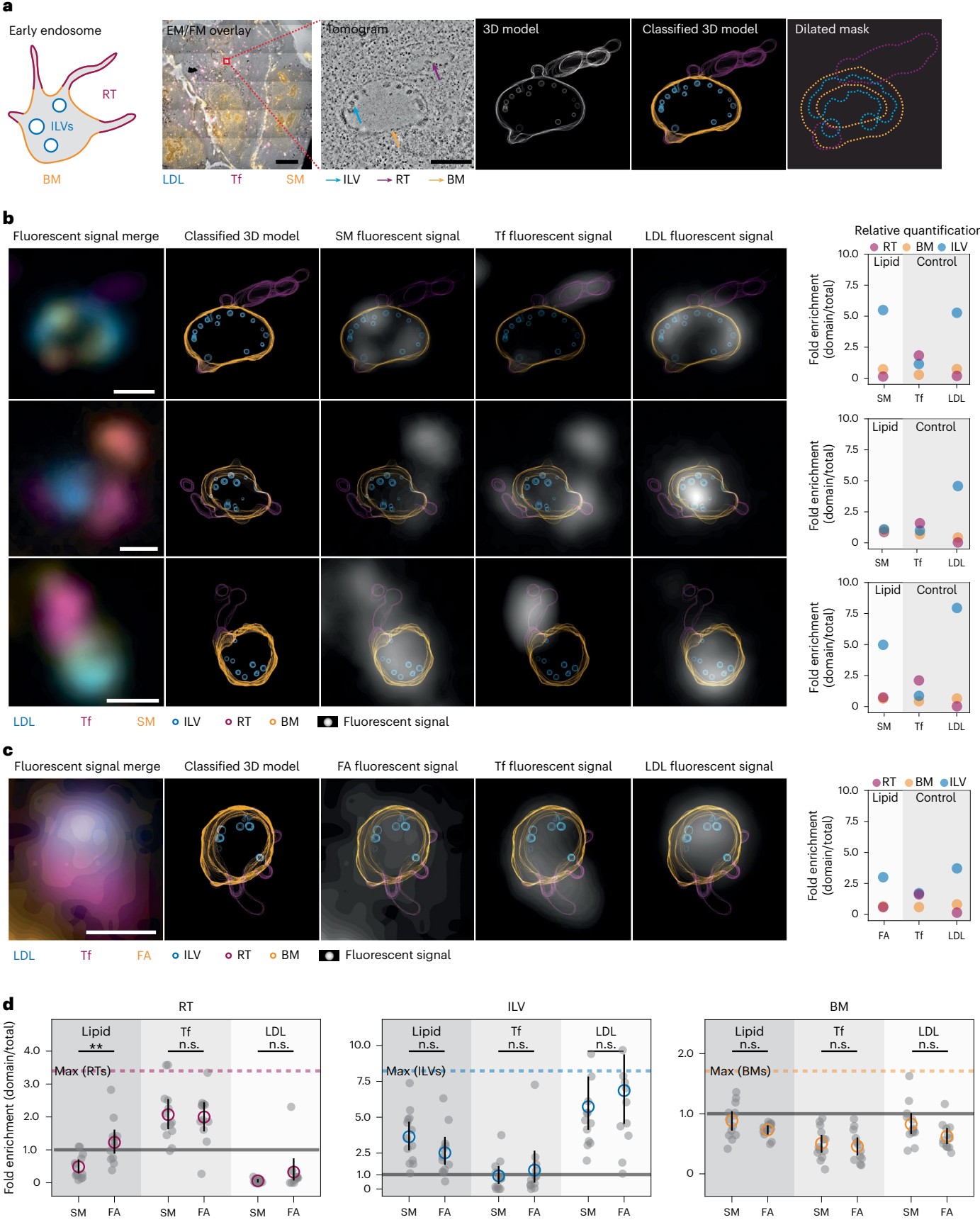

Taken together, our methodology allows us to measure lipid densities in nanoscale compartments of biological membranes. By directly observing lipid partitioning into subcompartments of the early endosome, we identify a checkpoint at which protein and lipid transport diverge during retrograde membrane trafficking. More generally, our approach will have major benefits for studying membrane nanoscale structures of complex architecture where a prominent role of lipids has long been hypothesized but was difficult to substantiate owing to lacking methodology. Such domains include, for instance, signalling clusters at the plasma membrane, endoplasmic reticulum exit sites, nuclear pore complexes and cristae junctions in mitochondria. We anticipate that our Lipid-CLEM approach represents a major step forward towards defining membrane nanoscale architecture for both lipids and proteins. Data of this type are required for developing a broadly applicable consensus model of biological membranes.

## Online content

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

## Methods

This research complies with all relevant ethical regulations.

### Statistics and reproducibility

**Samples sizes and randomization statement.** No statistical methods were used to predetermine sample sizes but our sample sizes are similar to those reported in previous publications[21,29]. Regions of interest (ROIs) to acquire images were selected automatically by the software of the microscope (CellSens software) for light microscopy. If necessary, with low cell density, regions for imaging were selected with cells in the field of view. For CLEM, endosomes were selected based on the co-localization of the Tf and LDL markers to identify early endosomes as explained in the main text.

**Blinding statement.** Data collection and analysis were not performed blind to the conditions of the experiments.

**Data exclusion statement.** Images were excluded if the acquisition process failed by either being out of focus or if unwanted cellular debris or clamps precluded image analysis. For tomograms and TEM, images were excluded if the tomogram reconstruction failed, if endosomes did not contain either RTs and/or ILVs or if cells were visibly not intact.

### Sample preparation

**Cell culture.** U2OS wild-type cells (DSMZ, ACC785) were cultured in McCoy's 5A (1×) + GlutaMAX (Gibco, 36600-021) medium supplemented with 10% fetal bovine serum (FBS; Gibco, 10270-106) and 100 U ml$^{-1}$ penicillin–streptomycin (Gibco, 15140-122) at 37 °C with 5% $CO_2$. Cells were passaged every 2–3 days using 0.05% Trypsin (Gibco, 25300-054) up to a passage of 20. Cell seeding on carbon-coated disks is described below. Cells were seeded 24 h before starting the experiment into glass-bottom 96-well plates (Greiner, 655891) at a density of 9,000 cells per well.

**U2OS Sec61 beta cell line.** For live-cell imaging with endoplasmic reticulum co-stain, a Sec61 beta Halo U2OS cell line was made. The Sec61 beta Halo plasmid (Addgene, 123285) was randomly integrated into the genome of U2OS wild-type cells (DSMZ, ACC785). Then 500 ng of the plasmid were transfected to 250,000 cells by neon electroporation at 1,230 V, 10 ms for 4 pulses. Cells were expanded for 5 days and stained using the Halo tag TMR (Promega, G8251) according to the manufacturer's recommendations. The cells were sorted using FACS. TMR-stained wild-type cells and non-stained transfected cells were used as negative controls.

**Plasmid design.** Synthetic DNA fragments encoding the yeast SNARE protein domain Spo20p[59], mEGFP and a nuclear export sequence derived from the protein kinase A inhibitor α (NES-PKI) were ordered from Twist Bioscience. Fragments were amplified by PCR using Twist Bioscience's forward primer 5′-CAATCCGCCCTCACTACAACCG-3′ and reverse primer 5′-TCCCTCATCGACGCCAGAGTAG-3′. The mEGFP sequence was inserted in the C-terminal of the two PABDs using BamHI (New England Biolabs, R3136S) and AscI (New England Biolabs, R0558L). The 2xPABD-mEGFP sequence was transferred into a vector for mammalian expression using NotI (New England Biolabs, R3189L) and AscI. The NES-PKI was inserted in the N-terminal of the two PABDs using NheI (New England Biolabs, R3131S) and NotI, mimicking the biosensor design from ref. 60.

**Transfection and live-cell imaging.** For transfection with a Rab11–GFP construct (a kind gift from the Zerial laboratory[61]), U2OS wild-type cells (DSMZ, ACC785) were seeded into six-well plates at a density of 100,000 per well. After 16 h, cells were treated with a Xtremegene9 transfection mix (Roche, XTG9-RO) according to the manufacturer's instructions (3,000 ng plasmid per well). The transfection mix was removed after 5 h

and the cells split into a 96-well plate at a density of 9,000 per well. Cells were allowed to recover for 24 h. For transfection with diacylglycerol and phosphatidic acid sensor, U2OS wild-type cells were seeded into eight-well Ibidi dishes (Ibidi, 80826) at a density of 40,000 per well the prior day. The following day, cells were transfected with a diacylglycerol sensor construct (C1-eGFP-NES) as reported previously[11] or a phosphatidic acid sensor using Xtremegene9 (500 ng or 130 ng plasmid per well, respectively). The transfection mix was removed after 5 h and cells were imaged after an additional 24 h. The samples were imaged in FBS-free medium. For calcium imaging, U2OS wild-type cells were seeded into Ibidi 8-well dishes at a density of 80,000 per well and grown overnight. The next day, samples were stained and imaged in imaging buffer (HEPES (20 mM), NaCl (115 mM), $CaCl_2$ (1.2 mM), $MgCl_2$ (1.2 mM), $K_2HPO_4$ (1.2 mM) and glucose (20 mM), pH 7.4). For calcium detection, samples were treated with the Fluo-4 Calcium Imaging Kit-1 (Thermo Fisher, F10489) according to the manufacturer's instructions. For calcium influx, these cells were also treated with 100 nM Ionomycene (MP biomedicals, 159611) in imaging buffer. All samples were imaged for 40 s, lipid mixtures or ionomycene were added and samples imaged for an additional 300 s.

**Sapphire preparation and cell seeding.** First, 3 mm × 0.05 mm Sapphire disks (Wohlwend GmbH, 405 or Leica, 16702766) were coated with a total of approximately 15 nm carbon film using the safematic CCU-010 at 10 A current at $5.0^{-5}$ mbar pressure. Coated sapphires were baked overnight at 180 °C and used within 1 week. Immediately before cell seeding, sapphires were glow discharged at 15 mA negative, under 0.24 mbar for 90 s using a Pelco easiGlow (model 91000). Sapphires were placed in 35-mm glass-bottom dishes (Cellvis, D35-14-1.5-N) and pre-incubated with cell culture medium for 30 min at 37 °C. In total, 240,000 U2OS cells per dish were seeded. Cells were grown for 12–18 h before high-pressure freezing.

**LDL labelling.** Human LDL (Invitrogen, L3486) was fluorescently labelled with Alexa Fluor 488 carboxylic acid, succinimidyl ester (Invitrogen, A200000) in sodium bicarbonate buffer. Approximately 1 mg LDL was labelled covalently with 10 mg ml$^{-1}$ (~200 nmol of reactive species) staining solution according to the manufacturer's recommendations for 1 h at room temperature. The freshly labelled mix was dialysed against phosphate-buffered saline (PBS) using a Slide-A-Lyzer Mini Dialysis (Thermo Scientific, 88402) at 4 °C. The LDL solution in PBS was supplemented with 10% w/v sucrose and stored at −80 °C. The final LDL-AF488 concentrations were determined by a bicinchoninic acid assay.

**LDL and Tf loading.** Before loading, cells were washed with serum-free medium. Labelled human LDL (see above) was dissolved in a serum-free medium at 5 µg ml$^{-1}$. The cells were incubated with freshly prepared LDL solution for 15 min and chased for another 15 min in serum-free medium without LDL at 37 °C. Labelled Tf-AF647 (Invitrogen, T23366) was dissolved in serum-free medium at 20 µg ml$^{-1}$ and cells were incubated with freshly prepared Tf solution for 4 min at 37 °C.

**Liposome preparation and lipid loading.** The synthesis of all bifunctional lipids used in this study was previously reported[21]. A suspension of 1.5 mM bifunctional lipid, 0.75 mM 1-palmitoyl-2-oleoyl-glycerol-3-phosphocholine (Avanti, 26853-31-6) and 0.75 mM cholesterol (Sigma, CAS no. 57-88-5, C8667-5G) in PBS was prepared. For NBD lipids (Avanti, 810131P, 810133P) a suspension of 1.5 mM NBD lipids, 0.75 mM 1-palmitoyl-2-oleolyl-glycerol-3-phosphocholine and 0.75 mM cholesterol was prepared. Liposomes were made from this suspension using an Avanti Mini-Extruder with 0.1-µm polycarbonate membranes (Avanti, 610005-1Ea), by extruding for a minimum of 21 times. Liposomes and α-methyl-cyclodextrin (Bio-Reagent, CDexA-076/BR, CAS no. 699020-02-5) were dissolved in serum-free medium to a

final concentration of 0.5 mM and 4 mM, respectively. This mix was incubated for at least 30 min at 37 °C before loading on cells. Before lipid loading, cells were gently washed three times with serum-free medium. Lipids were loaded for 4 min at 37 °C to cells unless stated otherwise. Cells were then washed with serum-free medium, followed immediately by UV activation. When loading bifunctional fatty acids for metabolic labelling, the fatty acid was dissolved in 100% ethanol and added directly in full medium to 5 μM for 17 h incubation duration.

**UV irradiation.** Samples were UV activated for 3 s using a custom built LED device (Violumas, VC2X2C45L9-365) after lipid loading. A hand-held single LED was used for samples grown on carbon-coated disks and a custom multi-LED was used for samples in the 96-well plate format.

**Chemical fixation, permeabilization and staining.** Samples were fixed immediately after UV irradiation. For all control experiments, unless stated otherwise, cells were fixed using 4% PFA (CAS no. 30525-89-4, TCI) in PBS (pH 7.3) for 15–20 min at room temperature. For CLEM, cells were fixed by high-pressure freezing (see below). For conventional lipid imaging, fixed cells were washed three times with 100 mM glycine in PBS and permeabilized using 0.1% Triton in PBS for 30 min at room temperature. Fixed and permeabilized samples were blocked at room temperature for 1 h in a blocking buffer consisting of PBS supplemented with 2% bovine serum albumin (BSA; Sigma, A7030-10G). Primary antibodies were incubated in blocking buffer for either 1 h at room temperature or overnight at 4 °C. Samples were washed three times for 15 min in blocking buffer at room temperature and incubated with the respective secondary antibody for 1 h at room temperature in blocking buffer. Samples were washed three times for 15 min in blocking buffer at room temperature and kept in PBS supplemented with 0.01% sodium azide (Sigma-Aldrich, 71290-100 G). If samples were also treated with a copper-catalysed click mixture as described below, the antibody stain was performed first. Dyes for live-cell imaging experiments were used according to the manufacturer. Supplementary Table 1 presents all primary and secondary antibodies and dyes used with their respective dilutions.

**High-pressure freezing.** High-pressure freezing was performed using a Leica EM-ICE (Leica Microsystems) or a Wohlwend Compact 03 (Wohlwend GmbH). Before high-pressure freezing, cells were treated as stated above with Tf, LDL and lipid. Sapphire discs are placed on the flat side of a 0.3 mm planchette (Wohlwend GmbH, 242) and covered using a 0.025/0.275-mm planchette (Wohlwend GmbH, 389) with the 0.025-mm side facing towards the cells. Planchettes were precoated with 1-hexadecene (TCI, H0610-100ML) and blotted before use. Fluoro-Brite DMEM (Thermo Fisher Scientific, A1896701) supplemented with 20% FBS (Gibco, 10270-106) was used as a cryo preservative.

**AFS.** The protocol for the automatic freeze substitution (AFS) is depicted in Supplementary Table 2. Frozen samples were transferred from liquid nitrogen to −90 °C cold acetone. Lowicryl HM20 (Embedding Kit 15924-1, Polysciences Inc.) and K11M (Embedding Kit 18163-1, Polysciences Inc.) were used according to the manufacturer's instructions. A Leica AFS2 machine and robot were used for embedding according to the manufacturer's instructions (Leica Microsystems). The samples were embedded in AFS sample wheels (Leica Microsystems, 16707154) and AFS sample ring holders (Leica Microsystems, 16707157). The freeze substitution medium contained 0.1% uranyl acetate (Electron Microscopy Sciences, 22400) in acetone.

**Sectioning.** Fully cured resin blocks were sectioned using a Leica EM UC6 and a 35° diamond knife ultra (Diatome, MT15630). Sections were cut to a thickness of 500 nm for tomography and 100 nm for standard TEM unless stated otherwise. HM20 was sectioned at a cutting speed

of 0.8 mm s$^{-1}$ and K11M at 1.4 mm s$^{-1}$. Sections were caught on copper 200-mesh copper grids with carbon support film (Electron Microscopy Sciences, CF200-CU-50).

**Click reactions.** Chemically fixed and permeabilized samples in PBS and sections were stained using a copper-catalysed click reaction by incubation at 37 °C with a click mix of 2 μM AF-594 picolyl-azide dye (Jena bioscience, CLK-1296-1), 0.1 mM CuSO$_4$, 5 mM ascorbic acid and 0.5 mM THPTA (Jena bioscience, CLK-1010-1G) in 100 mM HEPES at pH 7.3. Chemically fixed and permeabilized samples in PBS for spinning disc microscopy were stained once for 45 min. Sections were stained on grid mats (Electron Microscopy Sciences, 71172) twice for 30 min at 37 °C and washed afterwards for 1 h at 37 °C in 100 mM HEPES and 10 times in deionized water by blotting at room temperature. All solutions were filtered using the Millex 0.22-μm filter (Merck, SLGP033RS).

**Immunogold labelling.** Sections were labelled with picolyl-azide-biotin (Jena Bioscience, CLK-1167-5) twice for 30 min at 37 °C using the respective click mix (10 μM picolyl-azide-biotin, 0.1 mM CuSO$_4$, 5 mM ascorbic acid and 0.5 mM THPTA (Jena bioscience, CLK-1010-1G) in 100 mM HEPES at pH 7.3) and washed afterwards for 30 min at 37 °C in 100 mM HEPES and 10 times in deionized water by blotting. Sections were blocked using filtered blocking buffer (2% BSA in PBS (Sigma, A7030-10G)) for 10 min. Sections were then incubated with the primary antibody against biotin (Rockland, 100-4198) at a 1:5,000 dilution in blocking buffer for 1 h. Sections were washed in blocking buffer four times for 2 min each. Subsequently, sections were incubated with Protein A gold (CMC, PAG 10NM/S) in blocking buffer for 20 min at dilutions as indicated by the manufacturer. Sections were washed in 0.1% blocking buffer (2% BSA in PBS) four times for 2 min each. Next, sections were fixed in 1% glutaraldehyde in PBS for 5 min and subsequently rinsed in excess water. Antibody and gold bead labelling was repeated a total of three times. All incubation steps were performed at room temperature unless indicated otherwise. All solutions were filtered using the Millex 0.22-μm filter (Merck, SLGP033RS).

**Section post-processing.** Sections were next stained before light microscopy on one side with multicolour TetraSpecs by incubation of grids for 5 min on a drop of freshly sonicated fiducials in PBS at 1:100 dilution for 100-nm TetraSpecs (Thermo Fisher, T7279) for K11M sections and 1:25,000 dilution for custom ordered 50-nm TetraSpecs (Thermo Fisher, C47819) for HM20 sections. After imaging samples by light microscopy, grids were coated with freshly sonicated 15 nm gold beads on both sides at dilutions indicated by the supplier in PBS (CMC, PAG 15NM/S). Then, K11M sections were stained with 1% uranyl acetate solution (Electron Microscopy Sciences, 22400) in deionized water for 7 min at room temperature, washed thoroughly in deionized water and stained with a 0.04% (m/v) lead citrate (Electron Microscopy Sciences, 17800) aqueous solution for 2 min at room temperature. Sections were thoroughly washed in deionized water and dried before imaging by EM.

## Imaging

**Spinning disk imaging.** Images of chemically fixed and permeabilized samples in PBS were acquired on an Olympus IX83 microscope, equipped with a Yokogawa CSU-W1 SoRa unit, an ORCA-Fusion (Hamamatsu) and an ORCA-Flash 4.0 V3 digital CMOS camera using the FV10-ASW 1.7. software. A 100× immersion oil objective (Olympus UApoN OTIRF) at 65 nm pixel size was used for imaging. AlexaFluor488, NBD or eGFP were excited with a 488 nm laser and the emitted light was collected between 500 and 550 nm. The AlexaFluor 594 azide was excited using a 561 nm laser and the emitted light was collected between 580 and 654 nm. AlexaFluor 647 was excited with a 640 nm laser and emitted light was collected between 665 and 705 nm. All z stacks were acquired with a 0.5-μm distance in z between frames. To ensure that all images were acquired at the same starting plane, the Olympus TruFocus

Z-drift compensation system was used. All images for one dataset were acquired using identical settings.

**Widefield imaging.** CLEM sections were imaged by widefield imaging on an inverted stand Olympus IX83 by by-passing the disc. Whole-grid overview images were taken with an Olympus U Plan SApo 20× (0.85 NA) oil objective. Images were stitched using the Olympus cellSens 4.1 software. Image stacks of single grid squares were taken using an Olympus U-ApoN 100× (1.49 NA) oil objective and an Olympus UPlan APO-HR 60× (1.50 NA) oil objective. Stack step size was chosen at 0.1 µm using a ZDC2 hardware autofocus and stage-top z piezo. The total stack size was 1 µm. All fluorophores were excited using a CoolLED pE-4000 at 100% power for 500 ms exposure time. AlexaFluor405 was excited at 365 nm using a U-FUNA (365/10 BP EX; DM 410; 440/40 BP EM) filter cube. AlexaFluor488 was excited at 470 nm using a U-FBNA filter cube (EX 482/25; DM 505; EM 5,130/40). AlexaFluor594 was excited at 550 nm, using a TxRed ET filter cube (560/40ET BP EX; T585 LPXR DM; 630/75ET BP EM). AlexaFluor647 was excited at 635 nm using a Cy5 ET filter cube (620/60ET BP EX; T660 LPXR DM; 700/75ET BP EM). Detection was done with a Hamamatsu ORCA-Fusion BT digital CMOS camera at a 23 MHz pixel clock. DM, dichromatic mirror; BP, band pass; EX, excitation filter; ET, enhanced technology.

**TEM imaging and tilt-series acquisition.** TEM of thin sections up to 100 nm in thickness was performed on a Tecnai T12 (Thermo Fisher Scientific) at 100 kV acceleration voltage with the standard single tilt sample holder. The images were taken on an F416 camera (Tietz Video and Image Processing Systems GmbH) at 4,096 × 4,096 pixels. Tomography on thick sections from 500 nm was done on a Tecnai TF30 G2 FEG-TEM (Thermo Fisher Scientific) at 300 kV acceleration voltage with a Dual-Axis Tomography Holder (Fischione Instruments, model 2040). The images were captured on a Gatan OneView camera (Gatan) at 4,096 × 4,096 pixels. Imaging on both microscopes was done with the program SerialEM[62]. Grids were placed into the sample holder with the sample facing down. Images at the T12 were acquired at magnifications of 2,900× (3.761 nm pixel size) for grid square overview images and 13,000× (0.8211 nm pixel size) for high magnification zoom ins. Images at the TF30 were acquired at magnifications of 3,100× (3.897 nm pixel size) for the grid square overview and 12,000× (1.0308 nm pixel size) for the tomogram acquisition. Tilt series were acquired from two axes, from 0° to −60° and 0° to 60°.

**Image processing, correlation and tomogram reconstruction**
Image and data post-processing was performed using Python[63], Ilastik[64], Prism 10, ICY/ec-CLEM[65], imod/3dmod[66], ImageJ/Fiji[67,68] and Microsoft Excel (v16.104.1 (26010228)). Python version 3.8, ImageJ/Fiji version 2.14.0/1.54f, Ilastik version 1.4.0-OSX, imod/3dmod version 4.12.58, Prism version 10.4.0 (527) and ICY (first version) with the ec-CLEM plugin were used.

**Ilastik models.** Ilastik was used for segmentation for the following experiments: segmentation of the plasma membrane of the live-cell (uptake traces of NBD lipid probes) samples and fixed samples (uptake traces of bifunctional lipid probes, uptake traces of Tf and LDL). In all cases, we used the two-stage autocontext pixel classification workflow of Ilastik. The training was performed in 2D and using different frames of the z stack.

**Correlation and tomogram reconstruction.** Fluorescent image stacks were acquired by widefield fluorescence imaging. Images were first preprocessed. Out-of-focus images in each stack were removed. The image stacks were merged using ImageJ/Fiji with the average intensity setting. The contrast and brightness were adjusted per each merged image. To account for shifts due to chromatic aberration, ICY ec-CLEM were used for shift correction by picking the centre of single TetraSpecs. The lipid channels were kept as the original unchanged image. For the correlation between the fluorescence image and the medium magnification TEM image, the ICY ec-CLEM plugin was used. Based on the fluorescent signal, different regions for tomography were chosen. Tomograms were reconstructed using etomo.

**CLEM penetration depth.** For the assessment of the penetration depth in vertical sections, line profiles were taken of in-focus single-plane images (line width of 10) in the lipid channel using ImageJ/Fiji, perpendicular to the section surface. Line profiles were taken only in areas with cells (determined with the respective signal from LDL and Tf). Profiles were saved as .csv files. For each biological replicate (2 per condition), 15 line profiles were taken. To determine the resolution of the microscope set up, 100 nm fiducials were analysed similarly but using a line width of 1. For the assessment of the penetration depth in horizontal sections, whole subsequent sections were first correlated using the ICY ec-CLEM plugin. Next, for higher correlation accuracy, smaller regions with whole cells were correlated again. Using the 250714_penetration_depth_horizontal.py script, the Pearson coefficient was determined between subsequent background-corrected sections of the ROIs and plotted against the respective image pair.

**Analysis of fluorescence signal densities on membrane structures from CLEM data.** Image and data post-processing was performed using Python[63] (version 3.8). In the following, 'channel' is defined as the fluorescence signal detected for Tf, LDL or lipid, respectively, and 'mask' indicates the segmentation masks for the outer globular membrane of the endosome, ILVs and RTs. The 3D (for K11M tomograms) or 2D (for HM20) models of the membranes of endosomes were generated manually using imod/3dmod. The step size in z was adjusted to the reported thickness retrieved after the tomogram reconstruction of each tomogram. Membranes were classified into the main globular membrane (outer membrane), RT (membranes protruding outward from the BM, meshed with cap) and ILVs (vesicles enclosed by the outer membrane and/or RT, meshed with cap). Surface areas and outline lengths were determined using imodinfo. The outlines of each object were generated using .wimp files of the separated objects and the Python script suman_wimp_lineprofiles.py/ suman_wimp_lineprofile_2D.py. Using the Python script ML_pdf_analysis.py/ ML_pdf_analysis_singleplane.py, line profiles were blurred to generate masks. The width of the Gaussian blur (σ 30) was set to capture the majority of fluorescence based on the Tf and LDL channel. Masks were split into regions with no overlap with the other masks, overlap with another mask or overlap with all three masks. The background-corrected fluorescence intensity of each channel (LDL, Tf and lipid) was read out over each mask region without overlap. For the pixel values for non-overlapping masks, a probability density function was generated using a Gaussian kernel with a bandwidth of 1.0 using the scikit-learn kernel density estimator. Using the probability density function per mask and channel, partial pixel values were assigned in regions of overlapping masks at the determined probability. The fluorescent pixel values were recombined per mask from the non-overlapping region and regions with overlap. The total fluorescence intensity was determined per mask and channel and divided by the membrane surface area/outline length of the corresponding mask to obtain the fluorescence density (lipid per nm² (K11M) or per nm (HM20)) of each mask and channel. The background-corrected fluorescence per channel was also read out over the total mask (combining the three masks of the outer membrane, ILVs and RT) and normalized over the total membrane surface of the whole endosome. To determine fold enrichments over the average per channel, the fluorescence density assigned to each mask was divided by the fluorescence density of the whole endosome. Maximum possible fold enrichments were calculated by taking the ratio of the total membrane surface area/outline length over the membrane surface area/outline length of the mask.

**Calculation of fluorescent density, relative density and maximum possible densities.** For every fluorescent marker, the fluorescence intensity was measured per mask and normalized to the respective membrane domain surface area, resulting in fluorescence intensities per membrane $nm^2$ (fluorescence densities) as

$$\text{fluorescence density (domain)} = \frac{\text{fluorescence intensity (in mask of domain)}}{\text{surface area (domain)}}$$

with the domains of the outer membrane, ILVs or RTs. To calculate the relative fluorescence density, the fluorescent densities of the membrane domains were each normalized over the mean fluorescent density of the full endosome as

$$\text{Relative fluorescence density}$$
$$= \text{fluorescence density (domain)} \Big/ \left( \frac{\text{fluorescence intensity (total mask)}}{\text{surface area (total)}} \right).$$

To determine the maximum possible enrichment in a domain, we assume that the total fluorescence intensity of a label is localized to one domain. Maximum fold enrichments are calculated as follows:

$$\text{max fold enrichment}$$
$$= \left( \frac{\text{total fluorescence intensity}}{\text{surface area of domain}} \right) \Big/ \left( \frac{\text{total fluorescence intensity}}{\text{total surface area}} \right),$$

and thus

$$\text{max fold enrichment} = \left( \frac{\text{total surface area}}{\text{surface area of domain}} \right).$$

The relative densities were determined for the two sample types SM loading and fatty acid loading. The significance of changes between both sample types was determined by a Student's *t*-test. For this test the data distribution was assumed to be normal but this was not formally tested.

**Reporting summary**
Further information on research design is available in the Nature Portfolio Reporting Summary linked to this article.

## Data availability
The imaging data can be accessed at https://doi.org/10.17617/3.LZPQ3K (ref. 69) and https://doi.org/10.17617/3.9O0ZNY (ref. 70). All other data supporting the findings of this study are available from the corresponding author on reasonable request. Source data are provided with this paper.

## Code availability
The corresponding codes to analyse all data are avilable via Zenodo at https://doi.org/10.5281/zenodo.14723745 (ref. 71) and https://doi.org/10.5281/zenodo.17788790 (ref. 72).

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

## Acknowledgements

A.N. gratefully acknowledges financial support by the European Research Council (ERC) under the European Union's Horizon 2020 research and innovation programme (grant agreement nos. GA 758334 ASYMMEM and AURORA). A.N. acknowledges financial support by the Deutsche Forschungsgemeinschaft (DFG) via the TRR83 consortium. This research was supported by an Allen Distinguished Investigator Award, a Paul G. Allen Frontiers Group advised grant of the Paul G. Allen Family Foundation to A.N. O.A. gratefully acknowledges financial support by the Israel Science Foundation (grant no. 3729/20), the ERC under the European Union's Horizon 2020 research and innovation programme (grant agreement no. 851080), the Henry Chanoch Krenter Institute for Biomedical Imaging and Genomics and support given by the Heineman Foundation through Minerva. O.A. is the incumbent of the Miriam Berman presidential development chair. G.F. thanks the support by the Simons Foundation (CCBx programme, no. 1157392). We thank the following services and facilities at MPI-CBG Dresden for their support: the Electron Microscopy Facility, the Light Microscopy Facility, the Genome Engineering Facility, the Scientific Computing Facility and the Organoid and Stem Cell Facility. We thank J. Peychl, B. Schroth-Diez and T. Fürstenhaupt for their outstanding support and expert advice. We thank the Core Facility Cellular Imaging at the Faculty of Medicine Carl Gustav Carus at TU Dresden for technical support. We thank P. Ronchi and A. von Appen for expert advice during the development of the CLEM workflow and V. Oorschot for advising on immunogold labelling.

## Author contributions

K.B. synthesized the lipid probes. H.M.L. and W.L. prepared the samples and acquired the datasets. H.M.L. and W.L. sectioned blocks. H.M.L., S.K., W.L., G.F. and Y.K. analysed imaging data. F.E. constructed the LED photoreactors. H.M.L., S.K., N.S. and M.W.-B. developed the CLEM lipid imaging workflow. H.M.L., O.A. and A.N.

designed the project. O.A. and A.N. supervised the research. H.M.L. and A.N. wrote the manuscript. All authors read and commented on the manuscript.

## Funding

## Competing interests

The authors declare no competing interests.

## Additional information

**Extended data** is available for this paper at https://doi.org/10.1038/s41556-026-01915-x.

**Correspondence and requests for materials** should be addressed to H. Mathilda Lennartz or André Nadler.

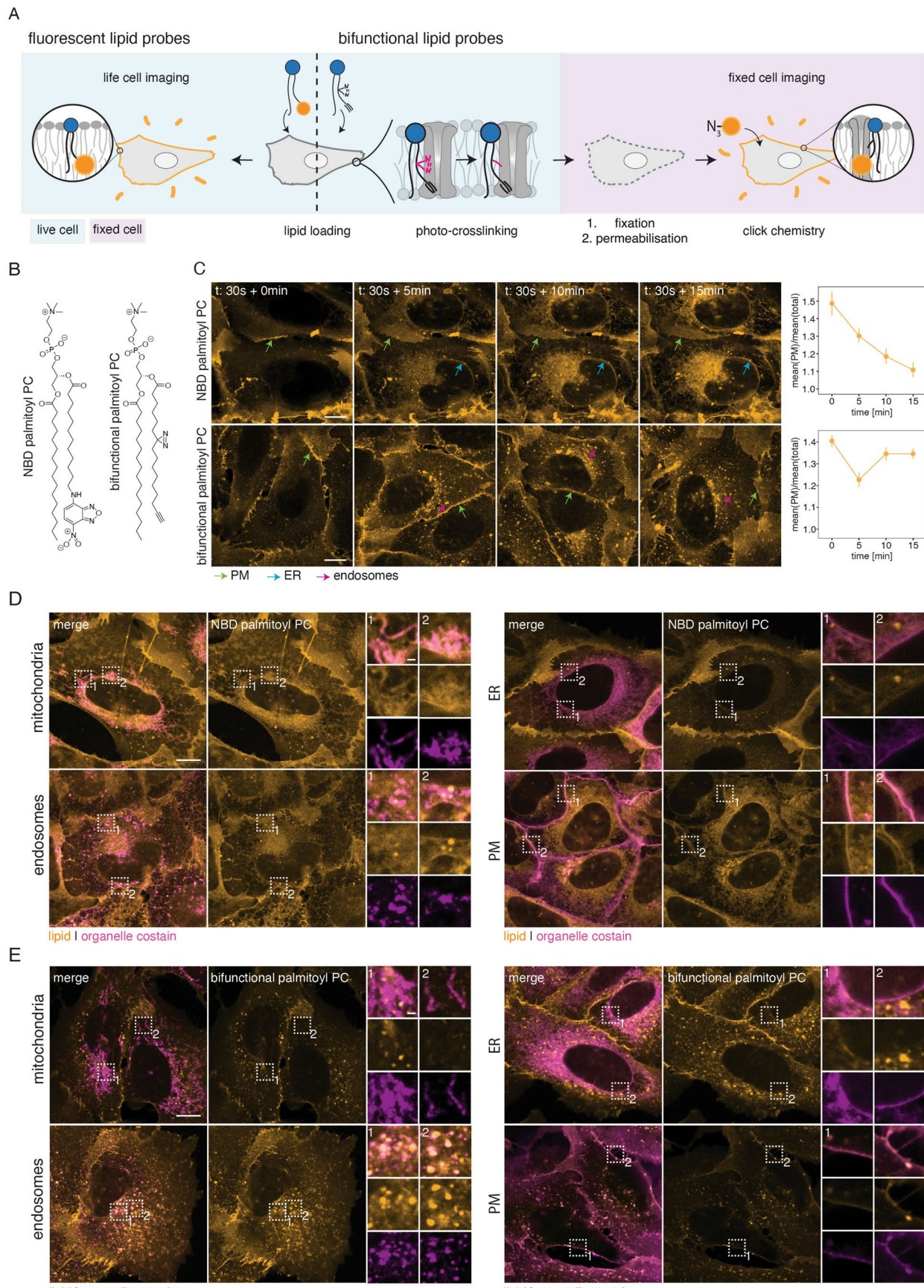

**Extended Data Fig. 1 | See next page for caption.**

**Extended Data Fig. 1 | NBD and diazirine-alkyne labeled lipid probes show strong differences in organellar localization. A:** Fluorescent lipid probes can be used for live-cell imaging, while bifunctional lipid probes are suitable for fixed cell imaging. Bifunctional lipid probes require loading into live cells, photo-crosslinking, fixation, and labeling of the fixed samples by click chemistry. **B:** The chemical structures of the NBD PC(16:0 | 06:0-NBD) and bifunctional diazirine-alkyne PC(16:0 | Y). **C:** U2OS wildtype cells were loaded with an NBD PC(16:0 | 06:0-NBD) for 30 s or a bifunctional diazirine-alkyne palmitoyl PC for 30 s and chased for 0, 5, 10 and 15 min each. Colored arrows indicate lipid localizations to different membrane-bound organelles. Lipid uptake was analyzed by assessing lipid amount in the plasma membrane over time relative to total lipid content. Mean and SEM are shown. Three independent samples per conditions and timepoint (n(field of view): 18) were analyzed. Each field of view includes at least 5 cells. **D, E:** Lipid localization to specific organelles was confirmed with organelle-specific labels against mitochondria, endosomes, ER, and the plasma membrane for the NBD lipid and the bifunctional diazirine-alkyne lipid. Scale bar: 10 μm. Representative images of 3 biological replicates are shown. Source numerical data are available in source data.

A: classic protocoll

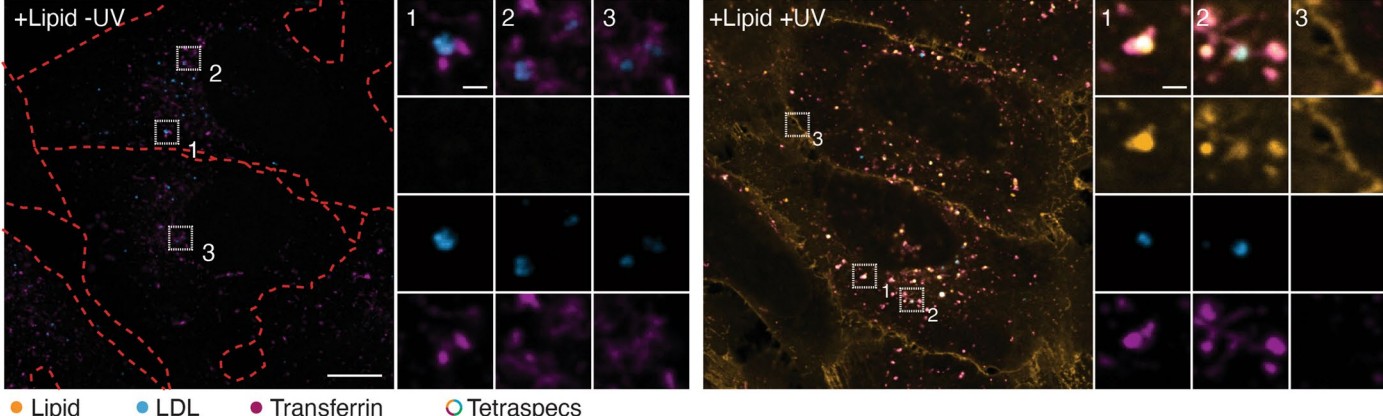

B: HM20

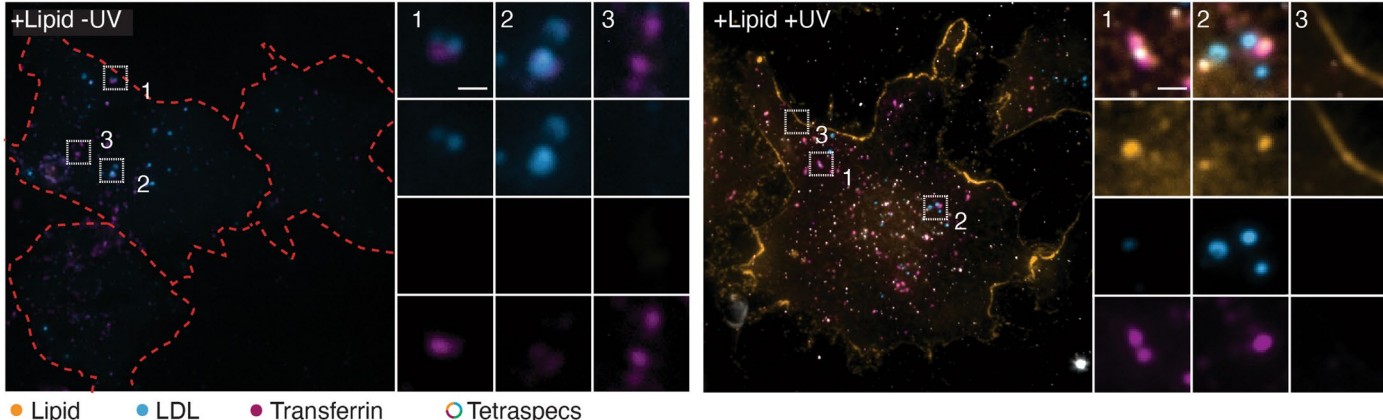

C: K11M

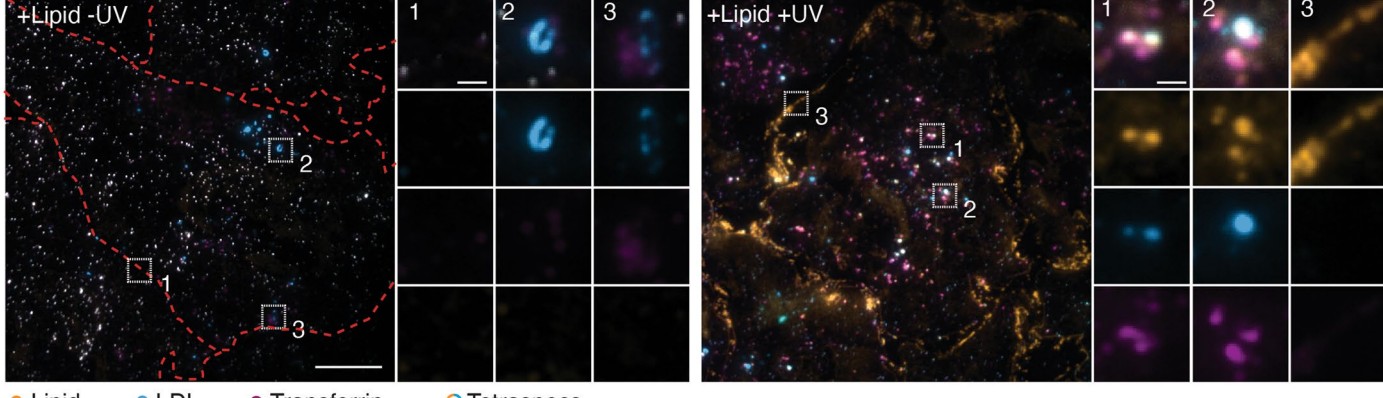

**Extended Data Fig. 2 | Non-photo-crosslinked bifunctional lipid probes are removed during automatic freeze substitution. A:** U2OS cells were loaded for 4 min with PC(16:0 | Y), not photoactivated ( + Lipid -UV) or photoactivated ( + Lipid +UV), and prepared according to the "classic" lipid imaging protocol by chemical fixation, permeabilization, stain and imaging in PBS, or **B:** by embedding in HM20 and **C:** K11M according to the here proposed Lipid-CLEM workflow. The HM20 and K11M sections are 100 nm thin. Red-dotted lines show the cell outlines. Scale bars: 10 μm. The images of the right column are shown in Figs. 1c and 2a. The datasets for all conditions include 4 (A), 3 (B) and 3 (C) independent replicates, that yielded similar results.

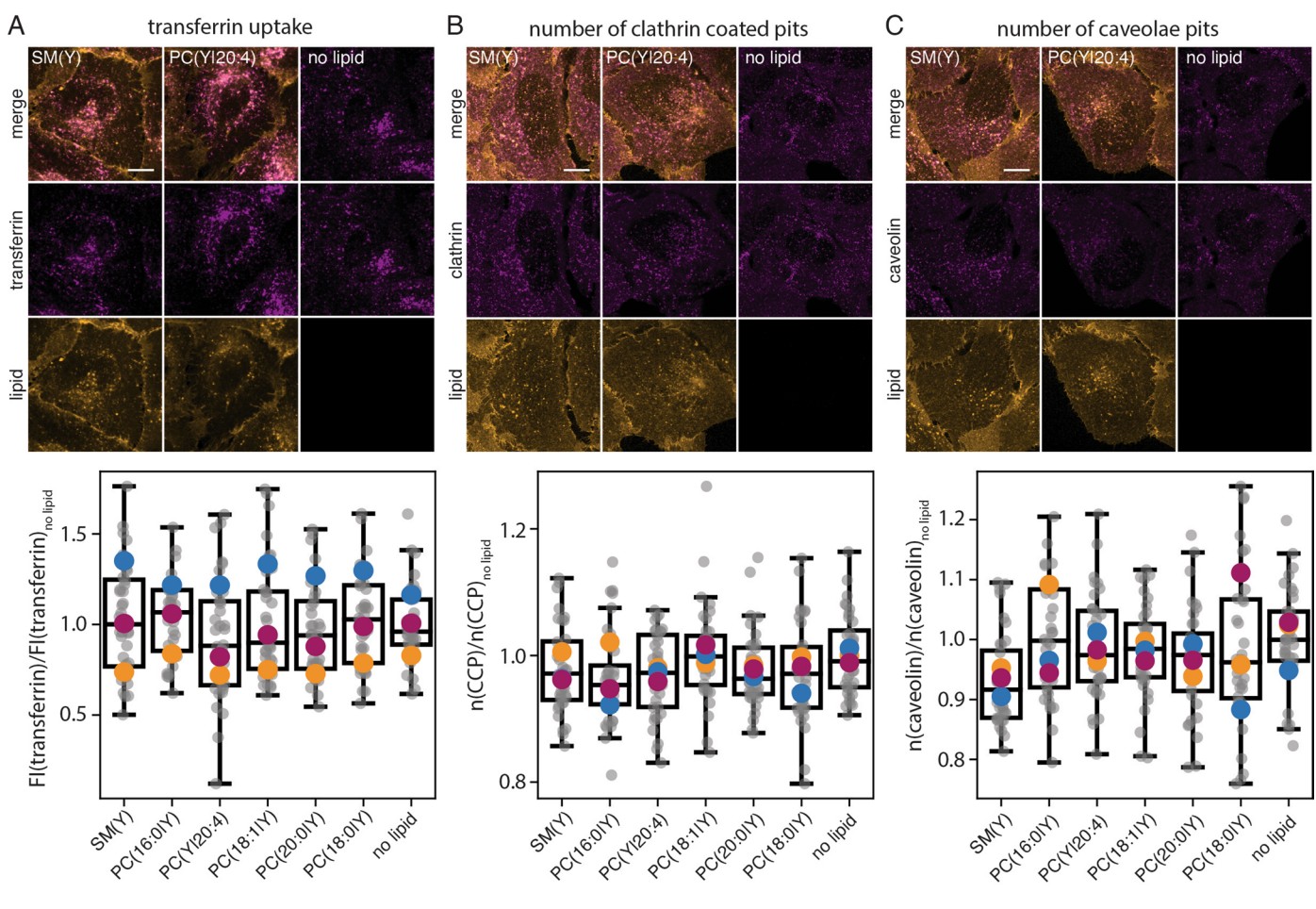

replicate 1    replicate 2    replicate 3

**Extended Data Fig. 3 | Lipid loading does not perturb major endocytic membrane remodeling events.** Several bifunctional lipids, including SM(Y) and PCs of different fatty acid composition, were loaded onto the PM of U2OS cells. Transferrin uptake (**A**), number of clathrin-coated pits (**B**), and caveolae number (**C**) at the PM were quantified. Three independent replicates are color-coded as indicated (n(field of view): 18, per condition, per lipid species). Each field of view includes at least 5 cells. Boxplots show the median (central line) and interquartile range (box bounds: Q1 to Q3), with Tukey-style whiskers extending to 1.5 × IQR. Individual data points are shown in grey. The colored dots highlight the mean value of each replicate. Scale bar: 10 μm. Source numerical data are available in source data.

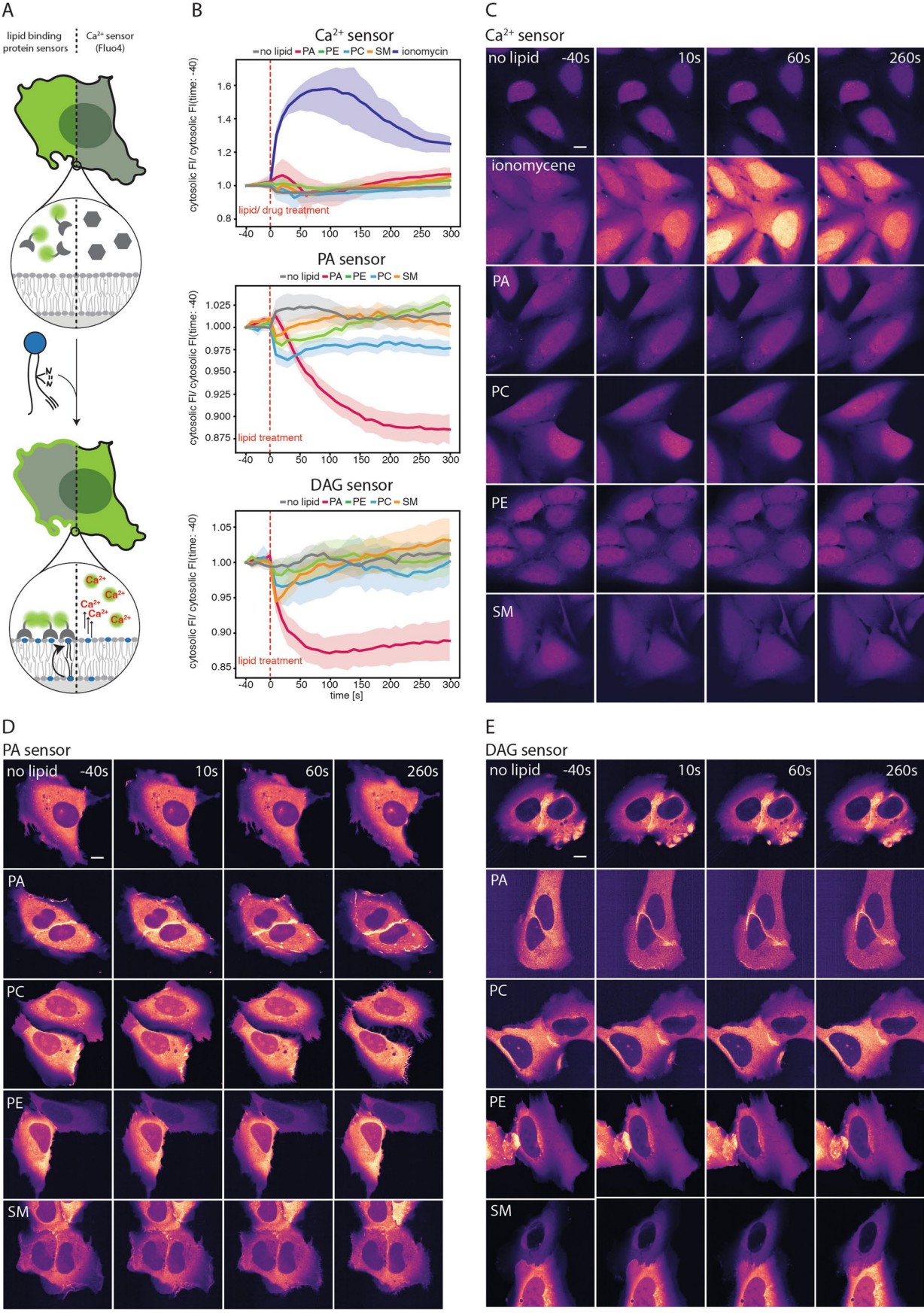

**Extended Data Fig. 4 | See next page for caption.**

**Extended Data Fig. 4 | Effect of lipid loading on cellular signaling events. A**: To test if lipid loading induces major signaling events, the localization of DAG and PA protein sensors, as well as intracellular Calcium levels, were monitored during lipid loading using live cell imaging. A drop of sensor intensity in the cytosol is expected for the PA and DAG sensor upon recruitment to other membranes, whereas the calcium sensor intensity increases upon binding Ca2+ in the cytosol. **B:** The traces of the three sensors are plotted as the sensor intensity in the cytosol normalized against the cytosolic sensor intensity at time point −40 s before adding the lipid. Images were acquired every 10 s. Representative images of the sensor intensity are shown for the timepoints −40 s, 10 s, 60 s, and 260 s, and for the Calcium sensor (**C**), PA sensor (**D**), and DAG sensor (**E**). Timepoint 0 marks the addition of the lipid. Scale bar: 10 μm. Three independent samples were measured (n: 30, per condition). Median and 95% confidence interval are plotted. Source numerical data are available in source data.

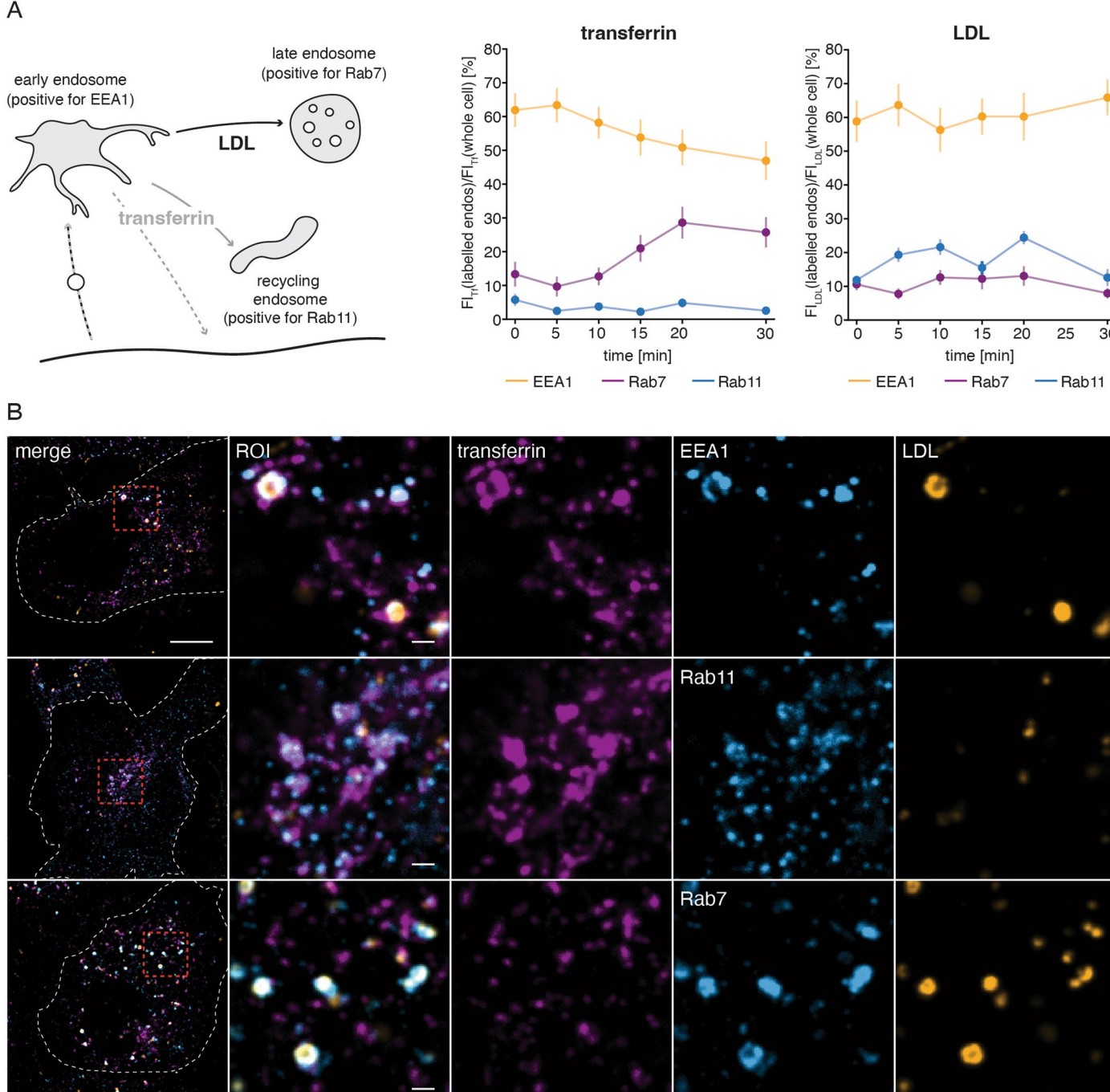

**Extended Data Fig. 5 | LDL and Transferrin specifically label endosomal compartments. A:** Transferrin and LDL were first assessed separately for their localization to distinct endosomal compartments. Transferrin was loaded for 4 min, and chased for 0, 5, 10, 15, 20 and 30 min. LDL was loaded for 15 min and chased for 0, 5, 10, 15, 20, and 30 min. EEA1 was used to mark early endosomes, Rab7 was used for late endosomes, and Rab11 was used for recycling endosomes. The fluorescent values of either transferrin or LDL were read out over the segmentation markers and normalized to the total transferrin or LDL fluorescence signal. The analysis excludes compartments that are positive for both early and late endosomal markers or markers of early and recycling endosomes. Mean values are shown, and error bars represent the standard deviation. The experiment was done as a single biological replicate (n(field of view): 6, per time point). Each field of view includes at least 5 cells. Source numerical data are available in source data. **B:** Transferrin and LDL were loaded together to assess their combined localization to distinct endosomal compartments. Representative images of U2OS cells labelled for 15 min with LDL (orange), chased for 15 min, and labeled for 4 min with transferrin (magenta). Endosomal compartments were either immunolabelled with EEA1, Rab7, or Rab11 (blue). The outline of the representative cell is highlighted by a white dotted line. Scale bar: 10 µm, ROI: 1 µm. The Antibody co-stain dataset includes 1 independent replicate.

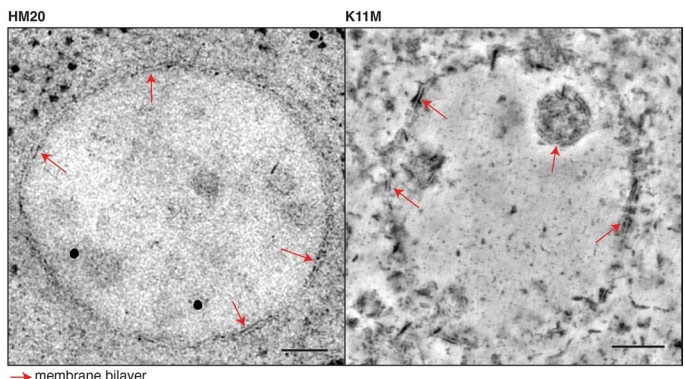

**Extended Data Fig. 6 | Preservation of membrane ultrastructure in different resins.** On the left, an early endosome is shown embedded in HM20 (100 nm section). On the right, an early endosome is embedded in K11M (tomogram image from a 500 nm section). Red arrows point to regions where the lipid bilayer is visible. Scale bars are 100 nm. The datasets include 12 (K11M) and 5 (HM20) independent replicates, that yielded similar results.

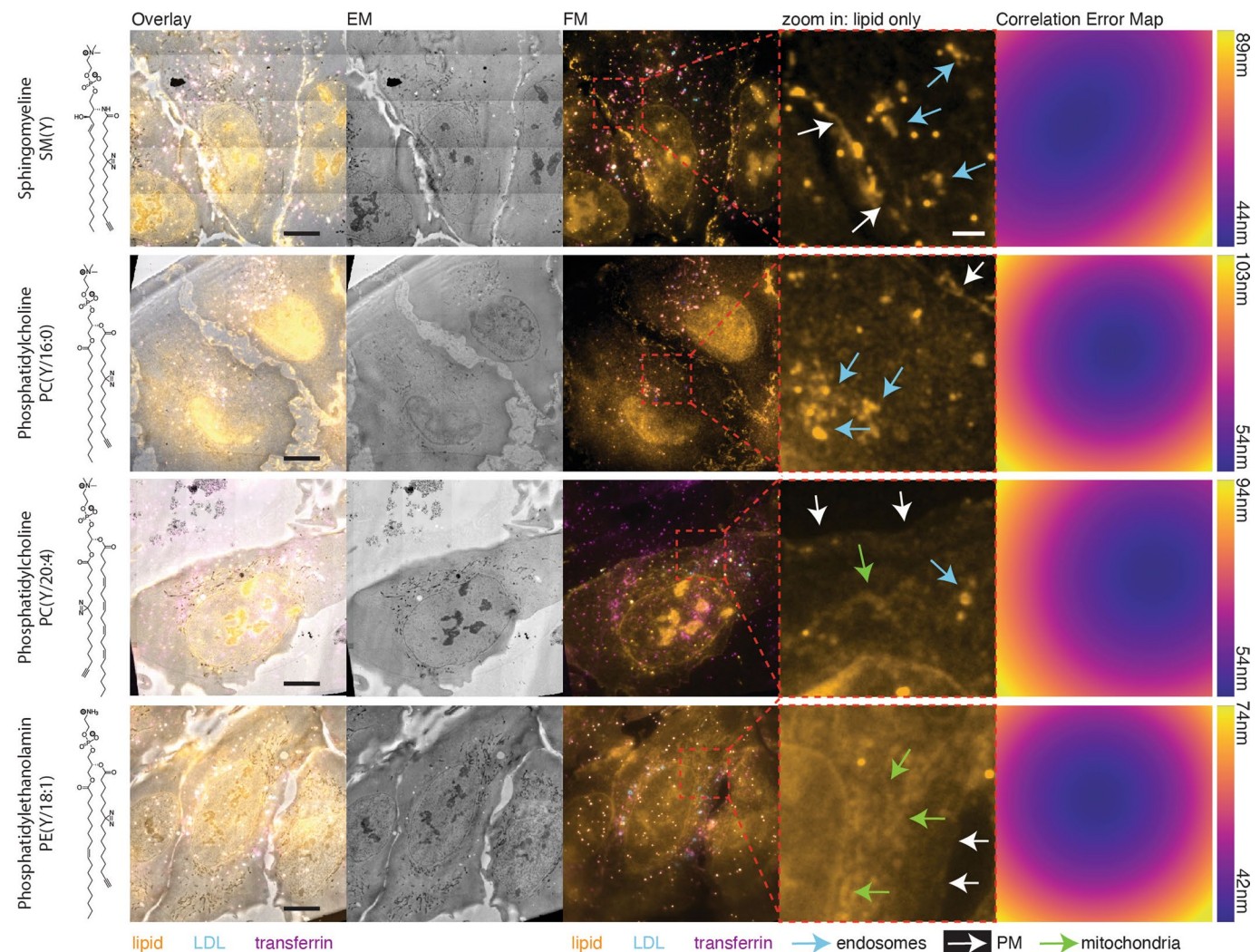

lipid · LDL · transferrin · lipid · LDL · transferrin · → endosomes · ⇒ PM · → mitochondria

**Extended Data Fig. 7 | Correlation maps for Lipid CLEM of various lipid classes and species.** All chemical structures are indicated on the left. Representative EM, FM and correlated images are shown for all lipids (the same images are shown in Main Fig. 4). A region of interest of each cell highlights the differential lipid distribution for the different lipids tested. Correlation error maps of the whole images are indicated on the right. Scale bar: 10 µm, zoom in: 1 µm. Independent experiments per lipid type: 2.

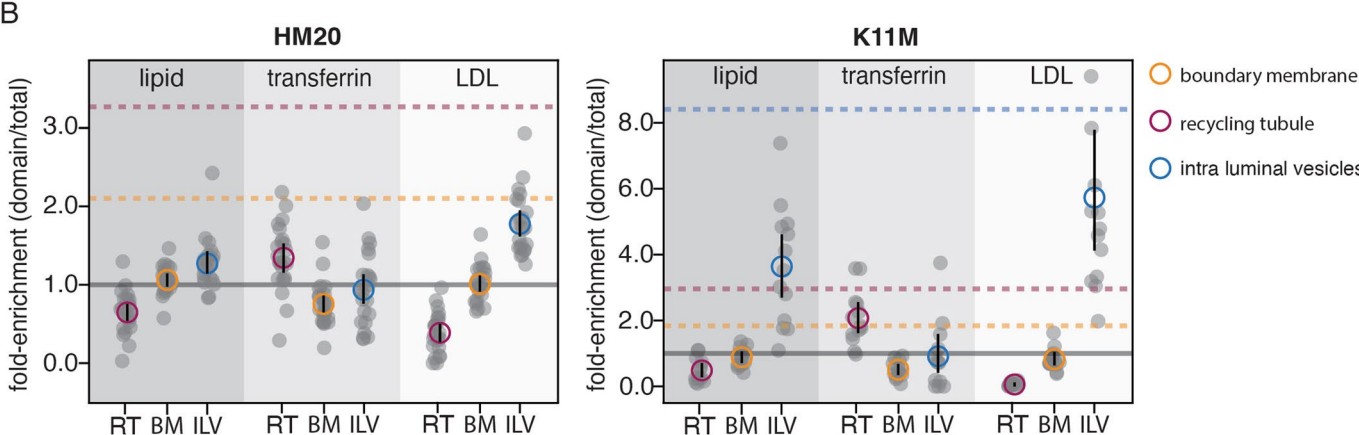

**Extended Data Fig. 8 | Endosomal lipid partitioning trends are recapitulated in HM20. A:** HM20 embedded samples loaded with SM(Y) were sectioned into 100 nm sections and stained. Three representative endosomes are shown: the corresponding EM image; the merge of the fluorescence channels of SM(Y) (orange), transferrin (Tf, purple), and LDL (blue); and the surface model of the endosomes membranes overlayed with each fluorescent signal. The individual quantification for each endosome is shown on the right. RT – recycling tubule, BM – boundary membrane, ILV – intraluminal vesicle. Scale bar: 200 nm. Fluorescence of each endosome is scaled individually. **B:** The relative fluorescent densities are plotted for all analyzed endosomes for the lipid SM(Y) of the samples embedded in HM20 and compared against K11M. For K11M relative values for lipid signal over membrane surface area (2D) are plotted. For HM20 relative values for lipid signal over membrane outlines (1D) area are plotted.

A direct quantitative comparison is thus not possible, but overall enrichment and depletion trends can be compared and are conserved. The data for K11M is also shown in main text Fig. 5d. The mean densities are shown for the different domains: recycling tubule (magenta), boundary membrane (orange), and ILV (blue), and for the respective fluorescent signals of lipid, transferrin, and LDL. Maximum possible fold-enrichments are indicated by the dotted horizontal lines for the boundary membrane (orange), recycling tubule (magenta), and intraluminal vesicles (blue). The maximum possible fold-enrichment of ILVs in HM20 (4.58) is not shown. No enrichment is indicated by the gray line at 1.0. Error bars are shown as the 95% confidence interval of the mean values. 2 independent biological replicates were performed both for HM20 (n: 24) and K11M (n: 13). Source numerical data are available in source data.

**Extended Data Fig. 9 | Transferrin and sphingomyelin co-localize in clathrin-coated pits. A:** A representative image of a U2OS cell loaded with transferrin (purple) and SM(Y) (orange) and immuno co-stained for clathrin (blue). The basal PM is shown. Scale bar: 10 μm. ROIs show clathrin-coated pits and their co-localisation with SM(Y) and transferrin. Scale bar: 500 nm. **B:** Quantification of the mean fluorescence (FI) of SM or transferrin signal red out over the mask of clathrin-coated pits, normalized over the mean FI over the whole cell. Boxplots show the median (central line) and interquartile range (box bounds: Q1 to Q3), with Tukey-style whiskers extending to 1.5 × IQR. Individual data points are shown in grey. Three independent samples were measured (n(field of view): 18). Each field of view includes at least 5 cells. Source numerical data are available in source data.

André Nadler

# Reporting Summary

## Statistics

For all statistical analyses, confirm that the following items are present in the figure legend, table legend, main text, or Methods section.

| n/a | Confirmed | |
|---|---|---|
| ☐ | ☒ | The exact sample size (*n*) for each experimental group/condition, given as a discrete number and unit of measurement |
| ☐ | ☒ | A statement on whether measurements were taken from distinct samples or whether the same sample was measured repeatedly |
| ☐ | ☒ | The statistical test(s) used AND whether they are one- or two-sided *Only common tests should be described solely by name; describe more complex techniques in the Methods section.* |
| ☒ | ☐ | A description of all covariates tested |
| ☒ | ☐ | A description of any assumptions or corrections, such as tests of normality and adjustment for multiple comparisons |
| ☐ | ☒ | A full description of the statistical parameters including central tendency (e.g. means) or other basic estimates (e.g. regression coefficient) AND variation (e.g. standard deviation) or associated estimates of uncertainty (e.g. confidence intervals) |
| ☐ | ☒ | For null hypothesis testing, the test statistic (e.g. *F*, *t*, *r*) with confidence intervals, effect sizes, degrees of freedom and *P* value noted *Give P values as exact values whenever suitable.* |
| ☐ | ☒ | For Bayesian analysis, information on the choice of priors and Markov chain Monte Carlo settings |
| ☒ | ☐ | For hierarchical and complex designs, identification of the appropriate level for tests and full reporting of outcomes |
| ☐ | ☒ | Estimates of effect sizes (e.g. Cohen's *d*, Pearson's *r*), indicating how they were calculated |

*Our web collection on statistics for biologists contains articles on many of the points above.*

## Software and code

Policy information about availability of computer code

| Data collection | Olympus cellSens Dimension (Version 4.1), SerialEM (Versions: 4.2.obeta and 4.1.obeta) |
|---|---|
| Data analysis | python 3 (Version 3.8)<br>Ilastik (Version 1.4.0-OSX)<br>Prism 10 (Version 10.4.0 (527))<br>ICY (first version), using ec-CLEM plugin (Version 1.1.0.0)<br>ImageJ/Fiji (Version 2.14.0/ 1.54f)<br>IMOD/ 3dmod (Version 4.12.58)<br>Microsoft® Excel (Version 16.104.1 (26010228)) |

For manuscripts utilizing custom algorithms or software that are central to the research but not yet described in published literature, software must be made available to editors and reviewers. We strongly encourage code deposition in a community repository (e.g. GitHub). See the Nature Portfolio guidelines for submitting code & software for further information.

## Data

Policy information about availability of data

All manuscripts must include a data availability statement. This statement should provide the following information, where applicable:

 - Accession codes, unique identifiers, or web links for publicly available datasets
 - A description of any restrictions on data availability
 - For clinical datasets or third party data, please ensure that the statement adheres to our policy

The imaging data can be accessed at https://doi.org/10.17617/3.LZPQ3K and https://doi.org/10.17617/3.9O0ZNY. The corresponding codes to analyze all data can be found at https://doi.org/10.5281/zenodo.14723745 and https://doi.org/10.5281/zenodo.17788790. Source data have been provided in Source Data. All other data supporting the findings of this study are available from the corresponding author on reasonable request.

## Research involving human participants, their data, or biological material

Policy information about studies with human participants or human data. See also policy information about sex, gender (identity/presentation), and sexual orientation and race, ethnicity and racism.

| Reporting on sex and gender | n/a |
| --- | --- |
| Reporting on race, ethnicity, or other socially relevant groupings | n/a |
| Population characteristics | n/a |
| Recruitment | n/a |
| Ethics oversight | n/a |

Note that full information on the approval of the study protocol must also be provided in the manuscript.

# Field-specific reporting

Please select the one below that is the best fit for your research. If you are not sure, read the appropriate sections before making your selection.

☒ Life sciences ☐ Behavioural & social sciences ☐ Ecological, evolutionary & environmental sciences

For a reference copy of the document with all sections, see nature.com/documents/nr-reporting-summary-flat.pdf

# Life sciences study design

All studies must disclose on these points even when the disclosure is negative.

| Sample size | No sample size calculation was performed. For classik lipid imaging/ light micrscopy experiments smaple sizes were chosen based on previous reports (Iglesias-Artola, J. M. et al; Nature 2025). All experimentens were performed in independent biological or technical triplicates for follow up quantification, unless stated otherwise. For the determinatin of the lipid densities in endosomes 2 biological replicates were performed per condition, comprising at least 5 endosomes per replicate, in both resins. The low variability between the different labeling conditions shows that a small number of independent repeats is sufficient. Sample sizes of CLEM experiments were chosen based on the sample sizes of previous comparable CLEM reports (Kukulski, W. et al.; J. Cell Biol. 2011). |
| --- | --- |
| Data exclusions | Images were excluded if the acquisition process failed by either being out of focus or if unwanted cellular debris or clamps precluded image analysis. For tomograms and TEM, images were excluded if the tomogram reconstruction failed, if endosomes did not contain either recycling tubules and/ or intraluminal vesicles, or if cells were visibly not intact. |
| Replication | All attempts at replication were successfull between biological and technical replicas. |
| Randomization | Region of interest to acquire images were selected automatically by the software of the microscope (CellSens software) for light microscopy. If necessary, with low cell density, regions for imaging were selected with cells in the field of view. For CLEM, endosomes were selected based on the co-localization of the transferrin and LDL markers in order to identify early endosomes as explained in the main text. |
| Blinding | Data collection and analysis were not performed blind to the conditions of the experiments. |

# Behavioural & social sciences study design

All studies must disclose on these points even when the disclosure is negative.

| | |
|---|---|
| Study description | *Briefly describe the study type including whether data are quantitative, qualitative, or mixed-methods (e.g. qualitative cross-sectional, quantitative experimental, mixed-methods case study).* |
| Research sample | *State the research sample (e.g. Harvard university undergraduates, villagers in rural India) and provide relevant demographic information (e.g. age, sex) and indicate whether the sample is representative. Provide a rationale for the study sample chosen. For studies involving existing datasets, please describe the dataset and source.* |
| Sampling strategy | *Describe the sampling procedure (e.g. random, snowball, stratified, convenience). Describe the statistical methods that were used to predetermine sample size OR if no sample-size calculation was performed, describe how sample sizes were chosen and provide a rationale for why these sample sizes are sufficient. For qualitative data, please indicate whether data saturation was considered, and what criteria were used to decide that no further sampling was needed.* |
| Data collection | *Provide details about the data collection procedure, including the instruments or devices used to record the data (e.g. pen and paper, computer, eye tracker, video or audio equipment) whether anyone was present besides the participant(s) and the researcher, and whether the researcher was blind to experimental condition and/or the study hypothesis during data collection.* |
| Timing | *Indicate the start and stop dates of data collection. If there is a gap between collection periods, state the dates for each sample cohort.* |
| Data exclusions | *If no data were excluded from the analyses, state so OR if data were excluded, provide the exact number of exclusions and the rationale behind them, indicating whether exclusion criteria were pre-established.* |
| Non-participation | *State how many participants dropped out/declined participation and the reason(s) given OR provide response rate OR state that no participants dropped out/declined participation.* |
| Randomization | *If participants were not allocated into experimental groups, state so OR describe how participants were allocated to groups, and if allocation was not random, describe how covariates were controlled.* |

# Ecological, evolutionary & environmental sciences study design

All studies must disclose on these points even when the disclosure is negative.

| | |
|---|---|
| Study description | *Briefly describe the study. For quantitative data include treatment factors and interactions, design structure (e.g. factorial, nested, hierarchical), nature and number of experimental units and replicates.* |
| Research sample | *Describe the research sample (e.g. a group of tagged Passer domesticus, all Stenocereus thurberi within Organ Pipe Cactus National Monument), and provide a rationale for the sample choice. When relevant, describe the organism taxa, source, sex, age range and any manipulations. State what population the sample is meant to represent when applicable. For studies involving existing datasets, describe the data and its source.* |
| Sampling strategy | *Note the sampling procedure. Describe the statistical methods that were used to predetermine sample size OR if no sample-size calculation was performed, describe how sample sizes were chosen and provide a rationale for why these sample sizes are sufficient.* |
| Data collection | *Describe the data collection procedure, including who recorded the data and how.* |
| Timing and spatial scale | *Indicate the start and stop dates of data collection, noting the frequency and periodicity of sampling and providing a rationale for these choices. If there is a gap between collection periods, state the dates for each sample cohort. Specify the spatial scale from which the data are taken* |
| Data exclusions | *If no data were excluded from the analyses, state so OR if data were excluded, describe the exclusions and the rationale behind them, indicating whether exclusion criteria were pre-established.* |
| Reproducibility | *Describe the measures taken to verify the reproducibility of experimental findings. For each experiment, note whether any attempts to repeat the experiment failed OR state that all attempts to repeat the experiment were successful.* |
| Randomization | *Describe how samples/organisms/participants were allocated into groups. If allocation was not random, describe how covariates were controlled. If this is not relevant to your study, explain why.* |
| Blinding | *Describe the extent of blinding used during data acquisition and analysis. If blinding was not possible, describe why OR explain why blinding was not relevant to your study.* |

Did the study involve field work? ☐ Yes ☐ No

## Field work, collection and transport

| | |
|---|---|
| Field conditions | *Describe the study conditions for field work, providing relevant parameters (e.g. temperature, rainfall).* |
| Location | *State the location of the sampling or experiment, providing relevant parameters (e.g. latitude and longitude, elevation, water depth).* |
| Access & import/export | *Describe the efforts you have made to access habitats and to collect and import/export your samples in a responsible manner and in compliance with local, national and international laws, noting any permits that were obtained (give the name of the issuing authority, the date of issue, and any identifying information).* |
| Disturbance | *Describe any disturbance caused by the study and how it was minimized.* |

# Reporting for specific materials, systems and methods

We require information from authors about some types of materials, experimental systems and methods used in many studies. Here, indicate whether each material, system or method listed is relevant to your study. If you are not sure if a list item applies to your research, read the appropriate section before selecting a response.

### Materials & experimental systems

| n/a | Involved in the study |
|---|---|
| ☐ | ☒ Antibodies |
| ☐ | ☒ Eukaryotic cell lines |
| ☒ | ☐ Palaeontology and archaeology |
| ☒ | ☐ Animals and other organisms |
| ☒ | ☐ Clinical data |
| ☒ | ☐ Dual use research of concern |
| ☒ | ☐ Plants |

### Methods

| n/a | Involved in the study |
|---|---|
| ☒ | ☐ ChIP-seq |
| ☒ | ☐ Flow cytometry |
| ☒ | ☐ MRI-based neuroimaging |

## Antibodies

| | |
|---|---|
| Antibodies used | Rab5 (rabbit), Cell Signaling Technology CST-3547S, Clone: C8B1, Lot: 7, dilution: 1:500<br>Rab7 (rabbit), Cell Signaling Technology CST-9367S, Clone: D95F2, Lot: 3, dilution: 1:200<br>Tom20 (mouse), Santa Cruz sc-17764, Clone: F-10, Lot: H0320, dilution: 1:1000<br>Calnexin 1 (mouse) Abcam ab112995, Clone: 6F12BE10, Lot: GR3246794-6, dilution: 1:1000<br>Lamin A/C (mouse), Cell Signaling Technology CST-4777S, Clone:4C11 Lot:5, dilution: 1:1000<br>BAP31 (mouse), Enzo Life Sciences ALX-804-601-C100, Clone: A1/182 Lot: L15093, dilution: 1:1000<br>Calreticulin (mouse), Abcam ab22683, Clone: FMC 75, Clone: Lot:GR3361946-5, dilution: 1:1000<br>Rab11 (mouse), BD Bioscience 610657, Clone: 47, Lot: 2181662, dilution: 1:200<br>EEA1 (mouse), BD Bioscience 610457, Clone: 14, Lot: 1117266, dilution: 1:200<br>MAVS (mouse), Santa Cruz, Sc-166583, Clone: E-3, Lot: K0722, dilution: 1:750<br>EEA1 (rabbit), kind gift from the Zerial lab, Clone: 07JF, dilution: 1:1000<br>biotin (rabbit), Rockland, # 100-4198, Clone: /, Lot: 50151, dilution: 1:5000<br>clathrin (rabbit), proteintech, #10852-1-AP, Clone: /, Lot: 00097561, dilution: 1:500<br>caveolin (rabbit), abcam, #2910, Clone: /, Lot: GR3362124-1, dilution: 1:500 |
| Validation | All primary antibodies were used directly from the manufacturer. All antibodies were validated by the manufacturers to be suitable for immunofluorescence (IF) and reactive with Human samples.<br><br>Rab5, CST-3547S -> Applications: WB, IF Species Reactivity: Human, Mouse, Rat, Monkey<br>Rab7, CST-9367S -> Applications: WB, IP, IF. Species Reactivity: Human, Mouse, Rat, Monkey<br>Tom20, sc-17764 -> Applications: WB, IP, IF, IHC(P), ELISA. Species Reactivity: Human, Mouse, Rat<br>Calnexin, ab112995 -> Applications: WB, IF, ICC, IHC(P), FC. Speacies Reactivity: Human<br>Lamin A/C, CST-4777S -> Applications: WB, IF, IP, IHC, IP. Species Reactivity: Human, Mouse, Rat, Monkey<br>BAP31, ALX-804-601-C100 ->Applications: WB, IF, IP, ICC, FC, ELISA, IHC. Species Reactivity: Human, Monkey<br>Calreticulin, ab22683 -> Applications: IF, ICC, FC, IHC(P). Species Reactivity: Human<br>Rab11, 610657 -> Applications: WB, IP, IHC(P). Species Reactivity: Human<br>EEA1, 610457 -> Applications: WB, IF, IP, IHC(P). Species Reactivity: Human<br>MAVS, Sc-166583 -> Applications: WB, IF, IP, IHC(P), ELISA. Species Reactivity: Human<br>EEA1 (Zerial) -> Applications: WB, IF. Species Reactivity: Human<br>biotin, 100-4198 -> Applications: ELISA. Target Reactivity: Biotin<br>clathrin, 10852-1-AP -> Applications: WB, IHC, IF/ICC, IP, ELISA. Target Reactivity: Human, Mouse, Rat, Canine, Monkey<br>caveolin, 2910 -> Applications: WB, IP, ICC/IF. Target Reactivity: Human, Mouse, Rat |

# Eukaryotic cell lines

Policy information about cell lines and Sex and Gender in Research

| | |
|---|---|
| Cell line source(s) | U-2 OS cells (ACC 785) were purchaed from DSMZ and were originally obatained from a female. |
| Authentication | STR analysis was performed on 17 loci to the global standard ANSI/ATCC ASN-0002.1-2021 (2021) resulted in an authentic STR profile of the reference STR database. |
| Mycoplasma contamination | All cell lines are routinely tested for Mycoplasma contamination, and tested negative for mycoplasma. |
| Commonly misidentified lines (See ICLAC register) | none |

# Palaeontology and Archaeology

| | |
|---|---|
| Specimen provenance | *Provide provenance information for specimens and describe permits that were obtained for the work (including the name of the issuing authority, the date of issue, and any identifying information). Permits should encompass collection and, where applicable, export.* |
| Specimen deposition | *Indicate where the specimens have been deposited to permit free access by other researchers.* |
| Dating methods | *If new dates are provided, describe how they were obtained (e.g. collection, storage, sample pretreatment and measurement), where they were obtained (i.e. lab name), the calibration program and the protocol for quality assurance OR state that no new dates are provided.* |

☐ Tick this box to confirm that the raw and calibrated dates are available in the paper or in Supplementary Information.

| | |
|---|---|
| Ethics oversight | *Identify the organization(s) that approved or provided guidance on the study protocol, OR state that no ethical approval or guidance was required and explain why not.* |

Note that full information on the approval of the study protocol must also be provided in the manuscript.

# Animals and other research organisms

Policy information about studies involving animals; ARRIVE guidelines recommended for reporting animal research, and Sex and Gender in Research

| | |
|---|---|
| Laboratory animals | *For laboratory animals, report species, strain and age OR state that the study did not involve laboratory animals.* |
| Wild animals | *Provide details on animals observed in or captured in the field; report species and age where possible. Describe how animals were caught and transported and what happened to captive animals after the study (if killed, explain why and describe method; if released, say where and when) OR state that the study did not involve wild animals.* |
| Reporting on sex | *Indicate if findings apply to only one sex; describe whether sex was considered in study design, methods used for assigning sex. Provide data disaggregated for sex where this information has been collected in the source data as appropriate; provide overall numbers in this Reporting Summary. Please state if this information has not been collected. Report sex-based analyses where performed, justify reasons for lack of sex-based analysis.* |
| Field-collected samples | *For laboratory work with field-collected samples, describe all relevant parameters such as housing, maintenance, temperature, photoperiod and end-of-experiment protocol OR state that the study did not involve samples collected from the field.* |
| Ethics oversight | *Identify the organization(s) that approved or provided guidance on the study protocol, OR state that no ethical approval or guidance was required and explain why not.* |

Note that full information on the approval of the study protocol must also be provided in the manuscript.

# Clinical data

Policy information about clinical studies
All manuscripts should comply with the ICMJE guidelines for publication of clinical research and a completed CONSORT checklist must be included with all submissions.

| | |
|---|---|
| Clinical trial registration | *Provide the trial registration number from ClinicalTrials.gov or an equivalent agency.* |
| Study protocol | *Note where the full trial protocol can be accessed OR if not available, explain why.* |
| Data collection | *Describe the settings and locales of data collection, noting the time periods of recruitment and data collection.* |

| Outcomes | *Describe how you pre-defined primary and secondary outcome measures and how you assessed these measures.* |
|---|---|

# Dual use research of concern

Policy information about dual use research of concern

### Hazards

Could the accidental, deliberate or reckless misuse of agents or technologies generated in the work, or the application of information presented in the manuscript, pose a threat to:

| No | Yes | |
|----|-----|---|
| ☒ | ☐ | Public health |
| ☒ | ☐ | National security |
| ☒ | ☐ | Crops and/or livestock |
| ☒ | ☐ | Ecosystems |
| ☒ | ☐ | Any other significant area |

### Experiments of concern

Does the work involve any of these experiments of concern:

| No | Yes | |
|----|-----|---|
| ☒ | ☐ | Demonstrate how to render a vaccine ineffective |
| ☒ | ☐ | Confer resistance to therapeutically useful antibiotics or antiviral agents |
| ☒ | ☐ | Enhance the virulence of a pathogen or render a nonpathogen virulent |
| ☒ | ☐ | Increase transmissibility of a pathogen |
| ☒ | ☐ | Alter the host range of a pathogen |
| ☒ | ☐ | Enable evasion of diagnostic/detection modalities |
| ☒ | ☐ | Enable the weaponization of a biological agent or toxin |
| ☒ | ☐ | Any other potentially harmful combination of experiments and agents |

# Plants

| Seed stocks | *Report on the source of all seed stocks or other plant material used. If applicable, state the seed stock centre and catalogue number. If plant specimens were collected from the field, describe the collection location, date and sampling procedures.* |
|---|---|
| Novel plant genotypes | *Describe the methods by which all novel plant genotypes were produced. This includes those generated by transgenic approaches, gene editing, chemical/radiation-based mutagenesis and hybridization. For transgenic lines, describe the transformation method, the number of independent lines analyzed and the generation upon which experiments were performed. For gene-edited lines, describe the editor used, the endogenous sequence targeted for editing, the targeting guide RNA sequence (if applicable) and how the editor was applied.* |
| Authentication | *Describe any authentication procedures for each seed stock used or novel genotype generated. Describe any experiments used to assess the effect of a mutation and, where applicable, how potential secondary effects (e.g. second site T-DNA insertions, mosiacism, off-target gene editing) were examined.* |

# ChIP-seq

### Data deposition

☐ Confirm that both raw and final processed data have been deposited in a public database such as GEO.

☐ Confirm that you have deposited or provided access to graph files (e.g. BED files) for the called peaks.

| Data access links | *For "Initial submission" or "Revised version" documents, provide reviewer access links. For your "Final submission" document, provide a link to the deposited data.* |
|---|---|
| *May remain private before publication.* | |
| Files in database submission | *Provide a list of all files available in the database submission.* |
| Genome browser session | *Provide a link to an anonymized genome browser session for "Initial submission" and "Revised version" documents only, to enable peer review. Write "no longer applicable" for "Final submission" documents.* |
| (e.g. UCSC) | |

## Methodology

| | |
|---|---|
| Replicates | *Describe the experimental replicates, specifying number, type and replicate agreement.* |
| Sequencing depth | *Describe the sequencing depth for each experiment, providing the total number of reads, uniquely mapped reads, length of reads and whether they were paired- or single-end.* |
| Antibodies | *Describe the antibodies used for the ChIP-seq experiments; as applicable, provide supplier name, catalog number, clone name, and lot number.* |
| Peak calling parameters | *Specify the command line program and parameters used for read mapping and peak calling, including the ChIP, control and index files used.* |
| Data quality | *Describe the methods used to ensure data quality in full detail, including how many peaks are at FDR 5% and above 5-fold enrichment.* |
| Software | *Describe the software used to collect and analyze the ChIP-seq data. For custom code that has been deposited into a community repository, provide accession details.* |

# Flow Cytometry

## Plots

Confirm that:

☐ The axis labels state the marker and fluorochrome used (e.g. CD4-FITC).

☐ The axis scales are clearly visible. Include numbers along axes only for bottom left plot of group (a 'group' is an analysis of identical markers).

☐ All plots are contour plots with outliers or pseudocolor plots.

☐ A numerical value for number of cells or percentage (with statistics) is provided.

## Methodology

| | |
|---|---|
| Sample preparation | *Describe the sample preparation, detailing the biological source of the cells and any tissue processing steps used.* |
| Instrument | *Identify the instrument used for data collection, specifying make and model number.* |
| Software | *Describe the software used to collect and analyze the flow cytometry data. For custom code that has been deposited into a community repository, provide accession details.* |
| Cell population abundance | *Describe the abundance of the relevant cell populations within post-sort fractions, providing details on the purity of the samples and how it was determined.* |
| Gating strategy | *Describe the gating strategy used for all relevant experiments, specifying the preliminary FSC/SSC gates of the starting cell population, indicating where boundaries between "positive" and "negative" staining cell populations are defined.* |

☐ Tick this box to confirm that a figure exemplifying the gating strategy is provided in the Supplementary Information.

# Magnetic resonance imaging

## Experimental design

| | |
|---|---|
| Design type | *Indicate task or resting state; event-related or block design.* |
| Design specifications | *Specify the number of blocks, trials or experimental units per session and/or subject, and specify the length of each trial or block (if trials are blocked) and interval between trials.* |
| Behavioral performance measures | *State number and/or type of variables recorded (e.g. correct button press, response time) and what statistics were used to establish that the subjects were performing the task as expected (e.g. mean, range, and/or standard deviation across subjects).* |

## Acquisition

| | |
|---|---|
| Imaging type(s) | *Specify: functional, structural, diffusion, perfusion.* |
| Field strength | *Specify in Tesla* |
| Sequence & imaging parameters | *Specify the pulse sequence type (gradient echo, spin echo, etc.), imaging type (EPI, spiral, etc.), field of view, matrix size, slice thickness, orientation and TE/TR/flip angle.* |
| Area of acquisition | *State whether a whole brain scan was used OR define the area of acquisition, describing how the region was determined.* |

Diffusion MRI ☐ Used ☐ Not used

## Preprocessing

| | |
|---|---|
| Preprocessing software | *Provide detail on software version and revision number and on specific parameters (model/functions, brain extraction, segmentation, smoothing kernel size, etc.).* |
| Normalization | *If data were normalized/standardized, describe the approach(es): specify linear or non-linear and define image types used for transformation OR indicate that data were not normalized and explain rationale for lack of normalization.* |
| Normalization template | *Describe the template used for normalization/transformation, specifying subject space or group standardized space (e.g. original Talairach, MNI305, ICBM152) OR indicate that the data were not normalized.* |
| Noise and artifact removal | *Describe your procedure(s) for artifact and structured noise removal, specifying motion parameters, tissue signals and physiological signals (heart rate, respiration).* |
| Volume censoring | *Define your software and/or method and criteria for volume censoring, and state the extent of such censoring.* |

## Statistical modeling & inference

| | |
|---|---|
| Model type and settings | *Specify type (mass univariate, multivariate, RSA, predictive, etc.) and describe essential details of the model at the first and second levels (e.g. fixed, random or mixed effects; drift or auto-correlation).* |
| Effect(s) tested | *Define precise effect in terms of the task or stimulus conditions instead of psychological concepts and indicate whether ANOVA or factorial designs were used.* |

Specify type of analysis: ☐ Whole brain ☐ ROI-based ☐ Both

| | |
|---|---|
| Statistic type for inference<br><br>(See Eklund et al. 2016) | *Specify voxel-wise or cluster-wise and report all relevant parameters for cluster-wise methods.* |
| Correction | *Describe the type of correction and how it is obtained for multiple comparisons (e.g. FWE, FDR, permutation or Monte Carlo).* |

## Models & analysis

| n/a | Involved in the study | |
|---|---|---|
| ☐ | ☐ | Functional and/or effective connectivity |
| ☐ | ☐ | Graph analysis |
| ☐ | ☐ | Multivariate modeling or predictive analysis |

| | |
|---|---|
| Functional and/or effective connectivity | *Report the measures of dependence used and the model details (e.g. Pearson correlation, partial correlation, mutual information).* |
| Graph analysis | *Report the dependent variable and connectivity measure, specifying weighted graph or binarized graph, subject- or group-level, and the global and/or node summaries used (e.g. clustering coefficient, efficiency, etc.).* |
| Multivariate modeling and predictive analysis | *Specify independent variables, features extraction and dimension reduction, model, training and evaluation metrics.* |

