## [Peer Review File · Nature Cell Biology]

Visualizing sub-organellar lipid distribution using correlative light and electron microscopy

Corresponding Author: Dr André Nadler

Version 0:

Decision Letter:

*Please delete the link to your author homepage if you wish to forward this email to co-authors.

Dear Dr. Nadler,

Your manuscript, "Visualizing sub-organellar lipid distribution using correlative light and electron microscopy", has now been seen by 3 referees, who are experts in membrane trafficking and lipid dynamics (referee 1); super-resolution imaging and membrane trafficking (referee 2); and super-resolution microscopy and ER dynamics (referee 3). As you will see from their comments (attached below) they find this work of potential interest, but have raised substantial concerns, which in our view would need to be addressed with considerable revisions before we can consider publication in Nature Cell Biology.

Nature Cell Biology editors discuss the referee reports in detail within the editorial team, including the chief editor, to identify key referee points that should be addressed with priority, and requests that are overruled as being beyond the scope of the current study. To guide the scope of the revisions, I have listed these points below. We are committed to providing a fair and constructive peer-review process, so please feel free to contact me if you would like to discuss any of the referee comments further.

In particular, it would be essential to:

(A) Evaluate whether the use of this approach leads to any artifacts in terms of membrane remodeling events, as mentioned by Reviewers #1 and #3. At the same time, please evaluate how loss of resolution with fluorescence imaging may impact signal quantification (per Reviewer #1) and the consistency of fluorescence labeling throughout sections (per Reviewer #3).

(B) Please validate results using immunogold labeling (per Reviewer #1) and follow Reviewer #3's suggestions to better outline the minimal amount of UV exposure required for cross-linking (or the benefits of using one resin over another).

(C) We agree with Reviewers #2-3 that the approach would benefit from evaluating its use in other lipid species, or (per major comment #1 from Reviewer #2) how specific lipid domains drive machinery or cargo segregation in endosomes.

(D) From an editorial perspective, it would be helpful to explicitly discuss the distinctions between Lipid-CLEM and the approach outlined in reference #20, in terms of any differences in experimental setup, and when a researcher should use one approach over another. Please include a detailed explanation in both a new cover letter and the main text of the revised manuscript.

(E) All other referee concerns pertaining to strengthening existing data, providing controls, methodological details, clarifications and textual changes, should also be addressed.

(F) Finally please pay close attention to our guidelines on statistical and methodological reporting (listed below) as failure to do so may delay the reconsideration of the revised manuscript. In particular please provide:

We would be happy to consider a revised manuscript that would satisfactorily address these points, unless a similar paper is published elsewhere, or is accepted for publication in Nature Cell Biology in the meantime.

- ensure that it conforms to our format instructions and publication policies (see below and <https://www.nature.com/nature/for-authors>).
- provide a point-by-point rebuttal to the full referee reports verbatim, as provided at the end of this letter.
- provide the completed Reporting Summary (found here <https://www.nature.com/documents/nr-reporting-summary.pdf>). This is essential for reconsideration of the manuscript will be available to editors and referees in the event of peer review. For more information see <http://www.nature.com/authors/policies/availability.html> or contact me.

Nature Cell Biology is committed to improving transparency in authorship. As part of our efforts in this direction, we are now requesting that all authors identified as 'corresponding author' on published papers create and link their Open Researcher and Contributor Identifier (ORCID) with their account on the Manuscript Tracking System (MTS), prior to acceptance. ORCID helps the scientific community achieve unambiguous attribution of all scholarly contributions. You can create and link your ORCID from the home page of the MTS by clicking on 'Modify my Springer Nature account'. For more information please visit www.springernature.com/orcid.

This journal strongly supports public availability of data. Please place the data used in your paper into a public data repository, or alternatively, present the data as Supplementary Information. If data can only be shared on request, please explain why in your Data Availability Statement, and also in the correspondence with your editor. Please note that for some data types, deposition in a public repository is mandatory - more information on our data deposition policies and available repositories appears below.

Link Redacted

We would like to receive a revised submission within six months.

We hope that you will find our referees' comments, and editorial guidance helpful. Please do not hesitate to contact me if there is anything you would like to discuss.

Best regards,

George

George Inglis, PhD

Senior Editor

[Research Cross-Journal Editorial Team](https://www.nature.com/ncb/research-cross-journal-editorial-team)
Nature Cell Biology

Reviewers' Comments:

Reviewer #1 (Remarks to the Author):

Lennartz and colleagues present a unique tool for visualizing lipids using CLEM. Using synthesized lipids with click chemistry groups, the manuscript describes the workflow for labeling and identifying lipids using fluorescent labels by photonic microscopy, and colocalizing to specific compartments using EM. Therefore, the authors show that combining click chemistry with electron microscopy provides a useful method for probing the subcellular localization of lipid species and quantifying their amount. The authors show for example that sphingomyelin is enriched in intra-lumenal vesicles, which is an interesting and new result and on which the authors should build a complete study. The work is overall of very high quality, nevertheless, I have some concerns about the novelty and the appropriateness for NCB.

The paper as such is mostly a method paper, and while the last results on sphingomyelin enrichment in ILVs is very promising, it would be important to develop a full and complete study with this. Thus, unless NCB offers a method section for this kind of paper, I think the work, because of its high quality, is most appropriated for Nature Methods.

While being of very high quality, there are some questions and additional experiments that can validate and strengthen the method.

Lipids and lipid precursors with clickable handles have been used before to probe localization via fluorescence microscopy. For example, Jao et al., 2009, PNAS published the metabolic incorporation of propargylcholine into cells and used clickable fluorophores. The authors should validate, like in the mentioned reference, whether their lipid loading experiments show an incorporation into various lipid species. TLC and/or MS should be used to determine the degree of incorporation of the loaded lipids.

Fluorescence microscopy inherently has a resolution limit and is a challenge when coupled to EM. In particular, it is very clear from the overlay of fluorescence images and EM images that several sub-compartments, distinct in the EM image, are merged in fluorescence. Also, most probably because the fluorescence image is "thicker" than the EM image, some parts visible on the fluorescence image are just not visible in the EM. While I understand that these dimensional and resolution problems affect the quantification that the authors are performing.

Authors should validate their claims about domain partitioning and sub-organellar localization using immuno-gold labeling. With the clickable handle, biotin-azide can be conjugated and probed with streptavidin-Au particles, as done by Jao et al. Immunogold labeling should be compatible with the resin. Authors should distinguish the advantage of using CLEM as opposed to immunoGold EM. Both techniques rely on immunocytochemistry, but immunogold may provide a more precise localization and all imaging can be done on a standard TEM. Authors should complement CLEM with immunogold to properly distinguish sub-compartments that are below the diffraction limit of optical microscopy.

For the applicability of the technique beyond PC lipids, the alpha-methyl-cyclodextrin-mediated lipid exchange between liposomes and the plasma membrane introduce concerns. If one was to probe a different lipid, for example one with high intrinsic curvature, how can one be sure that the lipid exchange with the outer leaflet of the PM will not induce artificial membrane remodeling events? In addition, regarding lipid abundance, how can authors be sure that localization reflects the native lipid localization as opposed to slow (or fast) trafficking from the plasma membrane? Lipids, like PA or SM, are signaling lipids and can inadvertently recruit proteins that may stabilize the lipid in wrong compartments. Can authors validate whether the localization of protein-based lipid probes, for example for PC, changes upon addition of exogenous lipid?

Reviewer #2 (Remarks to the Author):

Lennartz et al., report on a Lipid-CLEM approach which enables investigation of the lipid composition on cellular organelles at high resolution. The article is comprehensible and highlights the advantage of their novel approach to existing methods. Lastly, using their Lipid-CLEM approach they could show segregation of different cargo in EE, in line with the published literature.

The authors have very comprehensively described their Lipid-CLEM approach and its clear advantages over established methods. Overall, I believe their described method offers a valuable approach for studying lipid composition, which has remained challenging. The authors undoubtedly fill an important gap in the visualization of lipid domains in cells.

Major comment:

1. The existence of different nanodomains on EEs that would sort distinct cargo has been previously reported, though not including the visualization of lipids at such high resolution. Defining endosomes with specific markers is challenging as they may harbour different sorting machinery with different function. To strengthen the paper, the authors could look at machinery such as Rab5/EEA1/Rab4 and try to understand whether different lipid domains drive machinery/cargoes segregation on endosomes.
2. In the discussion the authors mention "protein and lipid transport routes diverge at the early endosome as sphingomyelin and transferrin arrive simultaneously in clathrin-coated vesicles, but partition differentially into endosomal compartments". Here, adding clathrin into the picture would help strengthen the authors conclusions. Furthermore, how can it be determined that the tubular compartment adjacent to the EEs is derived from/part of the compartment? Are these tubular compartments devoid of clathrin?

Minor comments:

3. Please clarify the description and Figure panels in Extended Data Figure 6:
 - i. "A mix of Rab5 and EEA1 was used to mark early endosomes" – are the investigated compartments are positive for both or either of the markers?
 - ii. In 'A', the separation of the cargo-endosome time graphs is misleading. The yellow curve (Rab5/EEA1) shows different

percentages in the left vs. right graph. I assume this results from the normalization of “total endosomes” - on the left graph, maybe EE + LE, and on the right, EE + RE? I disagree with the wording “total endosomes”, as to accurately represent the total number of endosomes, additional markers would be needed. Furthermore, the Rab11 curve for Tfn recycling appears unusually low, almost comparable to the Rab7 curve which disagrees with other studies (<https://doi.org/10.1083/jcb.149.4.901>). I recommend merging the graphs to avoid this confusion.

iii. In 'B', Rab11-positive REs appear predominantly localized around the perinuclear area, which is surprising given that REs have also been reported at the cell periphery. Additionally, most of the REs in the image appear round and less tubular compared to what is typically described in the literature. Could this be due to an unrepresentative image, the antibody used, or issues with structure preservation during fixation?

Reviewer #3 (Remarks to the Author):

The study introduces a novel Lipid-CLEM workflow that integrates bifunctional lipid probes, rapid photo-crosslinking, high-pressure freezing, and on-section click chemistry to achieve nanoscale visualization of lipid distribution. This method quantitatively demonstrates that sphingomyelin partitions differentially within early endosomal compartments, being enriched in intraluminal vesicles and depleted in recycling tubules. These findings offer new insights into membrane organization and the mechanisms underlying lipid sorting.

The study employs an innovative approach by developing the Lipid-CLEM workflow, which is technically impressive and successfully overcomes long-standing challenges in high-resolution lipid imaging. The dual approach, utilizing both surface labeling with HM20 resin and whole-section labeling with K11M resin, is particularly well conceived. Integrating light and electron microscopy enables detailed nanoscale quantification of lipid distribution, and the quantitative analysis of sphingomyelin partitioning in early endosomes is both compelling and adds significant value to the study. Furthermore, the methodology is executed with high rigor, as demonstrated by the detailed description of rapid fixation and on-section click chemistry, offering a reproducible framework for future research. Finally, the biological relevance of the work is clear, as the application of the method to study lipid sorting in early endosomes addresses a critical question in membrane biology.

This reviewer finds the study suitable for publication. Consideration of the feedback below would strengthen the study.

FEEDBACK:

- **Crosslinking Specificity & Efficiency:** Although Extended Data 3 includes negative controls that were not treated with UV light or PC(16:0Y), confirming that fluorescence requires both UV irradiation and the probe, it remains important to further quantify the crosslinking specificity and efficiency. Incorporating a time-course experiment to systematically vary UV exposure would help determine the minimum irradiation time needed for effective crosslinking and provide a clearer kinetic profile of the reaction. This additional quantification would solidify that the fluorescence signal is indeed a specific result of the photo-crosslinking process and not due to non-specific interactions.

- **Resin-Dependent Artifacts:** The manuscript discusses differences between HM20 and K11M resins—with Extended Data 2 and Extended Data 7 showing that HM20 restricts labeling to the section surface while K11M permits more uniform staining across the section. A direct quantitative comparison of lipid partitioning outcomes between these two resin types is necessary to ensure that the observed differences in sphingomyelin distribution (i.e., enrichment in intraluminal vesicles versus depletion in recycling tubules) are intrinsic to the biological phenomenon and not artifacts introduced by the embedding process. This direct comparison would reinforce the overall conclusions by confirming that resin-dependent variability does not confound the interpretation of lipid partitioning.

OTHER CONSIDERATIONS:

- **Probe Behavior vs. Native Lipids:** While Extended Data 1 provides a comparison between NBD lipids and the bifunctional probes, additional validation is needed to ensure that the minimally modified bifunctional probes truly recapitulate native lipid behavior. Employing an orthogonal method—such as mass spectrometry or an alternative imaging approach—could confirm that the probes do not introduce subtle artifacts affecting lipid distribution. This step is critical because the reliability of the method hinges on the probes mimicking native lipid properties as closely as possible.

- **Uniformity of Fluorescence Labeling:** Although Extended Data 7 demonstrates that HM20 resin limits labeling to the section surface and K11M resin offers deeper penetration, further quantitative analysis of labeling uniformity across the entire section is recommended. Ensuring that fluorescence labeling is consistent throughout the section is essential for accurate lipid density measurements. Variability in the labeling intensity could confound the interpretation of lipid partitioning, particularly when making quantitative comparisons across different membrane subdomains.

- **Generalizability:** The study currently focuses on sphingomyelin distribution in early endosomes within U2OS cells. To establish the broader applicability of the Lipid-CLEM workflow, extending the analysis to additional lipid species or other cell types would be beneficial. Demonstrating that the method works reliably across various biological contexts would not only reinforce its robustness but also highlight its versatility as a tool for investigating lipid organization in diverse cellular environments.

Overall, the study represents a significant advancement in lipid imaging technology. Addressing the critical points above—especially regarding controls for crosslinking specificity, reproducibility, resin artifacts, and statistical robustness—will substantially strengthen the manuscript. Regardless, this reviewer supports moving the study forward for publication.

Methods should be written concisely, but should contain all elements necessary to allow interpretation and replication of the results. As a guideline, Methods sections typically do not exceed 3,000 words. The Methods should be divided into subsections listing reagents and techniques. When citing previous methods, accurate references should be provided and any alterations should be noted. Information must be provided about: antibody dilutions, company names, catalogue numbers and clone numbers for monoclonal antibodies; sequences of RNAi and cDNA probes/primers or company names and catalogue numbers if reagents are commercial; cell line names, sources and information on cell line identity and authentication. Animal studies and experiments involving human subjects must be reported in detail, identifying the committees approving the protocols. For studies involving human subjects/samples, a statement must be included confirming that informed consent was obtained. Statistical analyses and information on the reproducibility of experimental results should be provided in a section titled "Statistics and Reproducibility".

All Nature Cell Biology manuscripts submitted on or after March 21 2016 must include a Data availability statement as a separate section after Methods but before references, under the heading "Data Availability". For Springer Nature policies on data availability see <http://www.nature.com/authors/policies/availability.html>; for more information on this particular policy see <http://www.nature.com/authors/policies/data/data-availability-statements-data-citations.pdf>. The Data availability statement should include:

- Accession codes for primary datasets (generated during the study under consideration and designated as "primary accessions") and secondary datasets (published datasets reanalysed during the study under consideration, designated as "referenced accessions"). For primary accessions data should be made public to coincide with publication of the manuscript. A list of data types for which submission to community-endorsed public repositories is mandated (including sequence, structure, microarray, deep sequencing data) can be found here <http://www.nature.com/authors/policies/availability.html#data>.
- Unique identifiers (accession codes, DOIs or other unique persistent identifier) and hyperlinks for datasets deposited in an approved repository, but for which data deposition is not mandated (see here for details <http://www.nature.com/sdata/data-policies/repositories>).
- At a minimum, please include a statement confirming that all relevant data are available from the authors, and/or are included with the manuscript (e.g. as source data or supplementary information), listing which data are included (e.g. by figure panels and data types) and mentioning any restrictions on availability.
- If a dataset has a Digital Object Identifier (DOI) as its unique identifier, we strongly encourage including this in the Reference list and citing the dataset in the Methods.

We recommend that you upload the step-by-step protocols used in this manuscript to protocols.io. More details can be found at <https://www.protocols.io/help/publish-articles>.

All imaging data should be accompanied by scale bars, which should be defined in the legend. Cropped images of gels/blots are acceptable, but need to be accompanied by size markers, and to retain visible background signal within the linear range (i.e. should not be saturated). The boundaries of panels with low background have to be demarked with black lines. Splicing of panels should only be considered if unavoidable, and must be clearly marked on the figure, and noted in the legend with a statement on whether the samples were obtained and processed simultaneously. Quantitative comparisons between samples on different gels/blots are discouraged; if this is unavoidable, it should only be performed for samples derived from the same experiment with gels/blots were processed in parallel, which needs to be stated in the legend.

The total number of Supplementary Figures (not including the “unprocessed scans” Supplementary Figure) should not exceed the number of main display items (figures and/or tables (see our Guide to Authors and March 2012 editorial <http://www.nature.com/ncb/authors/submit/index.html#supinfo>; <http://www.nature.com/ncb/journal/v14/n3/index.html#ed>). No restrictions apply to Supplementary Tables or Videos, but we advise authors to be selective in including supplemental data.

GUIDELINES FOR EXPERIMENTAL AND STATISTICAL REPORTING

REPORTING REQUIREMENTS – We are trying to improve the quality of methods and statistics reporting in our papers. To that end, we are now asking authors to complete a reporting summary that collects information on experimental design and reagents. The Reporting Summary can be found here <https://www.nature.com/documents/nr-reporting-summary.pdf>. If you would like to reference the guidance text as you complete the template, please access these flattened versions at <http://www.nature.com/authors/policies/availability.html>.

Version 1:

Decision Letter:

Our ref: NCB-TR57011A

13th January 2026

Dear Dr. Nadler,

Thank you for submitting your revised manuscript "Visualizing sub-organellar lipid distribution using correlative light and electron microscopy" (NCB-TR57011A). I apologize again sincerely for the delay in sharing our decision with you. The revision has now been seen by the original referees and their comments are below. The reviewers find that the paper has improved in revision, and therefore we'll be happy in principle to publish it in Nature Cell Biology, pending minor revisions to satisfy the referees' final requests and to comply with our editorial and formatting guidelines.

****Please note that the current version of your manuscript is in a PDF format; could you please email us a copy of the file in an editable format (Microsoft Word) as we cannot proceed with PDFs at this stage? Thank you in advance.****

Once we have the Word file, we will begin performing detailed checks on your paper and will send you a checklist detailing our editorial and formatting requirements in about 1-2 weeks. Please do not upload the final materials and make any revisions until you receive this additional information from us.

Thank you again for your interest in Nature Cell Biology. Please do not hesitate to contact me if you have any questions.

Sincerely,

Melina Casadio, PhD
Senior Editor, Nature Cell Biology
ORCID ID: <https://orcid.org/0000-0003-2389-2243>

Reviewer #1 (Remarks to the Author):

Th authors have done a fantastic work to answer my concerns. I agree with the publication strategy using the the technical report option of NCB and that SM partitioning could be studied in later studies. All other points are rigorously clarified in the rebuttal by appropriate references to the previous work of authors, or by additional experiments and information. I therefore now fully support publication at NCB of this manuscript.

Reviewer #2 (Remarks to the Author):

The authors have addressed all my concerns and I am happy to support publication and I only have a small comment. This is just a suggestion.

I appreciate the inclusion of Extended Data 7, however The title is misleading as SM is simply rather homogeneously distributed over the PM. Maybe a good alternative could be "SM is not enriched in TfR positive clathrin coated pits at the PM"?

Reviewer #2 (Remarks on Protocol(s)):

The methods are described in detail

Reviewer #3 (Remarks to the Author):

This reviewer is satisfied that the authors have adequately addressed all concerns raised in the initial review.

The new data comparing HM20 and K11M resins (Supplementary Figure 9) demonstrates that the key biological findings are conserved across both embedding approaches, which provides confidence that the observations reflect biology rather than technical artifacts. The expanded Figure 2 with quantitative analysis of labeling uniformity and the new Figure 4 showing application across four distinct lipid probes address the concerns about reproducibility and generalizability. The referenced characterization data for probe behavior and crosslinking efficiency are appropriate.

The revised manuscript represents a significant methodological advance for visualizing lipid distribution at the ultrastructural level. This reviewer recommends acceptance for publication.

Version 2:

Decision Letter:

Dear Dr. Nadler,

I am writing on behalf of my colleague, Dr. Melina Casadio, who is out of the office.

I am pleased to inform you that your manuscript, "Visualizing sub-organellar lipid distribution using correlative light and electron microscopy", has now been accepted for publication in *Nature Cell Biology*.

Over the next few weeks, your paper will be copyedited to ensure that it conforms to *Nature Cell Biology* style. Once your paper is typeset, you will receive an email with a link to choose the appropriate publishing options for your paper and our Author Services team will be in touch regarding any additional information that may be required.

Publication is conditional on the manuscript not being published elsewhere and on there being no announcement of this work to any media outlet until the online publication date in *Nature Cell Biology*.

Please note that *Nature Cell Biology* is a Transformative Journal (TJ). Authors may publish their research with us through the traditional subscription access route or make their paper immediately open access through payment of an article-processing charge (APC). Authors will not be required to make a final decision about access to their article until it has been accepted. <https://www.springernature.com/gp/open-research/transformative-journals> Find out more about Transformative Journals

Authors may need to take specific actions to achieve compliance with funder and institutional open access mandates. If your research is supported by a funder that requires immediate open access (e.g. according to [Plan S principles](https://www.springernature.com/gp/open-science/plan-s-compliance) or the [NIH public access policy](https://www.springernature.com/gp/open-science/us-federal-agency-compliance)) then you should select the gold OA route, and we will direct you to the compliant route where possible. Because authors warrant under our subscription licensing terms that they haven't committed to licensing any version of their article under a licence inconsistent with the terms of our agreement – including the applicable embargo period – publication under the subscription model isn't suitable for authors whose funders require no embargo.

If you have not already done so, we strongly recommend that you upload the step-by-step protocols used in this manuscript to protocols.io (<https://protocols.io>), an open online resource that allows researchers to share their detailed experimental know-how. All uploaded protocols are made freely available and are assigned DOIs for ease of citation. Protocols and Nature Portfolio journal papers in which they are used can be linked to one another, and this link is clearly and prominently visible in the online versions of both. Authors who performed the specific experiments can act as primary authors for the Protocol as they will be best placed to share the methodology details, but the Corresponding Author of the present research paper should be included as one of the authors. By uploading your Protocols onto protocols.io, you are enabling researchers to more readily reproduce or adapt the methodology you use, as well as increasing the visibility of your protocols and papers. You can also establish a dedicated workspace to collect your lab Protocols. Further information can be found at <https://www.protocols.io/help/publish-articles>.

Nature Cell Biology encourages authors presenting evidence for cell, biological, molecular, and genetic interactions to consider communicating these findings using Biofactoid (<https://biofactoid.org/>). This tool helps users share a searchable representation of interactions (e.g. binding, gene expression, post-translational modification) between genes, gene products, or chemicals. Information added to Biofactoid, with author attribution, is shared on social media and public databases, such as Pathway Commons, where it can be discovered and analyzed in the context of a large and growing corpus of knowledge.

All the best,

Christina

Christina Kary, PhD
Chief Editor
Nature Cell Biology
1 New York Plaza

** Visit the Springer Nature Editorial and Publishing website at http://editorial-jobs.springernature.com?utm_source=ejp_NCB_email&utm_medium=ejp_NCB_email&utm_campaign=ejp_NCB for more information about our career opportunities. If you have any questions please click [here](mailto:editorial.publishing.jobs@springernature.com).

Visualizing sub-organellar lipid distribution using correlative light and electron microscopy

Corresponding Authors: Dr. André Nadler, Dr. H. Mathilda Lennartz

General remarks:

We thank all reviewers for their comments, which provided helpful and constructive feedback. We believe that the revisions made the manuscript substantially stronger.

We have addressed all points experimentally that were feasible in the time frame of the revision. We have acquired extensive new data, which are shown in 1 new main text figure, 2 significantly revised and expanded main text figures, 3 new and 1 significantly revised Extended Data figure, and 4 new Supplementary Information figures. The main improvements of the manuscript are:

- We demonstrate the generalizability of our Lipid-CLEM approach by applying the method on 4 different lipid probes in total including different lipid classes and species.
- We include new datasets that strengthen our finding of lipid sorting into membrane domains at the early endosome.
- We now include nanogold labeling of bifunctional lipids as an alternative approach and define use cases for Lipid-CLEM and nanogold labeling.
- We performed extensive control experiments to exclude the possibility of membrane remodeling artefacts and resin depended artifacts.

The following section outlines the structure of this document and the revised version, and gives an overview of the restructuring of the original manuscript version, which was required to accommodate the new datasets.

1. Full reviewer comments are provided together with our point-by-point answers to maintain the context. Key results are included here as revised / new figures, and the primary conclusion from a new experiment/dataset is highlighted. Our answers to individual reviewer comments are marked by “**Answer**” and are highlighted in **blue** to enhance clarity.

2. We have added a new main text Figure (Figure 4) to include additional lipid probes in the Lipid-CLEM workflow, highlighting the generality of the method and the differences in nanoscale localization between lipids. Original Figure 4 is now Figure 5.

3. The following main text figure have been revised to include new datasets or datasets which were previously displayed in the supporting information: Figure 2. This figure has been significantly expanded and features new datasets highlighting the labeling differences between the resins used here (HM20 and K11M). Figure 3. This Figure has been significantly expanded and features a new dataset on labeling lipids using an immunogold labeling approach.

4. The following ED figures have been revised: ED6 (now ED5)

5. The following new figures have been added to the Extended Data:

ED2: The effect of lipid loading on membrane remodeling events.

ED3: The effect of lipid loading on cellular signaling events.

ED7: Co-uptake of SM(Y) with transferrin via clathrin mediated endocytosis

6. The following new figures have been added to the Supplementary Information:

SI4: Rab11 Antibody verification.

SI5: Exemplary Correlation Error Maps for Lipid-CLEM of various additional lipids (supporting the new main text Figure 4).

SI6: Connectivity of recycling tubule to the boundary membrane of endosomes.

SI9: Lipid-CLEM of SM(Y) in HM20, to determine lipid “densities” in 1D.

7. Supplementary movies 1 and 2 are now included in the dataset to visualize assessment of membrane connectivity in tomograms.

8. All textual changes in the revised manuscript have been highlighted in yellow.

Answers to specific reviewer questions:

Referee #1 (Remarks to the Author)

Lennartz and colleagues present a unique tool for visualizing lipids using CLEM. Using synthesized lipids with click chemistry groups, the manuscript describes the workflow for labeling and identifying lipids using fluorescent labels by photonic microscopy, and colocalizing to specific compartments using EM. Therefore, the authors show that combining click chemistry with electron microscopy provides a useful method for probing the subcellular localization of lipid species and quantifying their amount. The authors show for example that sphingomyelin is enriched in intraluminal vesicles, which is an interesting and new result and on which the authors should build a complete study. The work is overall of very high quality, nevertheless, I have some concerns about the novelty and the appropriateness for NCB.

The paper as such is mostly a method paper, and while the last results on sphingomyelin enrichment in ILVs is very promising, it would be important to develop a full and complete study with this. Thus, unless NCB offers a method section for this kind of paper, I think the work, because of its high quality, is most appropriated for Nature Methods.

Answer:

We thank Reviewer #1 for the positive overall assessment of our work, particularly regarding its methodological novelty and quality. The answer to the overall comments by reviewer #1 merit a two-part answer:

1. Reviewer #1 correctly notes that the work is method-centric, which is our intention. We submitted the manuscript as a technical report (*Nature Cell Biology's* methods paper format, with the following stated scope: "A Technical Report presents primary research data on a new technique that is likely to be influential") to *Nature Cell Biology* rather than *Nature Methods* for the following reason: The lipid composition of nanoscale membrane domains has long been a core theme in cell biology, with the ongoing discussions of the Raft hypothesis, the lipid composition of ER exit endocytic sites, signaling clusters at the plasma membrane and endosomal membranes just being a few examples. Until now, there has been a severe lack of methodology to characterize the lipid content of membrane nanodomains by microscopy. Our Lipid-CLEM technique now allows to address such questions, at least in a comparative fashion by relative quantification, and we are convinced that it will become an influential approach for membrane biology and lipid cell biology. We think that the readership of *Nature Cell Biology* is the more natural audience for our manuscript compared to *Nature Methods*, as *Nature Cell Biology* is emerging as the go-to journal for top stories in membrane and lipid biology, as demonstrated by a series of very recent high-profile stories in the field and clear editorial commitment (see e.g., "Time for lipid cell biology" *Nat. Cell Biol.* 2025¹).
2. We fully agree that SM partitioning in the membrane trafficking compartment and its trafficking more generally should be investigated in more depth, since the biological reason and the mechanism by which SM partitions remain largely unknown. Correlative light and electron microscopy, as reported here, will play a significant role in this regard. However, we believe that this work is beyond the scope of this method-centric manuscript and should be highlighted appropriately in a separate manuscript. We hope that Reviewer #1 can agree with us here.

While being of very high quality, there are some questions and additional experiments that can validate and strengthen the method. Lipids and lipid precursors with clickable handles have been

used before to probe localization via fluorescence microscopy. For example, Jao et al., 2009, PNAS published the metabolic incorporation of propargylcholine into cells and used clickable fluorophores. The authors should validate, like in the mentioned reference, whether their lipid loading experiments show an incorporation into various lipid species. TLC and/or MS should be used to determine the degree of incorporation of the loaded lipids.

Answer:

This is a very important point. Lipid transport, lipid localization, and lipid metabolism are closely linked, and when one aspect is analyzed, the others should be controlled for. We should have more explicitly stated that we did exactly this in our preceding paper (Quantitative Imaging of lipid transport in mammalian cells; Iglesias-Artola *et al.*, *Nature* **646**, 474–482 (2025); relevant figures: Main text figure 5, Extended Data Figures 7-11), which was available on BioRxiv at the time of submission of this manuscript. Specifically, we extensively assessed the metabolism of bifunctional lipids delivered to the plasma membrane of U2OS cells (the same cell type used for the present manuscript) over a period of 24 hours using ultra-high-resolution mass spectrometry by using the diazirine group as a tracer label. The lipid probes utilized in this revised manuscript [SM(Y), PC(16:0/Y), PE (18:1/Y), PC(Y/20:4)] were all included in the Nature story. The most salient observations were the following:

1. Little metabolic conversion occurs until the four-minute timepoint – which was used for sample preparation in the current manuscript to visualize early steps of retrograde lipid transport and sorting –, indicating that the localization information is indeed reflecting one distinct lipid species.
2. Over the course of 24 hours, we observed significant conversion of all glycerophospholipids probes, primarily by fatty acid remodeling, in contrast to SM, which appeared to be primarily secreted. Primary products of glycerophospholipid metabolism were phosphatidylcholines and neutral lipids, with smaller amounts of phosphatidylethanolamines – a distribution in line with the literature where metabolic labeling experiments using free bifunctional fatty acids were used (Haberkant et al. 2013).
3. Using a comparison with an isotope-labelled PC species, we established that bifunctional lipids are metabolized similarly to native lipids.
4. Lipid transport in general is between 10-60 times faster than metabolism, indicating that the early steps of retrograde lipid transport are not affected by metabolism.

We have adapted the manuscript text as follows:

Lines 51- 55:

We have previously shown that they are metabolized similarly to native lipids, and that the combined effect of the diazirine and alkyne modifications is roughly equivalent to that of one additional double bond, both with regard to biophysical properties and lipid metabolism ². Bifunctional lipids can be introduced into membranes of living cells and rapidly photo-crosslinked to neighboring proteins.

Lines 120-122:

We previously showed that little metabolic interconversion occurs during this 4-minute time interval in U2OS cells for all probes used for CLEM experiments here, indicating that the fluorescence signal is specific for the initially loaded lipid species ².

Below, we provide a compilation of the most relevant/illustrative datasets regarding bifunctional lipid metabolism from Iglesias-Artola et al. in Reviewer Figure 1.

Reviewer Figure 1 | Metabolism of bifunctional lipids. Datasets published in Iglesias-Artola *et al.*, *Nature* 646, 474–482 (2025). **a.** The fraction of initially supplied lipid species as a percentage of all bifunctional lipids in the cell at the 4 min timepoint is >90% for all probes used in the present study. **b.** Overall metabolism of bifunctional lipids. Solid lines indicate mono-exponential fits. SM(Y) data was not fitted as very little interconversion was observed; instead, a linear interpolation is shown. **a.**, **b.**, adapted from Iglesias-Artola *et al.* Error bars: Mean and SD of $n = 3$ independent experiments. **c.**, **d.** Exemplarily shown class- (Extended Data Figure 8d Iglesias-Artola *et al.*) and species-level (Extended Data Figure 9i Iglesias-Artola *et al.*) datasets for bifunctional lipid metabolism over 24 h for two probes used in the present study: [PC(20:4/Y), PE(18:1/Y)]. **e.** Comparison of bulk metabolism between bifunctional lipid probes and isotope labeled lipids indicates that bifunctional fatty acids behave similar to monounsaturated fatty acids in cellular assays (Extended Data Figure 5e, Iglesias-Artola *et al.*). **f.** Comparison of transport and metabolic rate constants shows that lipid transport is at least one order of magnitude faster.

Fluorescence microscopy inherently has a resolution limit and is a challenge when coupled to EM. In particular, it is very clear from the overlay of fluorescence images and EM images that several sub-compartments, distinct in the EM image, are merged in fluorescence. Also, most probably because the fluorescence image is "thicker" than the EM image, some parts visible on the fluorescence image are just not visible in the EM. While I understand that these dimensional and resolution problems affect the quantification that the authors are performing. Authors should validate their claims about domain partitioning and sub-organellar localization using immuno-gold labeling. With the clickable handle, biotin-azide can be conjugated and probed with streptavidin-Au particles, as done by Jao *et al.* Immunogold labeling should be compatible with the resin. Authors should distinguish the advantage of using CLEM as opposed to immunoGold EM. Both techniques rely on immunocytochemistry, but immunogold may provide a more precise localization and all imaging can be done on a standard TEM. Authors should complement CLEM with immunogold to properly distinguish sub-compartments that are below the diffraction limit of optical microscopy.

Answer:

Reviewer #1 raises the point of the inherent resolution limit of fluorescence microscopy and also aspects of sample preparation and labeling techniques. These are important aspects, and we realize that we should have included a more comprehensive description of the experimental approach in the main text and the reasons for choosing a CLEM approach rather than nanogold labeling. We also agree with Reviewer #1 that the paper would be strengthened by a direct comparison of CLEM and nanogold labeling using the same workflow, and we have therefore added nanogold datasets for both resins (HM20 and K11M)

Prior to our detailed answer and the description of our new data, a few clarifications:

1. Sample thickness: Fluorescence and electron microscopy imaging were performed using the same section; there is no loss of information in EM, as we acquired tomograms of the full section depth. For visualization purposes, we used an overlay of a selected plane derived from the tomogram with the fluorescence image (which is the only possible visualization in a 2D figure panel). We realize that this can give the impression of a different dimensionality of fluorescence and EM datasets (which is not the case) and have clarified this in the revised manuscript. Our quantifications are always based on the full 3D reconstruction of the membrane structure, derived from a tomogram of the full section depth in each case, ensuring that fluorescence and EM data correspond to the exact same sample dimensionality. We introduced the following sentence into the main text to clarify this point:

Lines 315-318:

All quantifications of lipid densities were based on high resolution tomograms of the entire section depth to ensure that there would be no difference in dimensionality (“sample thickness”) between fluorescence and electron microscopy data.

2. Lipid-CLEM does not rely on immunocytochemistry; the only required labeling step is the attachment of a small-molecule fluorophore, in contrast to the two-step labeling involving relatively large antibodies and gold particles required for nanogold labeling. This has important implications for labeling density and penetration depth, as discussed below.

As Reviewer #1 pointed out, regular fluorescence microscopy is inherently limited in its resolution, a known problem in all CLEM approaches. The remedy is typically to use spatial information from EM data to analyze the fluorescence data. In our case that means using the reconstructed membrane surface to inform a linear decomposition/ unmixing of the fluorescence signal and thus assigning it to membrane sub-compartments. Ideally, one would of course want to use a molecular label that can be used directly in EM. To enable the lipid density comparisons that we report here using Lipid-CLEM such a label would have to fulfill two criteria:

1. The labeling has to be specific and labeling density has to be sufficient for quantification
2. The label needs to penetrate throughout the section, to allow for actual lipid density quantification (lipid/ membrane surface area).

To assess the potential of nanogold labeling for this purpose, we followed the experimental suggestion of Reviewer #1, and performed nanogold labeling on samples that were generated using the same sample preparation routine that we used for the CLEM samples. This constitutes a divergence from the Jao et al. workflow which relies on thin Tokuyasu section but has the advantage of (a) ensuring direct comparability with the CLEM data and (b) being a much more widely used approach.

Specifically, we prepared bifunctional SM-loaded samples embedded in either HM20 or K11M resin, sectioned them, and subsequently labeled with biotin azide via click chemistry. Gold beads were conjugated to the biotin using an immunostaining approach; antibodies against biotin were conjugated with 10nm Protein A gold beads. A detailed description of the experimental workflow can be found under Immunogold Labeling in the revised Materials and Methods section. We assessed labeling nanogold labeling density by determining the number of gold beads per endosome for two independent samples of both resins. We made the following observations:

1. Labeling was highly specific compared with the negative control (no biotin labeling) for both resins. The labeling density was slightly higher in K11M (average: 11 gold beads per endosome) than in HM20 (average: 6 gold beads per endosome), whereas almost no gold beads were found on unlabeled endosomes. While the labeling is specific, it is also sparse and is not sufficient to derive lipid densities (see new Figure 3). Our labeling density is actually similar to the Jao et al 2009 upon visual inspection, which is on the one hand surprising since labeling with propargylcholine labels broadly labels PC lipids in the cell, whereas we monitor individual PC species. On the other hand, we assume that our workflow leads to better retention of lipids in the sample due to the photochemical crosslinking step, which likely counteracts the concentration effect discussed above.
2. We found that gold bead labeling occurred only on the surface of sections for both HM20 and K11M, which is in line with literature for immunogold labeling 3,4 (see new Figure 3).

These results allow us to conclude that nanogold labeling of individual lipid species is indeed possible – a further methodological development that was prompted by the suggestion of reviewer #1. Similar to Lipid-CLEM using apolar resins such as HM20, it allows to qualitatively assess the presence or absence of distinct lipid species in membrane structures exposed at the section surface, but it is not suitable to derive lipid densities in membrane structures, because whole-section labeling is so-far not possible. An additional consideration is that labeling density is necessarily lower compared to fluorescence – fluorescence labeling requires a single labeling step with a small fluorophore, whereas gold labeling necessitates three subsequent labeling reactions, the latter two with bulky reagents, which physically cannot reach sites in the sample that are accessible to small molecules.

We present these new data in a new Figure 3 and have added the following section to the main text:

Lines 249-260, 264-269:

To assess whether Lipid-CLEM offers unique advantages over more established labeling techniques, we adapted our workflow for imaging single lipid species at high resolution to allow for nanogold labeling, an approach that has previously been used to visualize entire lipid classes by metabolic labeling 5. For immunogold labeling, individual sections containing crosslinked lipids were labelled with biotin-picolyl azide, followed by immunolabeling against biotin and subsequent labeling with 10 nm Protein A gold beads. We found that immunolabeling was highly specific in both K11M and HM20, with barely any background detected (Figure 3). However, labeling was relatively sparse, in line with results 5. On average, we found 11 ± 1 (mean/ sem) gold beads per endosome in K11M samples and 6 ± 1 (mean/ sem) gold beads per endosome in HM20 samples. Tomograms of K11M samples confirmed that gold beads were exclusively localized at the section surface, in line with the expectation for nanogold labeling 3,4. (...) Similarly, nanogold labeling can be used as an alternative technique if there is a requirement to detect a molecular label directly in the EM image, but this will likely come at a lower labeling density due to the multi-step labeling procedure and steric bulk of the reagents, introducing additional experimental constraints. Direct

measurements of lipid densities on 3D membrane structures, on the other hand, are feasible by combining the Lipid-CLEM approach with polar resins such as K11M.

Figure 11 Lipid signal is well preserved and specific in K11M resin. A: U2OS wildtype cells were loaded for 4 min with PC(16:0IY), UV crosslinked, high pressure frozen, embedded in K11M by automated freeze substitution and sectioned into 100 nm thick sections (+Lipid +UV). Negative controls were not UV activated and not loaded with PC(16:0IY) (-Lipid -UV). All cells were co-labelled with fluorescent LDL (blue) and transferrin (magenta). Lipid signal is shown in orange. White dots correspond to multicolored Tetraspecs indicated by arrows. Dotted red lines represent cell borders if not visible otherwise. Scale bar of overview images: 10 μ m; Scale bar of ROIs: 1 μ m. B: The fluorescent overview image (FM), medium magnified TEM image (EM), overlay and correlation error map of 500 nm thick K11M sections are shown left to right in that order. A

representative ROI for an endosome (1) and the plasma membrane (2) are shown, acquired at higher magnification by tomography. One image plane of the tomogram is shown. Scale bar of overview images: 10 μm ; Scale bar of ROIs: 1 μm . Representative images of 3 biological replicates are shown. C/ D: SM(Y) loaded samples were embedded in K11M (C) or HM20 (D) and labelled with goldbeads. Exemplary ROIs of endosomes and the PM are shown for samples labelled with and without biotin, followed by immuno-labeling against biotin and subsequent immunolabeling with 10nm gold beads. Scale bar: 500nm. E: The quantification of gold beads per endosome ($n(\text{endosomes}) \geq 42$). This analysis includes 2 biological replicates. F: Sideview of a tomogram of a 500nm K11M section labelled with goldbeads (10 nm) against SM(Y) is shown. Three exemplary views in plane (xy) are shown, as indicated by the red dotted lines. Scale bar: 200 nm.

For the applicability of the technique beyond PC lipids, the alpha-methyl-cyclodextrin-mediated lipid exchange between liposomes and the plasma membrane introduce concerns. If one was to probe a different lipid, for example one with high intrinsic curvature, how can one be sure that the lipid exchange with the outer leaflet of the PM will not induce artificial membrane remodeling events?

Answer:

Reviewer #1 raises the concern that the incorporation of bifunctional lipids into the outer leaflet of the plasma membrane (PM) mediated by cyclodextrin might induce membrane remodeling events by altering the overall biophysical properties of the PM. This is a valid concern, which is why we decided to test experimentally whether this is the case under the chosen conditions beyond the datasets that we have published previously. Membrane remodeling can in principle occur through two primary compensatory mechanisms: a change in overall lipidome composition and/or through changes in biophysical properties (i.e. viscosity, resting curvature), that should directly affect membrane trafficking events. In this case, the PM would manifest in an alteration of the number of endocytic structures or the overall endocytic rate.

1. To assess the potential effects of lipid loading on the number of endocytic structures and cargo uptake, we performed the following experiment: We loaded U2OS cells with different lipids varying in fatty acid chain length and saturation that differ in their resting curvatures, including the bifunctional SM which is the primary probe used in this study as well as PC species with a wide range of different fatty acid compositions. To determine whether these probes affect endocytic structures and endocytic rates, we quantified transferrin uptake by endocytosis following lipid loading, as well as the number of clathrin-coated and caveolar pits at the PM. As shown in Extended Data 2, transferrin uptake was unchanged under all lipid-loading conditions compared with the control (no lipid). Likewise, we did not detect significant differences in the number of clathrin-coated pits. A modest reduction ($8 \pm 2\%$) in the number of caveolae associated with the PM was observed upon loading with bifunctional SM. This suggests that there could be a small effect on endocytosis via caveolae, however, the magnitude of this effect is below the variance between cells. We conclude that incorporating bifunctional lipids using alpha-methyl-cyclodextrin mediated incorporation into the plasma membrane does not affect membrane trafficking events at the plasma membrane in a cell biologically relevant fashion.

Extended Data 2I Lipid loading does not perturb major endocytic membrane remodeling events. Several bifunctional lipids, including SM(Y) and PCs of different fatty acid composition, were loaded onto the PM of U2OS cells. Transferrin uptake, number of clathrin-coated pits, and caveolae number at the PM were quantified. Three independent replicates are color-coded as indicated. The boxplot indicates the median and interquartile range. The colored dots highlight the mean value of each replicate.

2. In our recent study (Iglesias-Artola et al., Nature 2025), we analyzed the effects of lipid treatment on the composition of the native lipidome. We did not observe any significant short- or long-term changes in response to lipid loading into the plasma membrane, both at the level of lipid classes and lipid species (Iglesias-Artola et al., Nature 2025, Extended Data Figures 1 and 5, exemplarily summarized in Reviewer Figure 2 below). These data indicate that there is no membrane remodeling at the level of lipid metabolism in response to bifunctional lipid incorporation.

a Overall lipidome composition after bifunctional probe loading

b Bifunctional and native lipid classes at 4 min and 24 h

c Bifunctional and native PC species at 4 min and 24 h

Reviewer Figure 2I The native lipidome is unaffected after bifunctional lipid loading. **a.** Comparison of lipidome composition directly (native and bifunctional lipids combined) after lipid loading bifunctional lipid probes (4 min timepoint) with control lipidome. Arrows indicate supplied lipid type. Bars show the mean of 3 biological repeats containing 2 technical replicates each. (Iglesias Artola et al., 2025, ED figure 1c) **b.** Mol% profile acquired at two time points (see inset for color coding) of 23 lipid classes (light bars), of which 7 classes comprise lipids with bifunctional lipid moieties (dark bars) produced from PC Y/20:4 (Iglesias Artola et al., 2025, ED figure 5b). **c.** PC profile covering 22 species with 5 species containing the bifunctional fatty acid. PCs bearing a bifunctional fatty acid (16:1) are annotated as endogenous lipids having the same number of carbons and double bonds in both FA moieties, albeit having different (+28.0061 Da) masses (Iglesias Artola et al., 2025, ED figure 5c).

3. Finally, we would like to point towards our recent preprint:

<https://www.biorxiv.org/content/10.1101/2025.03.04.641423v1.abstract>, where we

show that the lipid composition in clathrin-coated pits is largely in line with bulk plasma membrane lipid composition, and that a wide range of structurally different lipids are taken up into endosomes at a similar rate. This also speaks against the occurrence of lipid-specific membrane remodeling events at the PM after bifunctional lipid incorporation, and supports the conclusion of the present manuscript that lipid and protein cargo routes diverge during retrograde lipid trafficking at the level of the early endosome.

We would like to note that while these data show that there is no observable cellular response in bulk membrane trafficking and overall lipid metabolism, which is expected in the case of major membrane remodeling, we do not claim that lipid incorporation has no effect on cellular biology. Our control experiments show that it is safe to assume that bifunctional lipids used at the

concentration in this study can be understood as “innocent” tracer molecules for studies of intracellular lipid localization, transport, and metabolism. When studying more specialized processes and/ or pathways, a new assessment based on tailored control experiments will be necessary. We would also like to note that these data do not allow the conclusion that cyclodextrin-mediated incorporation of bifunctional lipids is non-consequential at any concentration. At higher concentrations, there will be adverse effects.

We have adapted the manuscript text to include the new datasets and considerations detailed above:

Lines 122-134, 145-149:

Furthermore, bifunctional lipid loading does not induce remodeling of the native lipidome, suggesting that no membrane stress response is triggered ². To test whether bifunctional lipid loading into the plasma membrane induces further cellular responses, we monitored membrane remodeling and cellular signaling during lipid loading. We measured the number of clathrin-coated pits and caveolae in the plasma membrane and quantified transferrin uptake as a proxy for bulk endocytosis using a panel of lipids with varying chemical structures and biophysical properties [Sphingomyelin SM(Y), PC(16:0/Y), PC(Y/20:4), PC(18:1/Y), PC(20:0/Y), PC(18:0/Y)]. We found no significant differences in transferrin uptake and the number of clathrin-coated pits. A small reduction in the number of caveolae ($8 \pm 2\%$) was observed in SM(Y) loaded cells, whereas no effect was observed for all other lipids (Extended Data 2). Thus, we conclude that no changes in overall lipid metabolism and bulk membrane trafficking occur upon delivery of bifunctional lipids. (...) Taken together, bifunctional lipids that mimicking structural membrane lipids can be understood as “innocent” tracer molecules for studies of intracellular lipid localization, transport, and metabolism, when used at the concentration in this study. When studying other specialized processes and/ or pathways, a new assessment based on tailored control experiments will be necessary.

In addition, regarding lipid abundance, how can authors be sure that localization reflects the native lipid localization as opposed to slow (or fast) trafficking from the plasma membrane? Lipids, like PA or SM, are signaling lipids and can inadvertently recruit proteins that may stabilize the lipid in wrong compartments. Can authors validate whether the localization of protein-based lipid probes, for example for PC, changes upon addition of exogenous lipid?

Answer:

Reviewer #1 raises two concerns: Stabilization of lipids in wrong compartments by lipid-binding proteins and the inadvertent initiation of signaling events. Regarding the first concern: Most lipids are present in much higher molar concentration than proteins in the cell due to their much smaller molecular masses (on average two orders of magnitude), which means that at any given point in time, the “free” pool of lipids will be significantly larger than the protein-bound pool. Lipid-protein interactions are also typically short-lived (nanoseconds to microseconds for the vast majority of the interactions), which means that protein-bound and free lipid pools rapidly equilibrate. For an excellent recent discussion of these issues, please see Levental and Lyman, Nat. Rev. Mol. Cell. Biol., 2023, 107-122 ⁶. Therefore, we do not think that “buffering” of lipids by lipid-binding proteins is a relevant mechanism for the common glycerophospholipids (such as PC) and common sphingolipids (such as SM), where the stoichiometry of proteins and lipids in the membrane necessitates a large pool of “free” lipids. In this regard, we would also like to refer to Reviewer Figure 2 – while the incorporation of bifunctional lipids is detectable by mass spectrometry, overall lipidome composition is not affected much. Therefore, relocalization of proteins that bind common

membrane lipids is unlikely – the relative lipid abundance changes are just too small. Along these lines, this is probably why so far (to the best of our knowledge) no PC biosensor has been established in the field. Around 30% of all membrane lipids are PC species, which makes generating a sensor with a useful dynamic range very challenging. These considerations are very different for rare signaling lipids. When following the cellular localisation of lipid second messengers such as PA or DAG, protein binding events have to be taken into account, as Reviewer 1 pointed out. These rare lipids are, however, not the focus of the present story, where we aim at developing a methodology for tracing structural (much more common) membrane lipids in the trafficking compartment.

To test whether loading of bifunctional lipid probes designed to mimic common membrane lipids (e. g., PC, PE, SM) induces cellular signaling events, we decided to monitor three signaling readouts: DAG signaling, PA signaling, and Calcium signaling. Since these three second messengers are involved in many signaling cascades, combining them all serves as a reasonable proxy for whether cellular signaling events are initiated upon lipid loading. We used genetically encoded DAG and PA sensors that translocate from the cytoplasm to the membrane when concentrations of the respective signaling lipids are elevated ^{7,8}, and Fluo4AM, a small molecule calcium indicator, which reacts to elevated Calcium levels with an increase in fluorescence. We monitored sensor responses after loading of U2OS cells with PE(18:1/Y), PC(18:1/Y), and SM(Y) and compared the observed responses with two positive controls: Treatment with ionomycin, which raises intracellular Calcium levels and loading of phosphatidic acid [PA(18:1/Y)], which should lead to the recruitment of PA and DAG (after dephosphorylation) sensors to the plasma membrane. We found that the structural membrane lipids [PE(18:1/Y), PC(18:1/Y), and SM(Y)] did not trigger signaling responses, whereas the expected calcium increase was observed after ionomycin treatment, and PA(18:1/Y) loading triggered the relocalization of the PA sensor to the plasma membrane. A similar trend was observed for the DAG sensor, which responded only to PA loading, indicating rapid dephosphorylation of PA after loading, which is in line with our previous lipid mass spec data ². These data are included in the new Extended Data Figure **3**, shown below. In summary, live-cell imaging experiments using lipid signaling and calcium sensors indicate that loading various structural membrane lipids to the plasma membrane does not induce major signaling responses in U2OS cells.

We have included the following paragraph in the main text to reflect these findings:

Lines 134-145:

Next, we tested whether loading of bifunctional lipid probes induces cellular signaling events. We monitored three signaling readouts: diacylglycerol (DAG) signaling (via a DAG biosensor) ⁸, phosphatidic acid (PA) signaling (via a PA biosensor) ⁷, and calcium signaling (using Fluo4AM). We monitored second messenger levels after lipid loading with a phosphatidylethanolamine PE(18:1/Y), PC(18:1/Y), and SM(Y) and compared the observed responses with two positive controls: Treatment with ionomycin, which raises intracellular calcium levels, and loading of phosphatidic acid [PA(18:1/Y)], which recruits PA and DAG (after PA dephosphorylation) biosensors to the plasma membrane. We found that the loading of the tested membrane lipids [PE(18:1/Y), PC(18:1/Y), SM(Y)] did not trigger measurable signaling responses (Extended Data **3**), whereas loading the control lipid PA(18:1/Y) induced a robust signaling response by

recruitment of the PA and DAG sensors as predicted.

Extended Data 3l Effect of lipid loading on cellular signaling events. **a.** To test if lipid loading induces major signaling events, the localization of DAG and PA protein sensors, as well as intracellular Calcium levels, were monitored during lipid loading using live cell imaging. A drop of sensor intensity in the cytosol is expected for the PA and DAG sensor upon recruitment to other membranes, whereas the calcium sensor intensity increases upon binding Ca^{2+} in the cytosol. **b.** The traces of the three sensors are plotted as the sensor intensity in the cytosol normalized against the cytosolic sensor intensity at time point -40 s before adding the lipid. Images were acquired every 10 s. Representative images of the sensor intensity are shown for the timepoints -40 s, 10 s, 60 s, and 260 s, and for the Calcium sensor (**c**), PA sensor (**d**), and DAG sensor (**e**). Timepoint 0 marks the addition of the lipid.

Reviewer #2 (Remarks to the Author):

Lennartz et al., report on a Lipid-CLEM approach which enables investigation of the lipid composition on cellular organelles at high resolution. The article is comprehensible and highlights the advantage of their novel approach to existing methods. Lastly, using their Lipid-CLEM approach they could show segregation of different cargo in EE, in line with the published literature.

The authors have very comprehensively described their Lipid-CLEM approach and its clear advantages over established methods. Overall, I believe their described method offers a valuable approach for studying lipid composition, which has remained challenging. The authors undoubtedly fill an important gap in the visualization of lipid domains in cells.

Answer:

We thank Reviewer 2 for the overall positive assessment of our work, in particular with regard to the methodological novelty and advantage to pre-existing methods.

Major comment:

1. The existence of different nanodomains on EEs that would sort distinct cargo has been previously reported, though not including the visualization of lipids at such high resolution. Defining endosomes with specific markers is challenging as they may harbour different sorting machinery with different function. To strengthen the paper, the authors could look at machinery such as Rab5/EEA1/Rab4 and try to understand whether different lipid domains drive machinery/cargoes segregation on endosomes.

Answer:

We agree that a full investigation of the endosomal sorting machinery is an exciting application of our approach in combination with visualizing co-localisation of lipids to better elucidate molecular mechanisms of lipid sorting. Similar to our answer to reviewer 1 who raised a similar point, we would, however, argue that such a study is beyond the scope of the current manuscript, which is meant to primarily describe the Lipid-CLEM workflow and was accordingly submitted as a technical report (*Nature Cell Biology's* methods paper format, with the following stated scope: "A Technical Report presents primary research data on a new technique that is likely to be influential"). The analysis suggested by Reviewer 2 would require the establishment of well-characterized cell lines of the respective fluorescent proteins at endogenous levels to avoid trafficking artefacts (Immunofluorescence labeling is not compatible with the current workflow, as it would require chemical fixation at room temperature and membrane permeabilization, which will

likely induce artifacts). Thus, we hope that reviewer 2 can agree with us that the suggested investigation - while being a very exciting future research direction - is beyond the scope of the current story.

2. In the discussion the authors mention “protein and lipid transport routes diverge at the early endosome as sphingomyelin and transferrin arrive simultaneously in clathrin-coated vesicles, but partition differentially into endosomal compartments”. Here, adding clathrin into the picture would help strengthen the authors conclusions.

Answer:

This is a very good suggestion, which we decided to follow up on experimentally. To test whether transferrin and SM(Y) are taken up together via clathrin-mediated endocytosis, as proposed in our manuscript, we loaded U2OS cells with SM(Y) and transferrin and performed co-staining for clathrin. We found that SM(Y) was relatively evenly distributed in the plasma membrane and also present in clathrin-coated pits (Extended data 7). These data are in line with our recent preprint (), where we extensively characterize lipid enrichment in clathrin-coated pits across a library of 10 lipid probes representing the major membrane lipid classes. The conclusion of this study is that lipid incorporation into clathrin-coated vesicles is best explained by the underlying lipid composition of the plasma membrane and its inherent transbilayer lipid asymmetry, indicating that clathrin-mediated endocytosis combined selective protein and non-selective lipid uptake.

In the new dataset added to this manuscript, we find strong co-localization of transferrin with clathrin, though not all clathrin-positive structures were transferrin-positive. Quantitative analysis of the mean fluorescence intensity (FI) of SM or transferrin over clathrin masks revealed a modest enrichment of SM (which is in line with the expected higher membrane density compared to the plasma membrane due to the geometry of clathrin-coated pits, an effect explained in more detail in <https://www.biorxiv.org/content/10.1101/2025.03.04.641423v1>) and a stronger enrichment of transferrin at clathrin-positive structures.

In summary, our co-localization assay demonstrates that both SM and transferrin are present in clathrin-coated pits at the plasma membrane, strongly suggesting their co-uptake into cells during endocytosis and subsequent trafficking to early endosomes. Because SM(Y) is evenly distributed across the whole PM, while transferrin is partially excluded in some clathrin-coated pits, we conclude that SM(Y) can traffic independently from transferrin, but transferrin does not traffic independently from SM(Y). Since we select endosomes for quantification based on localisation of LDL and transferrin, we assume that to those endosomes lipid and transferrin are co-transported, and differential localization in the endosome (ILVs vs recycling tubules) is due to processes occurring at the endosome and not during the formation of clathrin coated vesicles.

Extended data 7I Transferrin and sphingomyelin co-localize with clathrin-coated pits. **a.** A representative image of a U2OS cell loaded with transferrin (purple) and SM(Y) (orange) and immuno co-stained for clathrin (blue). The basal PM is shown. Scale bar: 10 μm . ROIs show clathrin-coated pits and their co-localisation with SM(Y) and transferrin. Scale bar: 500nm. **b.** Quantification of the mean fluorescence (FI) of SM or transferrin signal red out over the mask of clathrin-coated pits, normalized over the mean FI over the whole cell.

We have included/updated the following paragraph in the main text to reflect these findings:

Lines 359-367:

To confirm whether the differential localization of SM(Y) and transferrin in early endosomes is due to sorting within the endosomal compartment rather than sorting into different populations of endocytic pits we simultaneously loaded SM(Y) and transferrin and quantified incorporation into clathrin-coated pits. We find that SM(Y) is present in all transferrin-positive clathrin-coated pits (Extended Data 7), in line with our recent analysis of lipid partitioning into clathrin-coated pits 36. This indicates that SM and transferrin were taken up together by endocytosis, but separated in early endosomal compartments, with transferrin being enriched in recycling tubule and SM in ILVs. Taken together, our data imply that sphingomyelin is sorted into distinct membrane sub-compartments within the early endosome.

Furthermore, how can it be determined that the tubular compartment adjacent to the EEs is derived from/ part of the compartment?

Answer:

We determine that the tubular compartment adjacent to the EEs is part of the compartment by checking the connectivity of the tubular compartment to the boundary membrane. This can be directly assessed from the EM data. For the analysis of K11M samples, we assess connectivity throughout the 3D tomogram. We would like to note that membrane ultrastructure in K11M is more perturbed compared to HM20, resulting in more fractionated appearing membranes. Thus, continuous connectivity between the recycling tubule and boundary membranes is not always given, but can be assessed throughout the tomogram via max projection of the respective image slices and visualization of tomograms as movies. We include exemplary movies in the updated supplementary datasets (Supplementary Figure 6, Supplementary Movies 1 and 2) to demonstrate how connectivity is assessed.

In the case of HM20, for which we repeated the lipid density dataset as requested by Reviewer 3, the connectivity between the boundary membrane and recycling tubule appears clearer, since membrane structure is better preserved. Two examples per resin are shown below to highlight the connectivity between the two compartments in Supplementary Figure 6 below in addition to the new supplementary movies 1 and 2. Since EM data is acquired in fixed samples, we cannot be certain that the tubule is derived from the EE from electron microscopy data alone, since there is the possibility that incoming tubules could, in theory, also fuse with the EE instead. However, to the best of our knowledge, for EEs, it is more common that vesicles (round appearance) fuse with EEs, and tubules (elongated appearance) are pulled away from the EE to eventually cleave off. Thus, we assume here that tubular membranes attached to the boundary membrane are derived from the EE, which is also in line with the transferrin enrichment, a hallmark of recycling tubules that originate from the endosome.

Supplementary Figure 6I Connectivity of recycling tubules to early endosomes. Exemplary early endosomes embedded in K11M and HM20 are shown. The area where the boundary membrane transitions to recycling tubules is highlighted and shown magnified. Red arrows highlight the neck region. Scale bars: 200 nm.

Are these tubular compartments devoid of clathrin?

Answer:

From our EM data, we cannot report clathrin localisation to recycling tubules; however, sample preparation was not optimized to maintain clathrin ultrastructure. Thus, the lack of visible clathrin lattices might be caused by the sample preparation method and not due to the actual lack of clathrin. To experimentally address this question, we made use of the data described above, where U2OS cells were loaded with lipid and transferrin and co-stained for clathrin (Compare Extended Data 7). Here, we did not image the basal plasma membrane but focused on intracellular compartments. We used the lipid signal to identify endosomal structures to which also transferrin localised. Transferrin in EEs localizes, as demonstrated also by our CLEM data, to recycling tubules. From these data, co-localisation with clathrin was assessed. In many but not all instances, we find clathrin signal present at recycling tubules. The clathrin signal, in most cases, however, is strongest in areas of high SM(Y) signal, which likely corresponds the boundary membrane of the endosome (see new Revier Figure 3, also shown below). This likely means that incoming clathrin coated vesicles primarily fuse with the boundary membrane, and the TfR-Tf complex partitions into recycling tubules, whereas SM(Y) is enriched in ILVs. The clathrin localization we find here is consistent with previous studies from the literature, which have reported on the localisation of clathrin at recycling tubules using fluorescence imaging 9,10.

Revier Figure 3I Clathrin localization at endosomes. U2OS cells were incubated with SM(Y) (orange) and Transferrin (purple) for 4min. Clathrin (blue) was immunostained. Four representative endosomes are shown, indicating clathrin localization also to the recycling tubule (marked by Tf). Scale bars: 1 μm

Minor comments: 3.

Please clarify the description and Figure panels in Extended Data Figure 6: i. "A mix of Rab5 and EEA1 was used to mark early endosomes" – are the investigated compartments are positive for both or either of the markers?

Answer:

We thank the reviewer for noticing this oversight and apologize for the error in the axis labeling of the graphs. In fact, only EEA1 was used as a co-stain to label early endosomes. We have corrected this mistake in Extended Data Figure 5 (as shown below), and the analyzed compartments are now correctly indicated as EEA1-positive. For a more representative comparison, we have also added the corresponding co-stain to Extended Data Figure 5b in place of the Rab5 co-stain.

ii. In 'A', the separation of the cargo-endosome time graphs is misleading. The yellow curve (Rab5/EEA1) shows different percentages in the left vs. right graph. I assume this results from the normalization of “total endosomes” - on the left graph, maybe EE + LE, and on the right, EE + RE? I disagree with the wording “total endosomes”, as to accurately represent the total number of endosomes, additional markers would be needed. Furthermore, the Rab11 curve for Tf_n recycling appears unusually low, almost comparable to the Rab7 curve which disagrees with other studies (<https://doi.org/10.1083/jcb.149.4.901>). I recommend merging the graphs to avoid this confusion.

Answer:

We thank Reviewer 2 for this valuable input. To improve the readability and clarity of Extended Figure 5a, we have incorporated all suggestions and combined the graphs accordingly. We also agree that the term “total” was potentially misleading, as it could be interpreted as referring to the *total number of endosomes*. Our intention was to indicate the *total fluorescence intensity* of either the transferrin or LDL signal within the sample. To clarify this, we now describe the data as the percentage of fluorescence intensity in the labelled endosomal area relative to the total cellular fluorescence intensity, denoted as $FI_{Tf}(labelled\ endosomes) / FI_{Tf}(whole\ cell) [\%]$ and $FI_{LDL}(labelled\ endosomes) / FI_{LDL}(whole\ cell) [\%]$. The updated Extended Data Figure 5 is shown below.

Extended Data 5I LDL and Transferrin specifically label endosomal compartments. **A:** Transferrin and LDL were first assessed separately for their localization to distinct endosomal compartments. Transferrin was loaded for 4 min, and chased for 0, 5, 10, 15, 20 and 30 min. LDL was loaded for 15 min and chased for 0, 5, 10, 15, 20, and 30 min. EEA1 was used to mark early endosomes, Rab7 was used for late endosomes, and Rab11 was used for recycling endosomes. The fluorescent values of either transferrin or LDL were read out over the segmentation markers and normalized to the total transferrin or LDL fluorescence signal. The analysis excludes compartments that are positive for both early and late endosomal markers or markers of early and recycling endosomes. Mean values are shown, and error bars represent the standard deviation. The experiment was done as a single biological replicate. **B:** Transferrin and LDL were loaded together to assess their combined localization to distinct endosomal compartments. Representative images of U2OS cells labelled for 15 min with LDL (orange), chased for 15 min,

and labeled for 4 min with transferrin (magenta). Endosomal compartments were either immunolabelled with EEA1, Rab7, or Rab11 (blue). The outline of the representative cell is highlighted by a white dotted line. Scale bar: 10 μ m, ROI: 1 μ m.

iii. In 'B', Rab11-positive REs appear predominantly localized around the perinuclear area, which is surprising given that REs have also been reported at the cell periphery. Additionally, most of the REs in the image appear round and less tubular compared to what is typically described in the literature. Could this be due to an unrepresentative image, the antibody used, or issues with structure preservation during fixation?

Answer:

We agree that the previous image was not fully representative, as it depicted only a small, cropped region of a cell. We have therefore updated Extended Figure 5 to include a more representative image showing Rab7, EEA1, and Rab11-positive structures across entire cells, rather than solely a magnified, cropped area. To further validate our findings, we confirmed the specificity of the antibody used for immunostaining by staining cells overexpressing Rab11-GFP, as shown in Supplementary Figure 4. In these samples, tubular structures are clearly visible, indicating that the fixation procedure sufficiently preserves endosomal morphology. However, we acknowledge that chemical fixation may still introduce alterations to the native structure of endosomal membranes. Notably, the tubular structures are more prominent in the Rab11-GFP channel, suggesting that the antibody used does not label tubular membranes as efficiently, which may account for the more clustered appearance observed in the immunostained samples.

Supplementary Figure 4 A Rab11 antibody binds Rab11 specifically. U2OS wildtype cells were transfected to express Rab11-GFP (orange). Samples were fixed and immunostained against Rab11 using a Rab11 antibody (blue). Scale bar: 10 μ m.

Lines 173 – 176:

The use of transferrin and LDL as suitable endosomal markers was demonstrated by co-staining with Rab5 (early endosomes), Rab7 (late endosomes), and Rab11 (recycling endosomes, **Antibody validated by co-expression of Rab11-GFP (Supplementary Figure 4)**) (Extended Data 5).

Reviewer #3 (Remarks to the Author):

The study introduces a novel Lipid-CLEM workflow that integrates bifunctional lipid probes, rapid photo-crosslinking, high-pressure freezing, and on-section click chemistry to achieve nanoscale visualization of lipid distribution. This method quantitatively demonstrates that sphingomyelin partitions differentially within early endosomal compartments, being enriched in intraluminal vesicles and depleted in recycling tubules. These findings offer new insights into membrane organization and the mechanisms underlying lipid sorting.

The study employs an innovative approach by developing the Lipid-CLEM workflow, which is technically impressive and successfully overcomes long-standing challenges in high-resolution lipid imaging. The dual approach, utilizing both surface labeling with HM20 resin and whole-section labeling with K11M resin, is particularly well conceived. Integrating light and electron microscopy enables detailed nanoscale quantification of lipid distribution, and the quantitative analysis of sphingomyelin partitioning in early endosomes is both compelling and adds significant value to the study. Furthermore, the methodology is executed with high rigor, as demonstrated by the detailed description of rapid fixation and on-section click chemistry, offering a reproducible framework for future research. Finally, the biological relevance of the work is clear, as the application of the method to study lipid sorting in early endosomes addresses a critical question in membrane biology.

This reviewer finds the study suitable for publication. Consideration of the feedback below would strengthen the study.

Answer:

We thank reviewer 3 for the overall positive assessment of our work, in particular with regard to the methodological novelty and rigor of the work.

FEEDBACK:

• **Crosslinking Specificity & Efficiency:** Although Extended Data 3 includes negative controls that were not treated with UV light or PC(16:0|Y), confirming that fluorescence requires both UV irradiation and the probe, it remains important to further quantify the crosslinking specificity and efficiency. Incorporating a time-course experiment to systematically vary UV exposure would help determine the minimum irradiation time needed for effective crosslinking and provide a clearer kinetic profile of the reaction. This additional quantification would solidify that the fluorescence signal is indeed a specific result of the photo-crosslinking process and not due to non-specific interactions.

Answer:

Reviewer 3 suggests further validation of crosslinking specificity via a time-course with varying UV exposure, and thus for determining the optimal exposure time. We fully agree that this is what one has to do – in fact, we have already done the requested experiment, which is included in our recent preprint on lipid partitioning into clathrin-coated pits (CCPs) at the plasma membrane (<https://www.biorxiv.org/content/10.1101/2025.03.04.641423v1>, Extended Data 1) 11.

In this study, we systematically characterize UV crosslinking efficiency across different LEDs and exposure durations, both in solution and in cells. We tested a 308 nm LED, a 365 nm LED without a lens, and the same 365 nm LED equipped with a lens; the latter configuration is used in the current study.

First, we monitored diazirine photolysis as a function of exposure time for the bifunctional fatty acid (Y) in DMSO solution by following the decrease of the 350 nm diazirine absorbance band. These measurements showed that 365 nm illumination yields the highest photoactivation efficiency, with a modest improvement when a lens is used. We then assessed photoactivation

performance in cells by loading U2OS cells with SM(Y) and delivering UV pulses of 0 s, 2 s, 5 s, and 10 s. Consistent with the data acquired in solution, the 365 nm LEDs produced the strongest lipid signal, and the signal reached a plateau by approximately 2 s, indicating near-maximal activation at short exposure.

Based on these results, we selected the 365 nm LED with a lens and a 3 s pulse for all experiments in the present work. This setting provides overall high photoactivation at high temporal resolution while minimizing thermal load and avoiding sample heating. The results of the previous optimization are shown below in Reviewer Figure 4. We have added the following reference to the optimization of crosslinking conditions into the main text of the manuscript:

Lines 154-157:

With this setup, a 3 s irradiation pulse was sufficient to achieve signal intensities comparable to irradiation through uncoated coverslips (Supplementary Figure 2), **in line with the results of an optimization of crosslinking conditions we previously performed**¹¹.

Reviewer Figure 4| Characterization of the diazirine photo-crosslinking efficiency. A: The chemical structure of the previously reported **2** bifunctional palmitic fatty acid (Y) is shown. B: The diazirine spectrum of Y is measured at 100mM concentration in DMSO. Spectra were measured after different exposure times of 0s (yellow), 2s (orange), 5s (red), and 10s (deep red) with a recently reported 308nm LED **2** compared to 2 high power LEDs at 365nm with and without a collimating lens. The spectra of the native fatty acid (FA, green) lack a diazirine peak. Photoactivation experiments were performed in triplicates. Mean values and standard deviation are depicted. C: U2OS wildtype cells were loaded for 4min with bifunctional SM, UV activated for different time intervals and with different LEDs. Signal intensities are scaled equally for all images. Scale bar: 20 μ m. (Lennartz et al., 2025, ED figure 1) **11**

•**Resin-Dependent Artifacts:** The manuscript discusses differences between HM20 and K11M

resins—with Extended Data 2 and Extended Data 7 showing that HM20 restricts labeling to the section surface while K11M permits more uniform staining across the section. A direct quantitative comparison of lipid partitioning outcomes between these two resin types is necessary to ensure that the observed differences in sphingomyelin distribution (i.e., enrichment in intraluminal vesicles versus depletion in recycling tubules) are intrinsic to the biological phenomenon and not artifacts introduced by the embedding process. This direct comparison would reinforce the overall conclusions by confirming that resin-dependent variability does not confound the interpretation of lipid partitioning.

Answer:

Reviewer 3 suggests a direct quantitative comparison for lipid partitioning outcomes between both HM20 and K11M resins to highlight how the different staining depths affect the outcome of our lipid density analysis. We agree that comparing lipid localization trends for the two resins is a good idea; however, there are intrinsic limitations to performing this type of analysis in HM20 samples. The calculation of lipid densities carried out here is in effect a normalization to the membrane surface areas derived from tomograms. Crucially, this normalization implies that the labeling of lipids is occurring on all membrane surfaces, which is not the case for HM20. The fraction of crosslinked lipids that are accessible for the staining reagent in HM20 samples is (i) unknown, but lower compared to K11M, and (ii) may be different for the respective membrane compartments due to their size and geometry. Thus, while overall trends can and should be compared, a truly quantitative comparison is not possible. The enrichment/ depletion values for K11M are in fact, relative lipid densities, whereas this is not the case for HM20. We decided to minimize this problem by converting the problem into a 2D problem and quantifying lipid enrichment in 2D on thin HM20 sections (100 nm) rather than 3D tomograms (see below). This means that the enrichment values are not directly comparable (2D vs 1D densities), but a comparison of trends is more meaningful, as less unlabeled membrane contributes to the EM segmentation.

Experimentally, we loaded cells with SM(Y), Tf, and LDL as described, but embedding was done in HM20. Since staining depth in HM20 is restricted to the resin surface (see updated Figure 2 and discussion below), blocks were sectioned to a thickness of 100nm to maximize the number of surface-exposed lipids relative to section volume. Staining and fluorescent image acquisition were performed as described. For EM, instead of tomography, we acquired 2D EM images of the thin sections. Notably, while this greatly improves throughput, we have to make the assumption that all visible membranes in the 2D image are actually exposed to the section surface and thus exposed for staining by click chemistry. This is likely not the case and thus introduces a systematic error which presumably leads to overall smaller effect sizes, especially for smaller membrane compartments such as the RTs and ILVs. The results of the experiment described above are shown below in Supplementary Figure 9. We find that the general trends observed for K11M are retained in HM20 samples, but all effect sizes are much reduced, both for the localization of lipid signal and LDL/transferrin signal, the latter probably due to the much thinner section. Specific trends for individual compartments are described below:

- (i) Recycling tubules (RTs): SM(Y) depletion is observed in both resins, with a somewhat bigger effect size in K11M. Transferrin shows enrichment in both resins as expected, with overall less enrichment for HM20. This difference could be caused by the different resin thickness, with a smaller section thickness (100nm, HM20) resulting in a less pronounced enrichment compared to thicker sections (K11M) due to less favorable signal-to-noise ratios.

- (ii) Intraluminal vesicles (ILVs): in both resins, LDL is enriched as expected, but less pronounced so in HM20. This difference can be caused by the difference in section thickness, as discussed in (i). For both resins, we measure lipid enrichment; however, the lipid enrichment is again much more pronounced in K11M. This difference is likely due to several issues. First, due to lower fluorescent intensity caused by surface stain only, we expect a smaller signal-to-noise ratio in HM20 that can result in less pronounced intensity differences between ILVs and the boundary membrane. Secondly, the different labeling depth efficiencies in both resins must be considered. We assume that all membranes of the endosomes are actually exposed to the section surface; however, this is more likely the case for larger membranes, such as the boundary membrane, and less likely for smaller objects such as ILVs. Thus, we very likely overestimate the exposed membrane for ILVs and subsequently underestimate ILV lipid densities compared to larger compartments. Finally, the section thickness issue is of course in play for the lipid signal as well.
- (iii) Boundary membrane: Lipid enrichment values are similar for both resins.

Taken together, we find that corresponding overall lipid enrichment/ depletion trends are observed for both K11M and HM20, and thus resin-independent.

We have added the following statement to the main text:

Lines 352-358:

To exclude resin-dependent artifacts, we performed a similar experiment using HM20 resin. Here, we acquired 2D EM images of thin (100 nm) sections to maximize the number of surface-exposed lipid residues available for staining and calculated 1D densities from 2D masks (Supplementary Figure 9). These values are not directly comparable to the 2D densities derived from K11M tomograms and effect sizes are generally smaller in thin HM20 sections. Despite these limitations, the general trend of SM(Y) depletion in recycling tubules and enrichment in intraluminal vesicles was maintained.

Furthermore, we included a new Supplementary Figure 9 to the Supplementary Information, which was also updated with a detailed description of the analysis strategy and a summary of the considerations above, which we did not include in the main text for space reasons.

Supplementary Figure 9I Endosomal lipid partitioning trends are recapitulated in HM20. **A:** HM20 embedded samples loaded with SM(Y) were sectioned into 100 nm sections and stained. Three representative endosomes are shown: the corresponding EM image; the merge of the fluorescence channels of SM(Y) (orange), transferrin (Tf, purple), and LDL (blue); and the surface model of the endosomes membranes overlaid with each fluorescent signal. The individual quantification for each endosome is shown on the right. RT – recycling tubule, BM – boundary membrane, ILV – intraluminal vesicle. Scale bar: 200nm. Fluorescence of each endosome is scaled individually. **B:** The relative fluorescent densities are plotted for all analyzed endosomes for the lipid SM(Y) of the samples embedded in HM20 and compared against K11M. For K11M relative values for lipid signal over membrane surface area (2D) are plotted. For HM20 relative values for lipid signal over membrane outlines (1D) area are plotted. A direct quantitative comparison is thus not possible, but overall enrichment and depletion trends can be compared and are conserved. The data for K11M is also shown in main text figure 5D. The mean densities are shown for the different domains: recycling tubule (magenta), boundary membrane (orange), and ILV (blue), and for the respective fluorescent signals of lipid, transferrin, and LDL. Maximum possible fold-enrichments are indicated by the dotted horizontal lines for the boundary membrane

(orange), recycling tubule (magenta), and intraluminal vesicles (blue). The maximum possible fold-enrichment of ILVs in HM20 (4.58) is not shown. No enrichment is indicated by the gray line at 1.0. Error bars are shown as the 95% confidence interval of the mean values. 2 independent biological replicates were performed both for HM20 and K11M.

OTHER

CONSIDERATIONS:

• **Probe Behavior vs. Native Lipids:** While Extended Data 1 provides a comparison between NBD lipids and the bifunctional probes, additional validation is needed to ensure that the minimally modified bifunctional probes truly recapitulate native lipid behavior. Employing an orthogonal method—such as mass spectrometry or an alternative imaging approach—could confirm that the probes do not introduce subtle artifacts affecting lipid distribution. This step is critical because the reliability of the method hinges on the probes mimicking native lipid properties as closely as possible.

Answer:

We fully agree that biophysical characterization of the bifunctional lipids is of high importance. As such, we have extensively addressed this aspect in our previous work (Iglesias-Artola et al., Nature 646, 474–482 (2025); Supplementary Figure 1), to which we would like to refer. To assess whether the modifications introduced into the bifunctional lipids altered their biophysical properties, we (i) compared the influence of bifunctional lipids on membrane nanodomain formation with their native analogues lacking diazirine and alkyne moieties. Additionally (ii), Laurdan temperature scans were performed on vesicles incorporating bifunctional PC and PE probes to assess the influence of the dual modifications on membrane phase behavior. These combined analyses demonstrate that the changes in biophysical properties imposed by the diazirine and alkyne units are minimal and are comparable to, or slightly less pronounced than, the effect of introducing a single additional double bond into the acyl chain; in other words, the bifunctional chain most closely resembles a 16:1 acyl chain.

Furthermore, we compared the metabolic processing of bifunctional lipids to that of endogenous lipids (Iglesias-Artola et al., Nature 646, 474–482 (2025); Extended Data Figure 5). Specifically, we compared the metabolic fate of bifunctional PC species carrying an oleic or palmitic acid with a native isotope-labelled species, PC(18:1/16:0[C¹³]). Bifunctional and isotope-labelled lipids were delivered to the outer leaflet of the plasma membrane by a cyclodextrin pulse. Our results indicate that PC(18:1/16:0[C¹³]) is metabolized similarly to the bifunctional PC species PC(Y/16:0) and PC(16:0/Y), whereas the species containing oleic acid (PC(Y/18:1) and PC(18:1/Y)) are metabolized at a somewhat faster rate. Importantly, the retention of the original PC(18:1/16:0[C¹³]) species, expressed as a percentage of all labelled lipids, is comparable to that of the palmitate-containing bifunctional lipids. This suggests that the metabolism of bifunctional lipid species closely recapitulates that of native lipids, with a monounsaturated fatty acid substituting for the bifunctional fatty acid, consistent with the in vitro biophysical characterization.

Given that the biophysical properties and metabolism of bifunctional lipids closely mirror those of endogenous analogues, we are confident that the bifunctional modification presents only a minimal perturbation.

These datasets are shown below in Reviewer Figure 5, we also included the following clarification in the main text to directly reference these datasets:

Lines 51- 55:

We have previously shown that they are metabolized similarly to native lipids, and that the combined effect of the diazirine and alkyne modifications is roughly equivalent to that of one additional double bond, both with regard to biophysical properties and lipid metabolism 2. Bifunctional lipids can be introduced into membranes of living cells and rapidly photo-crosslinked to neighboring proteins.

Reviewer Figure 5I Probe Behavior of bifunctional lipids. **a**. Comparison of bifunctional lipid metabolism with the native, isotope labelled, monounsaturated PC species PC(18:1/16:0 [¹³C]). **b**.

Development of the PC species distribution of PC(18:1/16:0^[13C]) and palmitate-containing bifunctional PC species shows similar persistence of the original species in the labelled lipidome and similar product species forming. Mean and 95% CI of 3 biological repeats containing 2 technical replicates each are shown for **a**, **b**, **c**. Bifunctional lipid probes were characterized for their biophysical properties. Formation of ganglioside nanodomains leading to faster deexcitation of Bodipy-FL-GM1 donors via FRET is unaffected by replacing the closest structurally related native lipid with the bifunctional variant for all probes. **d**. Laurdan temperature scans indicate that the biophysical implications of introducing a bifunctional fatty acid are small.

• **Uniformity of Fluorescence Labeling:** Although Extended Data 7 demonstrates that HM20 resin limits labeling to the section surface and K11M resin offers deeper penetration, further quantitative analysis of labeling uniformity across the entire section is recommended. Ensuring that fluorescence labeling is consistent throughout the section is essential for accurate lipid density measurements. Variability in the labeling intensity could confound the interpretation of lipid partitioning, particularly when making quantitative comparisons across different membrane subdomains.

Answer:

We agree with Reviewer 3's observation that the labeling uniformity is crucial for our quantitative analysis. To address this point, we performed two experimental variations to assess the uniformity of the fluorescent staining. Specifically, whole blocks of either K11M or HM20-embedded samples, pre-loaded with lipid, were subjected to click chemistry staining. These stained blocks were subsequently sectioned either vertically or horizontally (Figure 2A, D). The experiment and analysis of vertically sectioned samples were already performed in the first submitted version of our manuscript. However, the analysis of horizontally sectioned samples (previous Extended Data 7, removed due to the now more comprehensive Dataset included in Figure 2) was only performed for 1 independent sample of HM20 and without any additional quantitative analysis. We thus now performed the experiment of horizontally sectioned samples for 2 independent samples and for both resins. Briefly, the main findings of these experiments in regard to labeling uniformity can be summarized as follows:

For vertically cut blocks, staining depth was directly visualized using fluorescence microscopy. We observed that the average staining widths were approximately 450 nm (close to the resolution limit of fluorescence microscopy) for HM20 and 870 nm for K11M. In K11M samples, internal membranes were clearly identifiable in vertical sections, while in HM20 samples, the fluorescent signal was predominantly restricted to the block surface (Figure 2B, C). These results indicate that the click reaction reagent penetrates the full thickness of K11M blocks, but is possibly confined to the surface in HM20 blocks.

To further investigate the staining depth in HM20, given the resolution constraints of light microscopy for the previous approach, we implemented an alternative experiment. Stained blocks were cut horizontally into 100 nm sections, and the presence of the fluorescent stain was assessed in consecutive sections. For HM20, signal was present in up to six consecutive sections (or even more), but a Pearson coefficient analysis demonstrated mutual exclusivity of the signal between sections (Figure 2E, F). This is indicative of the fact that fluorescence is indeed limited to the block/ section surface. The stainings of subsequent sections likely results from surface unevenness rather than genuine penetration. In contrast, K11M samples exhibited overlapping signal across consecutive sections, as expected for uniform labeling.

We want to highlight that this horizontal-sectioning approach is subject to experimental limitations, primarily due to the inherent variability in section thickness during sectioning. This is particularly

problematic for K11M, which is a softer resin and poses additional cutting challenges. Therefore, we consider vertical sectioning to provide a more accurate estimate of actual staining penetration, while horizontal sectioning is better suited to assess whether staining is truly limited to the block surface. The new data are included in the expanded main text Figure 2. To avoid duplication we removed the previous Extended Data 7. We reference the new K11M data as follows in the main text:

Lines 211 – 214, 238-241:

Therefore, we additionally sectioned fully stained HM20 blocks horizontally into thin sections of 100 nm (Figure 2D-F). The signal of the consecutive sections was found to be mutually exclusive, suggesting that the click labeling is limited to the section surface in HM20 samples. (...) Next, fully stained K11M blocks were cut into horizontal sections and fluorescent signal was shown to be overlapping in subsequent sections. Overall, this indicates that the click reagents penetrate into the K11M resin (Figure 2D-F).

Figure 2I Comparison of dye diffusibility into HM20 and K11M resins. **A:** Workflow for testing the penetration depth of the click reaction mixture into resins. Whole resin blocks were stained and subsequently sectioned vertically to assess penetration depth. **B:** Representative images of stained and vertical sectioned blocks of HM20 (upper) and K11M (lower), of samples loaded with (+Lipid +UV) and without (-Lipid -UV) PC(16:0IY). Scale bar of overview images: 10 μm ; Scale bar of ROIs: 1 μm . Dotted red lines represent cell borders if not visible otherwise. **C:** The average line profile (indicated in B) of the fluorescence signal of vertical sections of HM20 and K11M with (black line) and without (dotted line) PC(16:0IY). The error of the average line profile is shaded in grey and given as the confidence interval of 95%. The full width at half maximum (\pm standard

deviation) for the lipid-loaded samples is indicated. The purple bar indicates the resolution limit of the optical system as determined with 100 nm beads. **D:** Workflow for testing the penetration depth of the click reaction mixture into resins. Whole resin blocks were stained and subsequently sectioned horizontally to assess penetration depth. **E:** Representative images are shown of the first six horizontally cut sections merged and the first 4 sections separately of stained blocks for HM20 and K11M. Scale bar: 10 μ m. Colors indicate the number of the section, with section 1 being the first section collected from the block. Section thickness: 100nm. **F:** The Pearson coefficients between serial sections were determined for the method shown in D. Coefficients were calculated for subsequent sections. Two independent samples were analyzed for each resin. High Pearson coefficients indicate an overlap of signal between sections, whereas low coefficients indicate mutually exclusive signals.

- **Generalizability:** The study currently focuses on sphingomyelin distribution in early endosomes within U2OS cells. To establish the broader applicability of the Lipid-CLEM workflow, extending the analysis to additional lipid species or other cell types would be beneficial. Demonstrating that the method works reliably across various biological contexts would not only reinforce its robustness but also highlight its versatility as a tool for investigating lipid organization in diverse cellular environments.

Answer:

To demonstrate a broader applicability of our approach, we now demonstrate that Lipid-CLEM works reliably across lipid probes from different lipid classes. Besides the phosphatidyl choline (PC(YI16:0)) used for the optimization of the protocol during this study and sphingomyelin (SM(Y)) used for quantification in early endosomes we here tested the localisation of two additional lipids: phosphatidylethanolamine PE(YI18:1) and a phosphatidylcholine carrying a polyunsaturated fatty acid (PC(YI20:4)). Using Lipid-CLEM we see differential localisation of all lipids after 4min of trafficking into the cells. PC(YI16:0) and SM(Y) show the strongest localisation to the plasma membrane and endosomes, while PC(YI20:4) shows a modest localisation towards mitochondria but reduced signal at the plasma membrane. The fluorescent signal of PE(YI18:1) is reduced both at the PM and in endosomes, while its localisation to mitochondria is much stronger. These localisation trends are in agreement with our previous study on lipid trafficking in U2OS cells². Overall, this demonstrates the versatility of our approach, showing that differential lipid localisation is recapitulated by our lipid-CLEM approach for a broad set of bifunctional lipid probes. Furthermore, while we did not include other cell types here due to the inherently low throughput of electron microscopy, bifunctional lipids have been used by us and others for fluorescence imaging in multiple cell lines, including HCT-116 cells ², Hela cells ¹², and iPSC-derived neurons ², indicating that the workflow should be readily adaptable to include other cell types. We are confident that our Lipid-CLEM approach works reliably in a broad set of different cell types as long as the bifunctional lipids can be delivered and the cell culture system is suitable for high-pressure freezing.

We have included the dataset above in the new Main Text Figure 4 and Supplementary Figure 5 (shown below), and included the following section in the main text:

Lines 287 – 295:

To demonstrate that the Lipid-CLEM workflow is compatible with various lipid classes and enables analyses of their organelle distribution, we acquired Lipid-CLEM datasets for different lipids: PC(Y/16:0), SM(Y), PC(Y/20:4) and PE(18:1/Y). We compared lipid localization in the plasma membrane, in endosomes, and in mitochondria. PC(Y/16:0) and SM(Y) showed a pronounced plasma membrane and endosomal localization with no noticeable localization in mitochondria. PC(Y/20:4) plasma membrane localization was found to be lower, and some signal was present in mitochondria, while we found a pronounced localization of PE(18:1/Y) in mitochondria, observations that are in line with our previous data (Figure 4, Supplementary Figure 5) ².

Figure 4 Lipid CLEM of various lipid classes and species. Cells were treated with various lipid classes and species, including bifunctional sphingomyelin (SM(Y)), bifunctional phosphatidylcholine of different fatty acid composition (PC(Y16:0), PC(Y120:4)), phosphatidylethanolamine (PE(Y18:1)). All chemical structures are indicated on the left. Representative fluorescent images of the lipid signal are shown for all lipids in orange, highlighting overall differential lipid localization across whole cells for the different lipids. The respective correlated EM image is shown to the right. Correlated regions of interest are shown for the plasma membrane, mitochondria, and endosomes. Fluorescence intensities for images of the same lipid are scaled the equally. Scale bar: 10 μ m. Correlation error maps of the images shown here can be found in the supplementary figure 3.

Supplementary Figure 3 Correlation maps for Lipid CLEM of various lipid classes and species. All chemical structures are indicated on the left. Representative EM, FM and correlated images are shown for all lipids (the same images are shown in Main Figure 5). A region of interest of each cell highlights the differential lipid distribution for the different lipids tested. Correlation error maps of the whole images are indicated on the right. Scale bar: $10\mu\text{m}$. Independent experiments per lipid type: 2.

Overall, the study represents a significant advancement in lipid imaging technology. Addressing the critical points above—especially regarding controls for crosslinking specificity, reproducibility, resin artifacts, and statistical robustness—will substantially strengthen the manuscript. Regardless, this reviewer supports moving the study forward for publication.

References

1. Chen, X.-W. *et al.* Time for lipid cell biology. *Nat. Cell Biol.* 27, 169–174 (2025).
2. Iglesias-Artola, J. M. *et al.* Quantitative imaging of lipid transport in mammalian cells. *Nature* 646, 474–482 (2025).
3. Stierhof, Y.-D., Schwarz, H. & Frank, H. Transverse sectioning of plastic-embedded immunolabeled cryosections: Morphology and permeability to protein A-colloidal gold complexes. *J. Ultrastruct. Mol. Struct. Res.* 97, 187–196 (1986).
4. Brorson, S.-H. & Skjorten, F. Mechanism for antigen detection on deplasticized epoxy sections. *Micron* 26, 301–310 (1995).
5. Jao, C. Y., Roth, M., Welti, R. & Salic, A. Metabolic labeling and direct imaging of choline phospholipids in vivo. *Proc. Natl. Acad. Sci.* 106, 15332–15337 (2009).
6. Levental, I. & Lyman, E. Regulation of membrane protein structure and function by their lipid nano-environment. *Nat. Rev. Mol. Cell Biol.* 24, 107–122 (2023).
7. Zhang, F. *et al.* Temporal production of the signaling lipid phosphatidic acid by phospholipase D2 determines the output of extracellular signal-regulated kinase signaling in cancer cells. *Mol. Cell Biol.* 34, 84–95 (2014).
8. Schuhmacher, M. *et al.* Live-cell lipid biochemistry reveals a role of diacylglycerol side-chain composition for cellular lipid dynamics and protein affinities. *Proc. Natl. Acad. Sci. U. S. A.* 117, 7729–7738 (2020).
9. Stockhammer, A. *et al.* ARF1 compartments direct cargo flow via maturation into recycling endosomes. *Nat. Cell Biol.* 26, 1845–1859 (2024).
10. Zhao, Y. & Keen, J. H. Gyrate Clathrin: Highly Dynamic Clathrin Structures Involved in Rapid Receptor Recycling. *Traffic* 9, 2253–2264 (2008).
11. Lennartz, H. M. *et al.* Quantification of lipid sorting during clathrin-mediated endocytosis. 2025.03.04.641423 Preprint at <https://doi.org/10.1101/2025.03.04.641423> (2025).
12. Höglinger, D. *et al.* Trifunctional lipid probes for comprehensive studies of single lipid species in living cells. *Proc. Natl. Acad. Sci.* 114, 1566 (2017).

Visualizing sub-organellar lipid distribution using correlative light and electron microscopy

Corresponding authors: H. Mathilda Lennartz & André Nadler

We would like to thank all reviewers for the constructive feedback, the quick turnaround time and their support of our work. The manuscript has been significantly improved during the review process, and we are grateful for the time and thought that the reviewers put into this process.

Reviewer #1:

Remarks to the Author:

The authors have done a fantastic work to answer my concerns. I agree with the publication strategy using the the technical report option of NCB and that SM partitioning could be studied in later studies. All other points are rigorously clarified in the rebuttal by appropriate references to the previous work of authors, or by additional experiments and information. I therefore now fully support publication at NCB of this manuscript.

We thank reviewer #1 for the positive evaluation of our work and the decision to support our publication strategy

Reviewer #2:

Remarks to the Author:

The authors have addressed all my concerns and I am happy to support publication and I only have a small comment. This is just a suggestion.

I appreciate the inclusion of Extended Data 7, however the title is misleading as SM is simply rather homogeneously distributed over the PM. Maybe a good alternative could be "SM is not enriched in TfR positive clathrin coated pits at the PM"?

Remarks on Protocol(s):

The methods are described in detail

We thank reviewer #2 for the positive evaluation of our work. We agree with the suggestion to rename ED 7 and have revised the manuscript to include the the ED 7 title suggestion of reviewer #2

Reviewer #3:

Remarks to the Author:

This reviewer is satisfied that the authors have adequately addressed all concerns raised in the initial review.

The new data comparing HM20 and K11M resins (Supplementary Figure 9) demonstrates that the key biological findings are conserved across both embedding approaches, which provides confidence that the observations reflect biology rather than technical artifacts. The expanded Figure 2 with quantitative analysis of labeling uniformity and the new Figure 4 showing application across four distinct lipid probes address the concerns about reproducibility and generalizability. The referenced characterization data for probe behavior and crosslinking efficiency are appropriate.

The revised manuscript represents a significant methodological advance for visualizing lipid distribution at the ultrastructural level. This reviewer recommends acceptance for publication.

We thank reviewer #3 for the positive evaluation of our work and the constructive experimental suggestions.

André Nadler

H. Mathilda Lennartz